# A Judge-Aware Ranking Framework
# for Evaluating Large Language Models without Ground Truth

**Mingyuan Xu** [* 1] **Xinzi Tan** [* 1] **Jiawei Wu** [* 1] **Doudou Zhou** [1]

## Abstract

Evaluating large language models (LLMs) on open-ended tasks without ground-truth labels is increasingly done via the LLM-as-a-judge paradigm. A critical but under-modeled issue is that judge LLMs differ substantially in reliability; treating all judges equally can yield biased leaderboards and misleading uncertainty estimates—more data can make evaluation more confidently wrong under misspecified aggregation. We propose a judge-aware ranking framework that extends the Bradley–Terry–Luce model by introducing judge-specific discrimination parameters, jointly estimating latent model quality and judge reliability from pairwise comparisons without reference labels. We establish identifiability up to natural normalizations and prove consistency and asymptotic normality of the maximum likelihood estimator, enabling confidence intervals for score differences and rank comparisons. Across multiple public benchmarks and a newly collected dataset, our method improves agreement with human preferences, achieves higher data efficiency than unweighted baselines, and produces calibrated uncertainty quantification for LLM rankings.

## 1. Introduction

Reliable evaluation of large language models (LLMs) has emerged as a central challenge in AI research. As LLMs are increasingly deployed as general-purpose assistants, the ability to accurately rank and compare their capabilities directly impacts both research progress and deployment decisions. Traditional benchmark suites such as MMLU (Hendrycks et al., 2021) and BIG-Bench (Srivastava et al., 2023) provide valuable measurements of specific competencies through tasks with known ground truth. However, these benchmarks often fail to capture user preferences in open-ended, real-world interactions. For instance, chat models fine-tuned with human feedback can be strongly preferred by users while exhibiting minimal differences on traditional benchmarks (Zheng et al., 2023), highlighting the gap between static evaluation metrics and user satisfaction.

To bridge this gap, evaluation protocols based on pairwise comparisons have gained widespread adoption. Human preference evaluation provides a direct signal but is expensive, time-consuming, and difficult to reproduce (Chiang & Lee, 2023). These limitations have motivated the "LLM-as-a-judge" paradigm, wherein LLMs assess outputs produced by other models (Gilardi et al., 2023; Huang et al., 2023; Zheng et al., 2023; Gu et al., 2026). When carefully prompted, LLM judges can approximate aggregate human preferences on open-ended tasks (Zheng et al., 2023). Moreover, prompting an LLM to directly compare two responses yields more stable judgments than single-answer grading, making pairwise comparison a natural choice for automated evaluation. As a result, LLM-as-a-judge has become a de facto infrastructure for scalable, reference-free LLM evaluation.

A critical but under-examined limitation of this paradigm is that *judge reliability varies substantially across models*. Empirical studies show that different LLM judges exhibit large variability in their ability to replicate human preferences across tasks (Bavaresco et al., 2025). Despite this heterogeneity, most existing ranking pipelines implicitly treat all judges as equally reliable and pool their comparisons uniformly. As we show in this work, such aggregation is *statistically misspecified*: ignoring judge heterogeneity leads to biased rankings and invalid uncertainty quantification. In particular, more data can make the resulting rankings *more confidently wrong*, as confidence intervals shrink around biased estimates under misspecified models.

Classic frameworks such as the Bradley–Terry–Luce (BTL) model (Bradley & Terry, 1952; Luce, 1959) provide a principled foundation for ranking from pairwise comparisons, but assume homogeneous evaluators. Existing approaches for ranking LLMs from automated judgments, such as

[1]Department of Statistics and Data Science, National University of Singapore. Correspondence to: Doudou Zhou <ddzhou@nus.edu.sg>.

*Proceedings of the 43rd International Conference on Machine Learning*, Seoul, South Korea. PMLR 306, 2026. Copyright 2026 by the author(s).

REM (Yang et al., 2024) and GTR/FTR (Dhurandhar et al., 2024), either retain this assumption or rely on heuristic weighting schemes that lack a unified statistical interpretation and do not support valid inference.

In this paper, we propose a **judge-aware ranking framework** that extends the BTL model by introducing judge-specific discrimination parameters, jointly estimating latent model quality scores and judge reliability from pairwise comparisons without requiring ground-truth labels. This formulation automatically down-weights unreliable judges through the likelihood and yields well-defined rankings under heterogeneous evaluators.

We establish identifiability up to natural normalization constraints and prove consistency and asymptotic normality of the maximum likelihood estimator (MLE), enabling principled uncertainty quantification for LLM rankings, including confidence intervals for score differences and rank comparisons.

Empirically, we demonstrate on multiple public benchmarks and a newly collected dataset that judge-aware aggregation improves alignment with human preferences and achieves higher data efficiency than unweighted baselines, particularly in low-budget regimes—with especially clear gains in heterogeneous judge pools where informative and noisy judges coexist.

Our main contributions are summarized as follows: (1) We identify judge heterogeneity as a fundamental source of bias and invalid uncertainty in LLM-as-a-judge rankings; (2) We propose a judge-aware generalization of the BTL model that jointly estimates model quality and judge reliability without ground-truth labels; (3) We establish identifiability and asymptotic theory for the resulting estimator, enabling valid confidence intervals for LLM rankings; and (4) We demonstrate empirically that judge-aware aggregation improves alignment with human preferences and data efficiency compared to unweighted approaches.

The remainder of this paper is organized as follows. Section 2 reviews prior work on statistical ranking models and LLM evaluation. Section 3 formulates the proposed judge-aware model, derives the maximum likelihood estimation algorithm, and establishes asymptotic properties. Section 4 validates the method through numerical simulations. Section 5 reports evaluation results on real-world datasets. Section 6 concludes with discussion and future directions. Our code and data are available on GitHub[1].

---

[1] https://github.com/TanXZfra/
Judge-Aware-Ranking-Framework-for-LLMs

## 2. Related Work

### 2.1. Statistical Models for Pairwise Ranking

Statistical models for pairwise comparisons provide the foundation for ranking problems, with the BTL model (Bradley & Terry, 1952; Luce, 1959) serving as a canonical example. Modern applications have motivated extensions that handle sparse comparison graphs, covariates, and structured heterogeneity. Representative works establish uncertainty quantification under sparse designs (Gao et al., 2023), develop asymptotic theory for covariate-assisted BTL models (Fan et al., 2024b; Yan, 2025), and incorporate sparse intrinsic scores or low-rank structure to capture population-level preference heterogeneity (Fan et al., 2024a; 2025), with applications to reinforcement learning from human feedback (Lee et al., 2024).

These extensions primarily address heterogeneity in preferences (e.g., different user populations) or comparison structure (e.g., sparse graphs and covariates), while continuing to assume that all observed comparisons are equally reliable. In contrast, we focus on heterogeneity in *judge reliability*, a distinct and orthogonal source of variation that directly affects ranking validity and uncertainty quantification when comparisons are produced by multiple LLM judges.

### 2.2. Judge Reliability in Pairwise Ranking

Modeling annotator quality has a long history in crowdsourcing and psychometrics. Chen et al. (2013) propose the Crowd-BT model, which captures annotator reliability through an honesty-type parameter corresponding to label flipping, a formulation tailored to settings with random or adversarial workers. Such models are ill-suited to LLM judges, which rarely behave adversarially but instead differ in their ability to resolve fine-grained quality differences.

Our model adopts a discriminability-type notion of reliability, in which the parameter $\gamma_k$ scales a judge's sensitivity to quality differences rather than flipping labels. This perspective is closer to Item Response Theory (IRT)-based aggregation models. Otani et al. (2016) introduce judge-specific discrimination parameters for aggregating crowdsourced pairwise evaluations of machine translation systems, but their framework relies on comparisons against fixed references and does not consider fully randomized, reference-free designs.

More broadly, existing IRT-inspired approaches either rely on fixed baselines or do not jointly infer model quality and judge reliability from uncalibrated comparisons, nor do they provide asymptotic guarantees for ranking uncertainty. In contrast, we establish consistency, asymptotic normality, and confidence intervals for both model quality and judge reliability in the LLM-as-a-judge setting.

## 2.3. Ranking Large Language Models

The LLM-as-a-judge paradigm has emerged as a scalable alternative to costly human evaluation (Zheng et al., 2023; Gu et al., 2026). Most existing LLM ranking methods implicitly assume homogeneous judges. Representative approaches include Ranking-based automatic Evaluation Method (REM) (Yang et al., 2024), which adapts the Glicko2 rating system for label-free ranking, and Greedy Triplet Ranking (GTR) (Dhurandhar et al., 2024), which aggregates triplet comparisons without accounting for judge quality. Related probabilistic approaches, such as Product-of-Experts aggregation (Fathullah & Gales, 2025), improve uncertainty estimates but do not explicitly model individual judge reliability.

To relax the homogeneity assumption, Dhurandhar et al. (2024) propose the Full Triplet Ranking (FTR) algorithm, which assigns heuristic reputation scores to models based on aggregate win rates and uses these scores when models act as judges. While this introduces adaptive weighting, the resulting rankings lack a formal probabilistic interpretation and do not support principled statistical inference. In addition, FTR uses the evaluated models themselves as judges, whereas our framework does not require the judge pool and candidate model pool to coincide.

Overall, existing methods either assume homogeneous judges or rely on heuristic weighting schemes, obscuring when ranking differences are statistically meaningful. Our framework addresses this gap by explicitly modeling judge reliability and enabling valid uncertainty quantification for LLM rankings.

# 3. Model and Method

We present a judge-aware ranking framework for aggregating pairwise comparisons produced by heterogeneous LLM judges. We introduce the model and design, describe maximum likelihood estimation under suitable normalizations, and establish asymptotic properties enabling valid uncertainty quantification.

## 3.1. Problem Setup and Model

We consider a pairwise comparison study among $N \geq 2$ candidate LLMs evaluated by $K \geq 2$ judges. For each comparison $t = 1, \ldots, T$, we observe a triple $X_t = (i_t, j_t, k_t)$, indicating that judge $k_t$ compares models $i_t$ and $j_t$ ($i_t < j_t$), together with a binary outcome $Y_t \in \{0, 1\}$, where $Y_t = 1$ indicates preference for model $i_t$. Judges may be external evaluators or overlap with the candidate set.

We assume an i.i.d. random design with

$$X_t \overset{\text{i.i.d.}}{\sim} \pi, \quad \pi_{ijk} \geq \pi_{\min} > 0,$$

for all $1 \leq i < j \leq N$ and $k \in [K]$. Thus, all model-pair–judge triples have positive sampling probability. For notational convenience, we represent each unordered pair $\{i, j\}$ as $(i, j)$ with $i < j$.

Conditional on $X_t = (i, j, k)$, the outcome $Y_t$ follows a judge-aware BTL model,

$$\mathbb{P}_\theta(Y_t = 1 \mid X_t = (i, j, k)) = \sigma(\gamma_k(s_i - s_j)), \quad (1)$$

where $\sigma(x) = (1 + e^{-x})^{-1}$. The parameter $\theta = (\mathbf{s}, \boldsymbol{\gamma})$ consists of latent model scores $\mathbf{s} = (s_1, \ldots, s_N)$ and judge-specific discrimination parameters $\boldsymbol{\gamma} = (\gamma_1, \ldots, \gamma_K)$ with $\gamma_k > 0$. The key departure from the standard BTL model is the multiplicative factor $\gamma_k$, which controls how sensitively judge $k$ responds to latent quality differences: when $\gamma_k$ is large, even modest score gaps lead to decisive preferences; when $\gamma_k \approx 0$, the judge's preferences approach random chance and contribute little information. The classical BTL model is recovered as the special case $\gamma_k \equiv 1$ for all judges. Importantly, $\gamma_k$ should be interpreted as discrimination strength relative to the latent ranking learned by the model, not as a measure of objective correctness.

Let $Z_t = (X_t, Y_t)$. Under the above assumptions, $\{Z_t\}_{t=1}^T$ are i.i.d. with joint distribution

$$p_\theta(y, i, j, k) = \pi_{ijk} \left[\sigma(\gamma_k(s_i - s_j))\right]^y \left[1 - \sigma(\gamma_k(s_i - s_j))\right]^{1-y},$$
$$(2)$$

for $y \in \{0, 1\}$, $1 \leq i < j \leq N$, and $1 \leq k \leq K$.

We impose a mild regularity condition to rule out degenerate cases.

**Assumption 3.1** (Non-degenerate quality scores). There exist $i \neq j$ such that $s_i \neq s_j$.

This assumption is necessary because if all quality scores are identical, the outcome distribution becomes independent of $\boldsymbol{\gamma}$, rendering judge discrimination parameters unidentifiable. The next result characterizes identifiability of the model.

**Theorem 3.2** (Identifiability). *Suppose Assumption 3.1 holds. If two parameter pairs $(\mathbf{s}, \boldsymbol{\gamma})$ and $(\tilde{\mathbf{s}}, \tilde{\boldsymbol{\gamma}})$ satisfy*

$$\sigma(\gamma_k(s_i - s_j)) = \sigma(\tilde{\gamma}_k(\tilde{s}_i - \tilde{s}_j)) \quad \text{for all } i \neq j \text{ and all } k,$$

*then there exist constants $a \in \mathbb{R} \setminus \{0\}$ and $b \in \mathbb{R}$ such that $\tilde{\mathbf{s}} = a\mathbf{s} + b\mathbf{1}$ and $\tilde{\boldsymbol{\gamma}} = \boldsymbol{\gamma}/a$. Conversely, any such transformation leaves all comparison probabilities unchanged.*

The theorem implies that model scores are identifiable up to a global shift, while judge discrimination parameters are identifiable up to a common scale. To obtain uniqueness, we impose the normalization

$$\sum_{i=1}^N s_i = 0, \qquad \sum_{k=1}^K \log \gamma_k = 0, \qquad (3)$$

which fixes the location of $\mathbf{s}$ and the scale of $\boldsymbol{\gamma}$.

**Corollary 3.3** (Uniqueness under normalization). *Under Assumption 3.1, imposing the constraints in (3) renders the model identifiable in the usual sense: if two parameter pairs satisfying these constraints induce the same distribution of observations, then they must coincide.*

Finally, we note that although the likelihood depends only on the products $\gamma_k(s_i - s_j)$ and is therefore formally well-defined for signed $\gamma_k$, we restrict attention to $\gamma_k > 0$. This sign convention ensures that larger values of $\gamma_k$ correspond to higher judge discrimination and guarantees that the logarithmic normalization is well-defined.

## 3.2. Maximum Likelihood Estimation

We derive the MLE for the normalized parameter space and describe the optimization procedure used in practice. From the joint distribution (2), the log-likelihood is

$$
\mathcal{L}_{\text{full}}(\mathbf{s}, \boldsymbol{\gamma}) = \sum_{t=1}^{T} \Big[ y_t \log \sigma\big(\gamma_{k_t}(s_{i_t} - s_{j_t})\big)
$$
$$
+ (1 - y_t) \log \big(1 - \sigma(\gamma_{k_t}(s_{i_t} - s_{j_t}))\big) \Big] + C, \tag{4}
$$

where $C = \sum_{t=1}^{T} \log \pi_{i_t, j_t, k_t}$ does not depend on $(\mathbf{s}, \boldsymbol{\gamma})$ and can be ignored for optimization.

For computational efficiency, we aggregate observations by comparison triple. Let $n_{ijk}$ denote the number of times judge $k$ compares models $(i, j)$, and let $\bar{y}_{ijk}$ denote the empirical fraction of times model $i$ is preferred over $j$ by judge $k$. Define the set of observed triples $\Omega = \{(i, j, k) : n_{ijk} > 0\}$. The aggregated log-likelihood can then be written as

$$
\mathcal{L}(\mathbf{s}, \boldsymbol{\gamma}) = \sum_{(i,j,k) \in \Omega} n_{ijk} \Big[ \bar{y}_{ijk} \log \sigma(z_{ijk})
$$
$$
+ (1 - \bar{y}_{ijk}) \log(1 - \sigma(z_{ijk})) \Big], \tag{5}
$$

where $z_{ijk} = \gamma_k(s_i - s_j)$. This representation highlights that the objective is a weighted sum of binary cross-entropy terms, with weights given by the comparison counts $n_{ijk}$. Computationally, each likelihood and gradient evaluation costs $O(T)$ on the raw comparison records, or $O(|\Omega|)$ after aggregating repeated observations over the same $(i, j, k)$ triple, where $|\Omega| \leq K\binom{N}{2}$.

To enforce the positivity constraint $\gamma_k > 0$, we reparameterize $\alpha_k = \log \gamma_k$ and optimize over $(\mathbf{s}, \boldsymbol{\alpha})$ subject to the constraints in (3) (equivalently, $\sum_{k=1}^{K} \alpha_k = 0$). We use projected Adam (Kingma & Ba, 2015) as a numerical solver for the normalized maximum likelihood problem. Adaptive learning rates are particularly helpful here because different comparison triples $(i, j, k)$ may have very different frequencies $n_{ijk}$, resulting in heterogeneous gradient magnitudes. Implementation details are provided in Appendix B.

## 3.3. Asymptotic Properties

We now establish the large-sample properties of the MLE, which form the basis for uncertainty quantification in model ranking. Let $\boldsymbol{\theta}_0 = (\mathbf{s}_0, \boldsymbol{\gamma}_0)$ denote the true parameter in the normalized parameter space, and let $\widehat{\boldsymbol{\theta}}_T = (\widehat{\mathbf{s}}, \widehat{\boldsymbol{\gamma}})$ be the MLE over this normalized parameter space based on $T$ comparisons. For the asymptotic analysis, we work on a compact normalized parameter space whose interior contains the true parameter.

**Theorem 3.4** (Consistency and asymptotic normality). *Under the i.i.d. full-support random design described in Section 3.1, Assumption 3.1, and the compact-localization condition above, the MLE $\widehat{\boldsymbol{\theta}}_T$ satisfies:*

1. ***Consistency:*** $\widehat{\boldsymbol{\theta}}_T \xrightarrow{p} \boldsymbol{\theta}_0$ as $T \to \infty$.

2. ***Asymptotic normality:***

$$
\sqrt{T}\,(\widehat{\boldsymbol{\theta}}_T - \boldsymbol{\theta}_0) \xrightarrow{d} \mathcal{N}(0, \boldsymbol{\Sigma}_{\boldsymbol{\theta}_0}),
$$

*where $\boldsymbol{\Sigma}_{\boldsymbol{\theta}_0}$ is the asymptotic covariance matrix, given explicitly in Appendix A.3.*

Theorem 3.4 enables principled uncertainty quantification for model scores and judge reliability. For any scalar component $\theta_i$ of $\boldsymbol{\theta}$, a $(1 - \varsigma)$ Wald confidence interval is given by

$$
\widehat{\theta}_{T,i} \ \pm \ z_{1-\varsigma/2} \sqrt{\frac{1}{T}\,\widehat{\boldsymbol{\Sigma}}_{\boldsymbol{\theta}, ii}},
$$

where $z_{1-\varsigma/2}$ is the $(1 - \varsigma/2)$ quantile of the standard normal distribution and $\widehat{\boldsymbol{\Sigma}}_{\boldsymbol{\theta}}$ is the plug-in estimate obtained by evaluating the asymptotic covariance at $\widehat{\boldsymbol{\theta}}_T$.

More generally, for any linear functional $h(\boldsymbol{\theta}) = c^\top \boldsymbol{\theta}$ (e.g., score differences $s_i - s_j$),

$$
\sqrt{T}\big(c^\top \widehat{\boldsymbol{\theta}}_T - c^\top \boldsymbol{\theta}_0\big) \xrightarrow{d} \mathcal{N}(0, c^\top \boldsymbol{\Sigma}_{\boldsymbol{\theta}} c).
$$

Replacing $\boldsymbol{\Sigma}_{\boldsymbol{\theta}}$ with $\widehat{\boldsymbol{\Sigma}}_{\boldsymbol{\theta}}$ yields a $(1 - \varsigma)$ Wald confidence interval

$$
c^\top \widehat{\boldsymbol{\theta}}_T \ \pm \ z_{1-\varsigma/2} \sqrt{\frac{1}{T}\,c^\top \widehat{\boldsymbol{\Sigma}}_{\boldsymbol{\theta}} c}.
$$

## 4. Simulation Study

We conduct simulation studies to validate the theoretical results established in Section 3 and to illustrate the practical consequences of modeling judge heterogeneity. Specifically, we address two questions: (i) whether the MLE achieves the $O(1/T)$ convergence rate implied by asymptotic normality, and (ii) whether the proposed judge-aware model yields well-calibrated confidence intervals under the data-generating model, in contrast to the standard BTL model that ignores judge reliability.

## 4.1. Convergence Rate of the MLE

We first examine how estimation error scales with the total number of pairwise comparisons $T$, validating the consistency and rate implied by Theorem 3.4.

For each configuration $(N, K)$, we generate a fixed ground-truth parameter $\boldsymbol{\theta}_0 = (\mathbf{s}_0, \boldsymbol{\gamma}_0)$ satisfying the normalization constraints. The quality scores $\mathbf{s}_0$ and discrimination parameters $\boldsymbol{\gamma}_0$ are drawn to exhibit realistic heterogeneity; details are provided in Appendix C. For each sample size $T$, we generate $T$ independent comparisons according to the judge-aware model (1), with uniformly sampled triples $(i, j, k)$ and outcomes $Y \sim \mathrm{Bernoulli}(\sigma(\gamma_{0,k}(s_{0,i} - s_{0,j})))$. We compute the resulting estimator $(\widehat{\mathbf{s}}^{(T)}, \widehat{\boldsymbol{\gamma}}^{(T)})$ using Algorithm A.1.

Estimation accuracy is measured by the mean squared errors $\mathrm{MSE}_s(T) = \frac{1}{N} \sum_{i=1}^{N} \left( \widehat{s}_i^{(T)} - s_{0,i} \right)^2$ and $\mathrm{MSE}_{\gamma,\log}(T) = \frac{1}{K} \sum_{k=1}^{K} \left( \log \widehat{\gamma}_k^{(T)} - \log \gamma_{0,k} \right)^2$, where the logarithmic scale for $\gamma_k$ ensures scale-invariant comparison. Each $(N, K, T)$ configuration is repeated $R = 100$ times.

Figure 1 plots the MSEs as functions of $T$ on logarithmic scales. After an initial transient regime, both $\mathrm{MSE}_s$ and $\mathrm{MSE}_{\gamma,\log}$ decay approximately linearly in the log–log plot, indicating power-law convergence. To quantify the rate, we fit linear regressions to the last five data points for each curve. The estimated slopes, reported in Table A.3 (see Appendix C), are all close to $-1$, consistent with the theoretical $O(1/T)$ rate implied by asymptotic normality.

### 4.2. Coverage and Width of Confidence Intervals

We evaluate the finite-sample performance of confidence intervals from the judge-aware model and compare them with those from the standard BTL model, which assumes homogeneous judges ($\gamma_k \equiv 1$). For each $(N, K, T)$ configuration, we generate $B = 500$ independent datasets using the same data-generating process as in Section 4.1, fit both models, and construct $95\%$ Wald confidence intervals for the scores $s_i$. We assess performance using empirical coverage and average interval width.

Figure 2 summarizes both empirical coverage probabilities (top) and average interval widths (bottom) as a function of $T$. The judge-aware model maintains coverage close to the nominal $95\%$ level across all sample sizes and configurations. In contrast, the unweighted BTL model exhibits systematic under-coverage that worsens as $T$ increases. This phenomenon arises from model misspecification: by ignoring judge heterogeneity, the unweighted model produces biased estimates whose bias does not vanish asymptotically, while the confidence intervals continue to shrink. At the same time, despite achieving correct coverage, the judge-aware model produces consistently shorter intervals than the

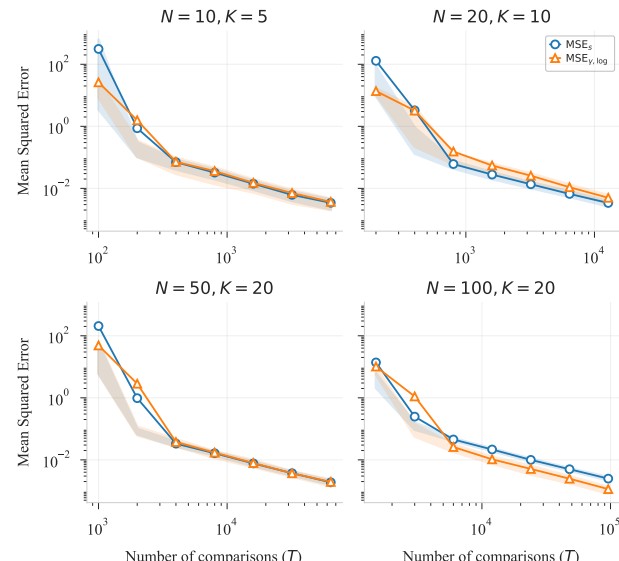

*Figure 1.* MSE versus the number of comparisons $T$ for four configurations $(N, K) \in \{(10, 5), (20, 10), (50, 20), (100, 20)\}$. Both axes are logarithmic. Solid lines show means over $R = 100$ runs, and shaded bands indicate interquartile ranges.

unweighted model. This efficiency gain reflects the model's ability to downweight unreliable judges and extract more information from high-quality comparisons, whereas the unweighted model dilutes informative signals with noise.

## 5. Experiments on LLM Ranking Benchmarks

We evaluate the proposed judge-aware ranking framework on three real-world datasets to assess (i) agreement with established human-preference benchmarks, (ii) statistical efficiency relative to unweighted aggregation, and (iii) robustness of the resulting rankings. We consider MT-Bench and Chatbot Arena (Zheng et al., 2023; Chiang et al., 2024), UltraFeedback (Cui et al., 2024) (evaluated using a panel of 20 selected judge LLMs), and a newly collected in-house dataset with 45 candidate models. For the in-house study, two initially selected judges were unavailable at collection time due to API access issues, resulting in 18 judge LLMs in the final panel. Dataset and judge details are provided in Appendices D and E. Additional results, including estimated judge reliabilities $\widehat{\boldsymbol{\gamma}}$, are deferred to Appendix F.

### 5.1. Experimental Setup

Each dataset provides comparison instances of the form (prompt, $\mathrm{model}_A$, $\mathrm{model}_B$) with two candidate responses. For MT-Bench and Chatbot Arena, we use the original conversations and paired responses. For UltraFeedback, we

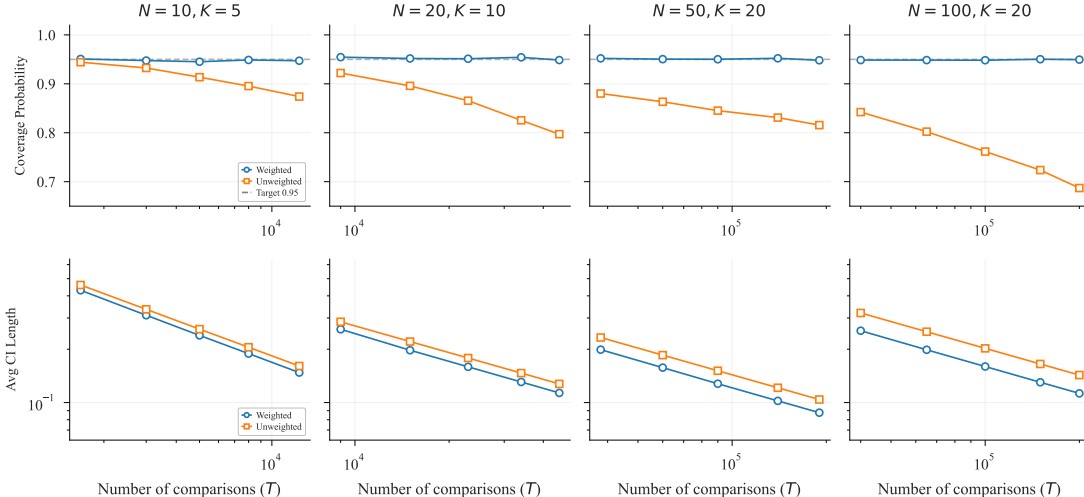

*Figure 2.* Empirical coverage probability (top) and average confidence-interval width (bottom) versus $T$ for four configurations $(N, K) \in \{(10, 5), (20, 10), (50, 20), (100, 20)\}$. In the top panel, the $x$-axis is logarithmic; in the bottom panel, both axes are logarithmic. Each point is based on $B = 500$ replications.

follow the dataset authors' filtering protocol (e.g., removing contaminated or duplicated items) to mitigate leakage.

Each instance is evaluated by an automated judge LLM using a standardized prompt template (Appendix E). In the real datasets, the judge may declare a tie or no preference between the two models. We encode such outcomes as $y = 0.5$ so that, when computing the aggregated response $\bar{y}_{ijk}$, a tie contributes equally to both sides. To verify that this choice does not drive our conclusions, we refit the model after removing tied comparisons; the resulting rankings and confidence intervals remain largely unchanged. Details are provided in Appendix F.4. We use a diverse judge panel spanning multiple model families and sizes to induce realistic heterogeneity in judge reliability. Before fitting, we verify that the comparison graph is connected (via breadth-first search), ensuring all models are comparable through observed comparisons. We fit both the judge-aware model (estimating $\mathbf{s}$ and $\boldsymbol{\gamma}$) and the standard unweighted BTL model (estimating $\mathbf{s}$ with $\gamma_k \equiv 1$) using Algorithm A.1.

### 5.2. Alignment with Established Benchmarks

We first examine whether the learned scores align with established benchmark evaluations. Table 1 compares our fitted scores with the original MT-Bench metrics reported by Zheng et al. (2023). The ranking induced by $\widehat{\mathbf{s}}$ matches the published MT-Bench ordering: GPT-4 ranks first, followed by GPT-3.5, Vicuna-13B, Alpaca-13B, and LLaMA-13B. Both weighted and unweighted fits preserve this monotone ordering, indicating that our approach recovers the benchmark preference structure directly from pairwise judgments without requiring ground-truth labels.

To further validate our model, we compare our estimated scores against UltraFeedback's preference ratings across four evaluation dimensions: helpfulness, truthfulness, honesty, and instruction-following. Figure 3 shows strong positive correlations across all dimensions: models with higher $\widehat{\mathbf{s}}$ consistently receive higher preference scores. This alignment suggests that the fitted scores reflect multi-dimensional quality assessments, beyond a single benchmark-specific ordering.

### 5.3. Efficiency of Weighted Aggregation

We next quantify whether modeling judge reliability improves sample efficiency using our in-house dataset with 45 candidate models and 18 judge LLMs.[2] As full-data references, we fit (i) the weighted judge-aware model and (ii) the unweighted BTL model to the complete dataset (all judges and all comparisons), yielding $\widehat{\mathbf{s}}_w^{\mathrm{ref}}$ and $\widehat{\mathbf{s}}_u^{\mathrm{ref}}$, respectively.

We then run a subsampling study that varies both the judge budget $K$ and the number of comparisons $T$. For each $(K, T)$, we (i) sample $K$ judges uniformly at random from the 18 available, (ii) restrict to $T$ comparisons involving these judges, and (iii) fit both the weighted and unweighted models to obtain $\widehat{\mathbf{s}}_w$ and $\widehat{\mathbf{s}}_u$. Agreement is measured by computing the Pearson correlation between $\widehat{\mathbf{s}}_w$ and $\widehat{\mathbf{s}}_w^{\mathrm{ref}}$ for the weighted model, and between $\widehat{\mathbf{s}}_u$ and $\widehat{\mathbf{s}}_u^{\mathrm{ref}}$ for the unweighted model. Each condition is repeated five times and averaged to reduce variability from judge selection.

Figure 4 displays the average correlation as a function of sample size $T$ for four judge budgets $K \in \{4, 8, 12, 16\}$.

---

[2]We use hosted models from https://www.together.ai/models.

*Table 1.* Evaluation results on MT-Bench. The first three columns are taken from Zheng et al. (2023). The two rightmost columns report fitted scores from the MT-Bench judgment data under the unweighted BTL model ($\hat{s}_u$) and the proposed judge-aware model ($\hat{s}_w$).

| Model | #Token | MMLU | TruthfulQA | MT-Bench Score | $\hat{s}_u$ | $\hat{s}_w$ |
|---|---|---|---|---|---|---|
| LLaMA-13B | 1T | 47.0 | 0.26 | 2.61 | -1.27 | -1.12 |
| Alpaca-13B | 4.4M | 48.1 | 0.30 | 4.53 | -0.62 | -0.54 |
| Vicuna-13B (all) | 370M | 52.1 | **0.35** | 6.39 | -0.29 | -0.25 |
| GPT-3.5 | – | 70.0 | – | 7.94 | 0.51 | 0.43 |
| GPT-4 | – | **86.4** | – | **8.99** | **0.83** | **0.73** |

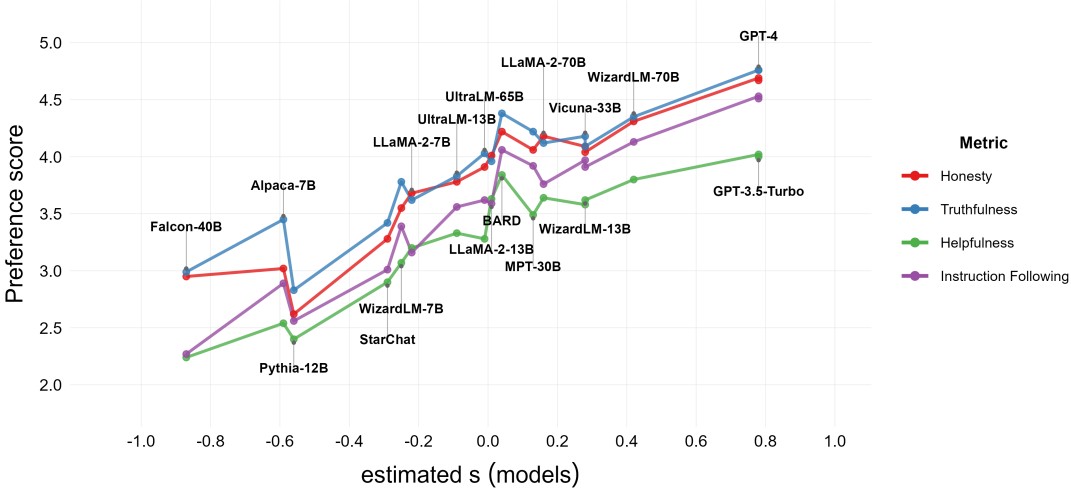

*Figure 3.* UltraFeedback preference scores versus $\hat{s}$ across four evaluation dimensions (helpfulness, truthfulness, honesty, and instruction-following). Each point corresponds to a model; higher fitted scores are associated with higher preference ratings across all dimensions.

Weighted aggregation consistently matches or outperforms unweighted aggregation across all configurations. The improvement is most pronounced in the small-to-moderate sample regime: the weighted model reaches a given correlation level using fewer comparisons, indicating higher statistical efficiency. As $T$ grows, both methods approach near-perfect agreement with their respective full-data references, consistent with diminishing returns when data become abundant.

Several additional patterns emerge. First, correlation increases monotonically with sample size, approaching 1.0 for large $T$. Second, increasing the judge budget $K$ improves performance and reduces the gap between weighted and unweighted methods. These patterns confirm that the weighted model increases statistical efficiency by down-weighting unreliable judges, with the largest gains occurring when data or judges are limited.

Table A.7 reports the complete set of model estimates with 95% confidence intervals under both methods.[3] The average CI width for the weighted model is 0.185, compared to 0.211 for the unweighted model (a reduction of 13.5%).

---

[3]One judge with $\hat{\gamma} \approx 0$ was excluded due to numerical instability; see Appendix F.3 for details.

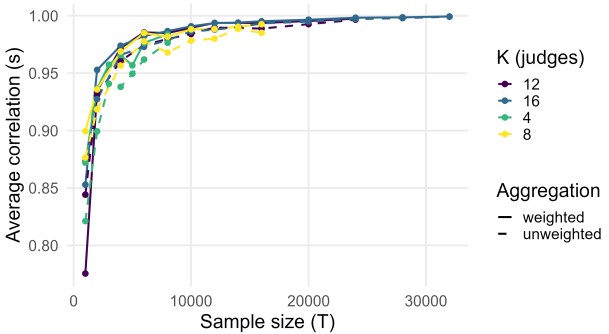

*Figure 4.* Average correlation of estimated s to the ground truth across different numbers of judges and sample sizes.

Similarly, Table 3 indicates that the weighted model yields an average CI width of 0.264, whereas the unweighted model gives 0.278, resulting in a reduction of 5.4%. This confirms the simulation findings (Section 4.2): by concentrating statistical weight on reliable judges, the weighted model extracts more information per comparison, yielding more precise rankings.

## 5.4. Judge heterogeneity

We further conduct a mixed-quality judge experiment. We construct a heterogeneous judge pool consisting of 6 lower-quality judges and 2 higher-quality judges, and restrict the evaluation data to comparisons produced by these selected judges. On this restricted mixed-quality subset, we fit both the weighted and unweighted models using the same set of pairwise comparisons. We then evaluate their estimated scores against the full-data weighted reference scores using Pearson and Spearman correlations, where the reference scores are obtained by fitting the judge-aware weighted model on the complete available comparison dataset.

We observe a clear and consistent improvement when accounting for judge heterogeneity: Pearson correlation increases from $0.8992$ to $0.9394$ and Spearman correlation increases from $0.8316$ to $0.9212$, corresponding to gains of $0.0403$ and $0.0896$, respectively. Relative to the full-data weighted reference scores, the unweighted model shows lower agreement, suggesting that ignoring judge heterogeneity can degrade ranking fidelity in this mixed-quality setting. Notably, the larger improvement in Spearman correlation indicates that modeling judge heterogeneity is particularly important for preserving the relative ordering of models, which is the primary objective of leaderboard construction. Overall, these results provide empirical support for the practical importance of judge heterogeneity and suggest that weighted frameworks can yield more reliable rankings than unweighted baselines, especially in mixed-quality judge settings.

The estimated results in Table 2 clearly demonstrate that weighted model successfully captures heterogeneity in judge quality through the learned scale parameters $\gamma_k$. In particular, kimi-k2-thinking-turbo ($\gamma = 3.511$) and moonshot-v1-128k ($\gamma = 2.914$) receive the two highest $\gamma$ values among all judges. Notably, these two judges were manually identified as high-quality judges a priori, and the model is able to recover this structure purely from pairwise comparison data. In contrast, judges such as deepseek-ai/DeepSeek-R1-Distill-Llama-70B ($\gamma \approx 0.046$) are effectively down-weighted, as their comparisons are close to random noise and provide little useful ranking information.

## 5.5. Case Study: Chatbot Arena

Finally, we study Chatbot Arena to examine whether the judge-aware model yields rankings consistent with widely accepted model relationships. Table 3 reports the top 20 models under both weighted and unweighted fits with 10 judge LLMs.

For the top-tier models, the weighted ranking places the top four models as: GPT-4, Claude-v1, Claude-Instant-v1, and GPT-3.5-Turbo, corresponding to the GPT and Claude families widely regarded as the strongest general-purpose chat models. Notably, the weighted model ranks GPT-4 first, whereas the unweighted model places it third behind both Claude variants. This correction aligns with the broad consensus that GPT-4 was the leading model at the time of data collection.

The weighted ranking also correctly orders the Vicuna family by size: Vicuna-13B (rank 7) > Vicuna-7B (rank 9). The unweighted ranking reverses this relationship in terms of scores (though not ranks), suggesting that judge weighting reduces noise from uneven sampling across model variants.

Moreover, the quartet Koala-13B, Alpaca-13B, Dolly-v2-12B, and LLaMA-13B preserves the same relative ordering reported in the original Chatbot Arena paper. This agreement indicates that the learned judge weights encode evaluation signals consistent with established baselines.

By contrast, the unweighted model exhibits several anomalies: RWKV-4-Raven-14B moves from rank 14 (weighted) to rank 11 (unweighted), and GPT-4 drops from rank 1 to rank 3. These shifts likely reflect the undue influence of a small subset of judges or over-represented comparison contexts. The weighted model mitigates such effects by systematically down-weighting unreliable judges.

Finally, we compare the Spearman correlations between the rankings and the BTL model fitted using human evaluation rankings from (Chiang et al., 2024): the weighted ranking achieves $\rho = 0.9955$, while the unweighted ranking achieves $\rho = 0.9699$. This indicates that the weighted ranking aligns more closely with human judgments.

## 6. Conclusion and Discussion

We introduce a judge-aware ranking framework for evaluating large language models without ground-truth labels. By extending the BTL model with judge-specific discrimination parameters, our approach jointly estimates latent model quality scores and judge-specific discrimination, automatically down-weighting low-discrimination evaluators through the likelihood function. We establish the statistical foundations of the proposed model and demonstrate empirical improvements over unweighted baselines on real-world benchmarks. Beyond evaluating LLMs, the proposed judge-aware framework is also applicable to other settings where items are assessed via pairwise comparisons by judges with heterogeneous discrimination ability, such as expert panels in medicine or peer-review-style evaluations.

Several limitations point to directions for future investigation. The judge-specific discrimination parameter $\gamma_k$ is a global scalar and does not capture task- or domain-specific variation in judge reliability; extending to covariate-dependent discrimination is a natural next step, and Ap-

*Table 2.* $\gamma$ of the judges from the heterogeneous judge pool.

| Judge | $\gamma$ |
|---|---|
| Qwen/Qwen2.5-7B-Instruct-Turbo | 1.419 |
| deepseek-ai/DeepSeek-R1-Distill-Llama-70B | 0.046 |
| moonshot-v1-128k | 2.914 |
| kimi-k2-thinking-turbo | 3.511 |
| meta-llama/Llama-4-Scout-17B-16E-Instruct | 1.872 |
| google/gemma-3n-E4B-it | 0.627 |
| mistralai/Mixtral-8x7B-Instruct-v0.1 | 0.447 |
| Qwen/Qwen3-235B-A22B-Instruct-2507-tput | 2.844 |

*Table 3.* Chatbot Arena rankings: weighted vs. unweighted. We show $\hat{s}$ and confidence intervals for both methods. Subscripts $u$ and $w$ denote the unweighted and weighted fits, respectively.

| $\text{rank}_w$ | $\text{rank}_u$ | Model | $\hat{s}_w$ | $\hat{s}_u$ | $\text{CI}_w^{\text{low}}$ | $\text{CI}_w^{\text{up}}$ | $\text{CI}_u^{\text{low}}$ | $\text{CI}_u^{\text{up}}$ |
|---|---|---|---|---|---|---|---|---|
| 1 | 3 | GPT-4 | 0.728 | 0.942 | 0.515 | 0.941 | 0.820 | 1.064 |
| 2 | 1 | Claude-v1 | 0.725 | 1.106 | 0.513 | 0.938 | 0.978 | 1.234 |
| 3 | 2 | Claude-Instant-v1 | 0.701 | 1.088 | 0.488 | 0.915 | 0.936 | 1.239 |
| 4 | 4 | GPT-3.5-Turbo | 0.431 | 0.616 | 0.296 | 0.566 | 0.507 | 0.724 |
| 5 | 5 | Guanaco-33B | 0.213 | 0.246 | 0.066 | 0.360 | 0.030 | 0.462 |
| 6 | 8 | WizardLM-13B | 0.165 | -0.010 | 0.030 | 0.300 | -0.216 | 0.196 |
| 7 | 6 | Vicuna-13B | 0.159 | 0.078 | 0.086 | 0.231 | -0.017 | 0.173 |
| 8 | 7 | PaLM 2 | 0.131 | 0.020 | 0.047 | 0.216 | -0.107 | 0.146 |
| 9 | 10 | Vicuna-7B | 0.072 | -0.045 | -0.010 | 0.154 | -0.179 | 0.088 |
| 10 | 9 | Koala-13B | -0.038 | -0.029 | -0.098 | 0.023 | -0.126 | 0.068 |
| 11 | 12 | GPT4All-13B-Snoozy | -0.078 | -0.127 | -0.199 | 0.043 | -0.334 | 0.081 |
| 12 | 15 | MPT-7B-Chat | -0.130 | -0.294 | -0.218 | -0.042 | -0.431 | -0.157 |
| 13 | 13 | Alpaca-13B | -0.230 | -0.165 | -0.319 | -0.140 | -0.272 | -0.058 |
| 14 | 11 | RWKV-4-Raven-14B | -0.246 | -0.113 | -0.344 | -0.149 | -0.232 | 0.006 |
| 15 | 17 | OASST-Pythia-12B | -0.259 | -0.429 | -0.353 | -0.165 | -0.532 | -0.325 |
| 16 | 14 | ChatGLM-6B | -0.335 | -0.242 | -0.454 | -0.216 | -0.364 | -0.119 |
| 17 | 16 | FastChat-T5-3B | -0.425 | -0.421 | -0.565 | -0.285 | -0.548 | -0.295 |
| 18 | 18 | Dolly-v2-12B | -0.486 | -0.581 | -0.644 | -0.328 | -0.721 | -0.440 |
| 19 | 20 | StableLM-Tuned-Alpha-7B | -0.516 | -0.822 | -0.681 | -0.350 | -0.969 | -0.676 |
| 20 | 19 | LLaMA-13B | -0.584 | -0.816 | -0.774 | -0.395 | -0.982 | -0.650 |

pendix F.6 provides a preliminary illustration using question length. The current model assumes conditionally independent judge errors given the latent scores and judge-specific parameters, leaving correlated errors or shared biases among similar judges to future work. Finally, Appendix F.7 provides an empirical robustness check under connected sparse designs, while a systematic theory for highly imbalanced, limited-overlap, or adaptive comparison graphs remains future work.

## Acknowledgments

The authors would like to thank the anonymous reviewers and area chairs for their helpful and constructive comments.

This work was supported by the MOE AcRF Tier 1 Grant A-8003569-00-00 and the NUS Start-up Grant A-0009985-00-00.

## Impact Statement

This paper presents a statistical framework for ranking large language models using automated judges, aiming to advance the field of machine learning evaluation. We identify several potential societal implications:

First, our framework reduces the cost and time required for LLM evaluation by enabling scalable automated assessment without human annotation. This democratizes model evaluation, allowing researchers with limited resources to conduct rigorous comparisons. The uncertainty quantification provided by our method also promotes more transparent reporting of ranking reliability.

However, automated evaluation systems, including ours, may inherit or amplify biases present in judge models. Rankings produced by our framework could influence deployment decisions, resource allocation, and public perception of model capabilities. We encourage practitioners to use our confidence intervals to avoid over-interpreting small score differences and to validate rankings against diverse evaluation criteria.

Thus, we recommend using diverse judge panels spanning multiple model families to reduce systematic bias, and we will release our code to enable scrutiny and reproducibility. The judge-specific discrimination parameters estimated by our model can help identify potentially biased or unreliable judges for further investigation.

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

# A. Proofs of Main Results

## A.1. Proof of Theorem 3.2

*Proof of Theorem 3.2. Sufficiency.* For any $a \in \mathbb{R} \backslash \{0\}$ and $b \in \mathbb{R}$,

$$\tilde{\gamma}_k(\tilde{s}_i - \tilde{s}_j) = \frac{\gamma_k}{a}\big(as_i + b - as_j - b\big) = \gamma_k(s_i - s_j),$$

so all choice probabilities remain unchanged.

*Necessity.* If the probabilities coincide for all $i \neq j$ and all $k$, then, since $\sigma$ is strictly increasing,

$$\gamma_k(s_i - s_j) = \tilde{\gamma}_k(\tilde{s}_i - \tilde{s}_j) \qquad (\forall\, i \neq j,\ \forall k).$$

Fix a judge $k$ and a reference item $j_0$. For all $i$,

$$\gamma_k s_i - \tilde{\gamma}_k \tilde{s}_i = \gamma_k s_{j_0} - \tilde{\gamma}_k \tilde{s}_{j_0} =: c_k,$$

and hence

$$\tilde{s}_i = \frac{\gamma_k}{\tilde{\gamma}_k} s_i - \frac{c_k}{\tilde{\gamma}_k} =: a_k s_i + b_k \qquad (\forall\, i).$$

For another judge $\ell$ we similarly obtain $\tilde{s}_i = a_\ell s_i + b_\ell$ for all $i$. Subtracting the two affine representations yields

$$(a_k - a_\ell)s_i + (b_k - b_\ell) = 0 \qquad (\forall\, i).$$

By Assumption 3.1 there exist $i_1 \neq i_2$ with $s_{i_1} \neq s_{i_2}$, so

$$(a_k - a_\ell)(s_{i_1} - s_{i_2}) = 0 \ \Rightarrow\ a_k = a_\ell,$$

and then $b_k = b_\ell$. Thus there exist $a \in \mathbb{R}\backslash\{0\}$ and $b \in \mathbb{R}$ such that $\tilde{\mathbf{s}} = a\mathbf{s} + b\mathbf{1}$. Plugging this into $\gamma_k(s_i - s_j) = \tilde{\gamma}_k(\tilde{s}_i - \tilde{s}_j)$ gives

$$\gamma_k(s_i - s_j) = \tilde{\gamma}_k\, a\, (s_i - s_j) \qquad (\forall\, i \neq j),$$

and hence $a\tilde{\gamma}_k = \gamma_k$ for each $k$, i.e., $\tilde{\boldsymbol{\gamma}} = \boldsymbol{\gamma}/a$. This completes the proof. $\square$

## A.2. Proof of Corollary 3.3

*Proof of Corollary 3.3.* By Theorem 3.2, under Assumption 3.1, if two parameter pairs $(\mathbf{s}, \boldsymbol{\gamma})$ and $(\tilde{\mathbf{s}}, \tilde{\boldsymbol{\gamma}})$ lead to the same probabilities, then there exist $a \in \mathbb{R}\backslash\{0\}$ and $b \in \mathbb{R}$ such that

$$\tilde{\mathbf{s}} = a\mathbf{s} + b\mathbf{1} \qquad \text{and} \qquad \tilde{\boldsymbol{\gamma}} = \boldsymbol{\gamma}/a.$$

Based on the constraints in (3),

$$0 = \sum_{i=1}^{N} \tilde{s}_i = \sum_{i=1}^{N}(as_i + b) = a\sum_{i=1}^{N} s_i + Nb = a \cdot 0 + Nb,$$

we obtain $b = 0$, and hence $\tilde{\mathbf{s}} = a\mathbf{s}$. Next, considering the constraint on the judge discrimination parameters, we have

$$0 = \sum_{k=1}^{K} \log \tilde{\gamma}_k = \sum_{k=1}^{K} \log \frac{\gamma_k}{a} = \sum_{k=1}^{K} (\log \gamma_k - \log a) = 0 - \log a,$$

which implies $\log a = 0$ and therefore $a = 1$. Substituting $a = 1$ and $b = 0$ gives $\tilde{\mathbf{s}} = \mathbf{s}$ and $\tilde{\boldsymbol{\gamma}} = \boldsymbol{\gamma}$, i.e., $(\mathbf{s}, \boldsymbol{\gamma}) = (\tilde{\mathbf{s}}, \tilde{\boldsymbol{\gamma}})$. $\square$

## A.3. Proof of Theorem 3.4

*Proof of Theorem 3.4.* Before the proof, we fix the notation. Let $\mathbf{1}$ denote the all-ones vector (with dimension clear from context). For any integer $M \geq 1$, let $\mathbf{I}_M$ denote the $M \times M$ identity matrix, and let $\mathbf{e}_i \in \mathbb{R}^M$ be the $i$-th standard basis vector, i.e., $(\mathbf{e}_i)_j = \mathbf{1}\{j = i\}$ for $j = 1, \ldots, M$. We use $\mapsto$ to denote a mapping, e.g., $t \mapsto e^t$. We write $\mathrm{Bern}(p)$ for the Bernoulli distribution with success probability $p$. We write $X_n \xrightarrow{p} X$ for convergence in probability and $X_n \xrightarrow{d} X$ for convergence in distribution. For a symmetric matrix $\mathbf{W}$, we write $\mathbf{W} \succ 0$ (resp. $\mathbf{W} \succeq 0$) to mean that $\mathbf{W}$ is positive definite (resp. positive semidefinite). We use $\|\cdot\|$ for the Euclidean norm and $\|\cdot\|_\infty$ for the $\ell_\infty$ norm.

Following the identifiability conditions in Corollary 3.3, we impose the zero-sum constraints

$$\mathbf{1}^\top \mathbf{s} = 0, \qquad \mathbf{1}^\top \boldsymbol{\alpha} = 0, \qquad \text{where} \quad \alpha_k := \log \gamma_k.$$

Let $\mathbf{A_s} \in \mathbb{R}^{N \times (N-1)}$ and $\mathbf{A_\alpha} \in \mathbb{R}^{K \times (K-1)}$ have orthonormal columns spanning the corresponding zero-sum subspaces, i.e.

$$\mathbf{1}^\top \mathbf{A_s} = \mathbf{0}, \ \ \mathbf{A_s}^\top \mathbf{A_s} = \mathbf{I}_{N-1}, \qquad \mathbf{1}^\top \mathbf{A_\alpha} = \mathbf{0}, \ \ \mathbf{A_\alpha}^\top \mathbf{A_\alpha} = \mathbf{I}_{K-1}.$$

Reparameterize

$$\mathbf{s} = \mathbf{A_s}\mathbf{u}, \qquad \boldsymbol{\alpha} = \mathbf{A_\alpha}\mathbf{v}, \qquad \boldsymbol{\vartheta} := (\mathbf{u}, \mathbf{v}) \in \mathbb{R}^d, \quad d = (N-1) + (K-1).$$

This gives a one-to-one linear mapping between the constrained parameter $(\mathbf{s}, \boldsymbol{\alpha})$ and the unconstrained vector $\boldsymbol{\vartheta} \in \mathbb{R}^d$. Let $\boldsymbol{\vartheta}_0$ denote the image of the true parameter $(\mathbf{s}_0, \boldsymbol{\alpha}_0)$.

Write $Z_t = (X_t, Y_t)$ with $X_t = (i_t, j_t, k_t)$ and $\boldsymbol{\Delta}_{ij} := \mathbf{e}_i - \mathbf{e}_j$. Now, we can rewrite the natural parameter (linear predictor) as

$$\eta_{\boldsymbol{\vartheta}}(X_t) = \exp(\alpha_{k_t})(s_{i_t} - s_{j_t}) = \exp(\mathbf{e}_{k_t}^\top \mathbf{A_\alpha}\mathbf{v})\, \boldsymbol{\Delta}_{i_t j_t}^\top \mathbf{A_s}\mathbf{u}.$$

First, we prove the existence of the MLE. Recall the conditional log-likelihood (omitting the design constant $\sum_t \log \pi_{i_t j_t k_t}$):

$$\ell_T(\mathbf{s}, \boldsymbol{\gamma}) = \sum_{t=1}^T \left\{ Y_t\, \gamma_{k_t}(s_{i_t} - s_{j_t}) - \log\big(1 + \exp(\gamma_{k_t}(s_{i_t} - s_{j_t}))\big) \right\}.$$

Under the reparameterization $\mathbf{s} = \mathbf{A_s}\mathbf{u}$, $\boldsymbol{\alpha} = \mathbf{A_\alpha}\mathbf{v}$, $\gamma_k = e^{\alpha_k}$, let $\boldsymbol{\vartheta} = (\mathbf{u}, \mathbf{v})$ and define

$$\ell_T(\boldsymbol{\vartheta}) = \sum_{t=1}^T \left\{ Y_t\, \eta_{\boldsymbol{\vartheta}}(X_t) - \log\big(1 + \exp(\eta_{\boldsymbol{\vartheta}}(X_t))\big) \right\} = \sum_{t=1}^T m(Z_t; \boldsymbol{\vartheta}),$$

with $m(z; \boldsymbol{\vartheta}) = y\, \eta_{\boldsymbol{\vartheta}}(x) - \log\big(1 + e^{\eta_{\boldsymbol{\vartheta}}(x)}\big)$ for $z = (x, y)$. Set the sample and population criteria

$$\widehat{Q}_T(\boldsymbol{\vartheta}) := \frac{1}{T}\ell_T(\boldsymbol{\vartheta}) = \frac{1}{T}\sum_{t=1}^T m(Z_t; \boldsymbol{\vartheta}), \qquad Q_0(\boldsymbol{\vartheta}) := \mathbb{E}[m(Z; \boldsymbol{\vartheta})].$$

By a standard compact-localization argument, we fix a compact set $\Theta \subset \mathbb{R}^d$ containing the true parameter $\vartheta_0$ and consider maximizers of $\widehat{Q}_T$ over $\Theta$.

**Proposition A.1** (Existence of an MLE on $\Theta$). *For every $T \geq 1$, the argmax set $\arg\max_{\boldsymbol{\vartheta} \in \Theta} \widehat{Q}_T(\boldsymbol{\vartheta})$ is nonempty. Hence a (localized) MLE $\widehat{\boldsymbol{\vartheta}}_T \in \Theta$ exists.*

*Proof.* Fix $T \geq 1$ and work on the compact set $\Theta \subset \mathbb{R}^d$. For each fixed $z = (x, y)$ with $x = (i, j, k)$, write

$$\boldsymbol{\Delta}_{ij} := \mathbf{e}_i - \mathbf{e}_j, \qquad a(\boldsymbol{\vartheta}) := \mathbf{e}_k^\top \mathbf{A_\alpha}\mathbf{v}, \qquad b(\boldsymbol{\vartheta}) := \boldsymbol{\Delta}_{ij}^\top \mathbf{A_s}\mathbf{u},$$

so that

$$\eta_{\boldsymbol{\vartheta}}(x) = e^{a(\boldsymbol{\vartheta})}\, b(\boldsymbol{\vartheta}), \qquad m(z; \boldsymbol{\vartheta}) = y\, \eta_{\boldsymbol{\vartheta}}(x) - \log\big(1 + e^{\eta_{\boldsymbol{\vartheta}}(x)}\big).$$

Since $a(\boldsymbol{\vartheta})$ and $b(\boldsymbol{\vartheta})$ are linear in $\boldsymbol{\vartheta} = (\mathbf{u}, \mathbf{v})$, and the maps $t \mapsto e^t$, $(t, s) \mapsto ts$, and $t \mapsto yt - \log(1 + e^t)$ are smooth on $\mathbb{R}$, the composition $\boldsymbol{\vartheta} \mapsto m(z; \boldsymbol{\vartheta})$ is continuous. Consequently $\widehat{Q}_T(\boldsymbol{\vartheta}) = T^{-1}\sum_{t=1}^T m(Z_t; \boldsymbol{\vartheta})$ is continuous on the compact $\Theta$. By the extreme value theorem, $\widehat{Q}_T$ attains its maximum on $\Theta$, so $\arg\max_{\boldsymbol{\vartheta} \in \Theta} \widehat{Q}_T(\boldsymbol{\vartheta}) \neq \varnothing$. $\square$

We now establish the consistency of the (localized) MLE $\widehat{\vartheta}_T$ in the free coordinates $\vartheta \in \mathbb{R}^d$.

**Lemma A.2** (KL characterization of the population criterion). *Let $X = (I, J, K) \sim \pi$ and, under the true parameter $\vartheta_0$, let $Y \mid X \sim \mathrm{Bern}(p_0(X))$ with $p_0(x) := \sigma(\eta_{\vartheta_0}(x))$. For any $\vartheta$, let $p_\vartheta(x) := \sigma(\eta_\vartheta(x))$. Then*

$$Q_0(\vartheta) - Q_0(\vartheta_0) = -\mathbb{E}_{X \sim \pi}\big[\mathrm{KL}\big(\mathrm{Bern}(p_0(X)) \,\|\, \mathrm{Bern}(p_\vartheta(X))\big)\big] \;\leq\; 0,$$

*with equality iff $p_\vartheta(X) = p_0(X)$ $\pi$-a.s. (equivalently $\eta_\vartheta(X) = \eta_{\vartheta_0}(X)$ $\pi$-a.s., since $\sigma$ is strictly increasing).*

*Proof.* Recall $m(z; \vartheta) = y\,\eta_\vartheta(x) - \log(1 + e^{\eta_\vartheta(x)})$ for $z = (x, y)$ with $x = (i, j, k)$. Using the logistic identities

$$\log p_\vartheta(x) = \eta_\vartheta(x) - \log\big(1 + e^{\eta_\vartheta(x)}\big), \qquad \log\big(1 - p_\vartheta(x)\big) = -\log\big(1 + e^{\eta_\vartheta(x)}\big),$$

we can rewrite the single-observation log-likelihood as

$$m(z; \vartheta) = y \log p_\vartheta(x) + (1 - y) \log\big(1 - p_\vartheta(x)\big).$$

Taking conditional expectation given $X = x$ under the true law $Y \mid X = x \sim \mathrm{Bernoulli}(p_0(x))$,

$$\mathbb{E}[m(Z; \vartheta) \mid X = x] = p_0(x) \log p_\vartheta(x) + \big(1 - p_0(x)\big) \log\big(1 - p_\vartheta(x)\big).$$

Similarly,

$$\mathbb{E}[m(Z; \vartheta_0) \mid X = x] = p_0(x) \log p_0(x) + \big(1 - p_0(x)\big) \log\big(1 - p_0(x)\big).$$

Subtracting and using the Bernoulli KL divergence,

$$\mathbb{E}[m(Z; \vartheta) \mid X = x] - \mathbb{E}[m(Z; \vartheta_0) \mid X = x]$$
$$= p_0(x) \log\frac{p_\vartheta(x)}{p_0(x)} + \big(1 - p_0(x)\big) \log\frac{1 - p_\vartheta(x)}{1 - p_0(x)}$$
$$= -\mathrm{KL}\big(\mathrm{Bern}(p_0(x)) \,\|\, \mathrm{Bern}(p_\vartheta(x))\big) \;\leq\; 0.$$

Finally, take expectation over $X \sim \pi$ to obtain

$$Q_0(\vartheta) - Q_0(\vartheta_0) = -\mathbb{E}_{X \sim \pi}\big[\mathrm{KL}\big(\mathrm{Bern}(p_0(X)) \,\|\, \mathrm{Bern}(p_\vartheta(X))\big)\big] \;\leq\; 0.$$

Nonnegativity of KL shows equality holds iff $p_\vartheta(x) = p_0(x)$ for $\pi$-a.e. $x$. Since $\sigma$ is strictly increasing, this is equivalent to $\eta_\vartheta(x) = \eta_{\vartheta_0}(x)$ $\pi$-a.s. $\qquad\square$

**Lemma A.3** (Consistency of the localized MLE). *Let $\widehat{Q}_T(\vartheta) = T^{-1} \sum_{t=1}^{T} m(Z_t; \vartheta)$ and $Q_0(\vartheta) = \mathbb{E}[m(Z; \vartheta)]$. Under the i.i.d. random-design model, compact localization $\vartheta_0 \in \Theta$, and the zero-sum identification, any maximizer $\widehat{\vartheta}_T \in \arg\max_{\vartheta \in \Theta} \widehat{Q}_T(\vartheta)$ satisfies $\widehat{\vartheta}_T \xrightarrow{p} \vartheta_0$.*

*Proof.* We apply the consistency theorem for M-estimators (Newey & McFadden, 1994, Theorem 2.1).

*(1) Compact parameter set.* By construction, we maximize over the compact set $\Theta$.

*(2) Continuity of $Q_0$.* For each fixed $z$, $m(z; \vartheta)$ is continuous in $\vartheta$ (proved in Proposition A.1). Since $X = (I, J, K)$ takes values in the finite set $[N] \times [N] \times [K]$ and $\Theta$ is compact, there exist constants

$$C_\alpha := \sup_{\vartheta \in \Theta} \|\mathbf{A}_{\boldsymbol{\alpha}}\mathbf{v}\|_\infty < \infty, \qquad C_u := \sup_{\vartheta \in \Theta} \|\mathbf{A}_{\mathbf{s}}\mathbf{u}\| < \infty,$$

so that for all $x = (i, j, k)$ and $\vartheta \in \Theta$,

$$|\eta_\vartheta(x)| = \exp(\mathbf{e}_k^\top \mathbf{A}_{\boldsymbol{\alpha}}\mathbf{v})\,|\boldsymbol{\Delta}_{ij}^\top \mathbf{A}_{\mathbf{s}}\mathbf{u}| \leq e^{C_\alpha}\,\|\boldsymbol{\Delta}_{ij}\|\,\|\mathbf{A}_{\mathbf{s}}\mathbf{u}\| \leq e^{C_\alpha}\sqrt{2}\,C_u =: B < \infty.$$

Hence, for all $z = (x, y)$ and $\vartheta \in \Theta$,

$$|m(z; \vartheta)| = \left|y\eta_\vartheta(x) - \log\left(1 + e^{\eta_\vartheta(x)}\right)\right| \leq M_1 := B + \log(1 + e^B) < \infty.$$

Thus $\sup_{\boldsymbol{\vartheta} \in \Theta} |m(Z; \boldsymbol{\vartheta})| \leq M_1$ almost surely, and $M_1$ is integrable. Let $\boldsymbol{\vartheta}_n \to \boldsymbol{\vartheta}$. For each fixed $z$, $m(z; \boldsymbol{\vartheta}_n) \to m(z; \boldsymbol{\vartheta})$ by continuity. By the dominated convergence theorem,

$$Q_0(\boldsymbol{\vartheta}_n) = \mathbb{E}\big[m(Z; \boldsymbol{\vartheta}_n)\big] \longrightarrow \mathbb{E}\big[m(Z; \boldsymbol{\vartheta})\big] = Q_0(\boldsymbol{\vartheta}),$$

so $Q_0$ is continuous on $\Theta$.

*(3) Uniform law of large numbers.* Since $\Theta$ is compact, $m(z; \boldsymbol{\vartheta})$ is continuous in $\boldsymbol{\vartheta}$ (for each fixed $z$), and there exists an integrable envelope $M_1$ with $\sup_{\boldsymbol{\vartheta} \in \Theta} |m(Z; \boldsymbol{\vartheta})| \leq M_1$, the i.i.d. uniform LLN (Newey & McFadden, 1994, Lemma 2.4) gives

$$\sup_{\boldsymbol{\vartheta} \in \Theta} \big|\widehat{Q}_T(\boldsymbol{\vartheta}) - Q_0(\boldsymbol{\vartheta})\big| \xrightarrow{p} 0.$$

*(4) Unique maximizer of $Q_0$.* By Lemma A.2,

$$Q_0(\boldsymbol{\vartheta}) - Q_0(\boldsymbol{\vartheta}_0) = -\mathbb{E}_{X \sim \pi}\Big[\mathrm{KL}\big(\mathrm{Bern}(p_0(X)) \,\|\, \mathrm{Bern}(p_{\boldsymbol{\vartheta}}(X))\big)\Big] \leq 0,$$

with equality iff $\eta_{\boldsymbol{\vartheta}}(X) = \eta_{\boldsymbol{\vartheta}_0}(X)$ $\pi$-a.s. Under the zero-sum (centering and geometric-mean) identification constraints, Corollary 3.3 implies $(\mathbf{s}, \boldsymbol{\alpha}) = (\mathbf{s}_0, \boldsymbol{\alpha}_0)$. Since $(\mathbf{s}, \boldsymbol{\alpha}) = (\mathbf{A_s u}, \mathbf{A_\alpha v})$ defines a one-to-one linear map between $(\mathbf{s}, \boldsymbol{\alpha})$ and $\boldsymbol{\vartheta} = (\mathbf{u}, \mathbf{v})$, we then have $\boldsymbol{\vartheta} = \boldsymbol{\vartheta}_0$. Hence $Q_0$ is uniquely maximized at $\boldsymbol{\vartheta}_0$.

All four conditions hold, so by Newey & McFadden (1994, Theorem 2.1), $\widehat{\boldsymbol{\vartheta}}_T \xrightarrow{p} \boldsymbol{\vartheta}_0$. $\square$

Next, we can prove the central limit theorem for the score function. The per-observation score in the free coordinates is

$$\psi(Z; \boldsymbol{\vartheta}) = \nabla_{\boldsymbol{\vartheta}} m(Z; \boldsymbol{\vartheta}) = \big(Y - \sigma(\eta_{\boldsymbol{\vartheta}}(X))\big) g(X; \boldsymbol{\vartheta}), \qquad g(X; \boldsymbol{\vartheta}) := \nabla_{\boldsymbol{\vartheta}} \eta_{\boldsymbol{\vartheta}}(X). \tag{A.1}$$

For each $\boldsymbol{\vartheta}$, the Fisher information matrix (for the model indexed by $\boldsymbol{\vartheta}$) is

$$\mathcal{I}_{\boldsymbol{\vartheta}}(\boldsymbol{\vartheta}) := \mathbb{E}_{\boldsymbol{\vartheta}}\big[\psi(Z; \boldsymbol{\vartheta})\psi(Z; \boldsymbol{\vartheta})^\top\big] = \mathbb{E}_{\boldsymbol{\vartheta}}\Big[\big(Y - \sigma(\eta_{\boldsymbol{\vartheta}}(X))\big)^2 g(X; \boldsymbol{\vartheta})g(X; \boldsymbol{\vartheta})^\top\Big],$$

where $\mathbb{E}_{\boldsymbol{\vartheta}}$ denotes expectation when $(X, Y)$ is distributed under parameter $\boldsymbol{\vartheta}$.

Using the law of total expectation and $Y \mid X \sim \mathrm{Bernoulli}(\sigma(\eta_{\boldsymbol{\vartheta}}(X)))$ under $P_{\boldsymbol{\vartheta}}$,

$$\mathcal{I}_{\boldsymbol{\vartheta}}(\boldsymbol{\vartheta}) = \mathbb{E}_{\boldsymbol{\vartheta}}\Big[g(X; \boldsymbol{\vartheta})g(X; \boldsymbol{\vartheta})^\top \mathbb{E}_{\boldsymbol{\vartheta}}\big[\big(Y - \sigma(\eta_{\boldsymbol{\vartheta}}(X))\big)^2 \mid X\big]\Big]$$
$$= \mathbb{E}_{\boldsymbol{\vartheta}}\big[\sigma(\eta_{\boldsymbol{\vartheta}}(X))\left(1 - \sigma(\eta_{\boldsymbol{\vartheta}}(X))\right) g(X; \boldsymbol{\vartheta})g(X; \boldsymbol{\vartheta})^\top\big],$$

since for $Y \in \{0, 1\}$ and $p \in (0, 1)$, $\mathbb{E}[(Y - p)^2 \mid X] = p(1 - p)$. At the true parameter $\boldsymbol{\vartheta}_0$, we write

$$\mathcal{I}_{\boldsymbol{\vartheta}}(\boldsymbol{\vartheta}_0) = \mathbb{E}\big[\sigma(\eta_{\boldsymbol{\vartheta}_0}(X))\big(1 - \sigma(\eta_{\boldsymbol{\vartheta}_0}(X))\big) g(X; \boldsymbol{\vartheta}_0)g(X; \boldsymbol{\vartheta}_0)^\top\big].$$

**Lemma A.4** (CLT for the score). *Let $\psi(Z; \boldsymbol{\vartheta})$ and $g(X; \boldsymbol{\vartheta})$ be as above. Under the i.i.d. random-design model and finite design support,*

$$\sqrt{T}\, \nabla_{\boldsymbol{\vartheta}} \widehat{Q}_T(\boldsymbol{\vartheta}_0) = \frac{1}{\sqrt{T}} \sum_{t=1}^{T} \psi(Z_t; \boldsymbol{\vartheta}_0) \xrightarrow{d} \mathcal{N}\big(0, \mathcal{I}_{\boldsymbol{\vartheta}}(\boldsymbol{\vartheta}_0)\big),$$

*where $\mathcal{I}_{\boldsymbol{\vartheta}}(\boldsymbol{\vartheta}_0) = \mathbb{E}\big[\sigma(\eta_{\boldsymbol{\vartheta}_0}(X))(1 - \sigma(\eta_{\boldsymbol{\vartheta}_0}(X))) g(X; \boldsymbol{\vartheta}_0)g(X; \boldsymbol{\vartheta}_0)^\top\big]$.*

*Proof.* Write $\psi_t := \psi(Z_t; \boldsymbol{\vartheta}_0)$.

*Mean zero.* Since $\mathbb{E}[Y_t \mid X_t] = \sigma(\eta_{\boldsymbol{\vartheta}_0}(X_t))$, we have $\mathbb{E}[\psi_t \mid X_t] = 0$ and hence $\mathbb{E}[\psi_t] = 0$.

*Covariance.* By the law of total variance and the fact that $g(X_t; \boldsymbol{\vartheta}_0)$ is $X_t$-measurable,

$$\mathrm{Var}(\psi_t) = \mathbb{E}[\mathrm{Var}(\psi_t \mid X_t)] = \mathbb{E}\Big[\mathrm{Var}\Big((Y_t - \sigma(\eta_{\boldsymbol{\vartheta}_0}(X_t))) g(X_t; \boldsymbol{\vartheta}_0) \,\Big|\, X_t\Big)\Big]$$
$$= \mathbb{E}\big[\sigma(\eta_{\boldsymbol{\vartheta}_0}(X_t))\big(1 - \sigma(\eta_{\boldsymbol{\vartheta}_0}(X_t))\big) g(X_t; \boldsymbol{\vartheta}_0)g(X_t; \boldsymbol{\vartheta}_0)^\top\big] =: \mathcal{I}_{\boldsymbol{\vartheta}}(\boldsymbol{\vartheta}_0).$$

*Finite second moment.* Since $X_t$ takes values in the finite set $[N] \times [N] \times [K]$, there exists $C_g < \infty$ such that $\sup_x \|g(x; \boldsymbol{\vartheta}_0)\| \leq C_g$. Moreover $0 \leq \sigma(u)(1 - \sigma(u)) \leq 1/4$ for all $u \in \mathbb{R}$. Hence

$$\mathbb{E}\|\psi_t\|^2 = \mathbb{E}\big[\sigma(\eta_{\boldsymbol{\vartheta}_0}(X_t))\big(1 - \sigma(\eta_{\boldsymbol{\vartheta}_0}(X_t))\big)\|g(X_t; \boldsymbol{\vartheta}_0)\|^2\big] \leq \tfrac{1}{4}C_g^2 < \infty.$$

*CLT.* The vectors $\{\psi_t\}$ are i.i.d., mean zero, with covariance $\mathcal{I}_{\boldsymbol{\vartheta}}(\boldsymbol{\vartheta}_0)$ and finite second moment. By the multivariate i.i.d. CLT (Lindeberg–Lévy),

$$\frac{1}{\sqrt{T}}\sum_{t=1}^{T}\psi_t \xrightarrow{d} \mathcal{N}\big(0, \mathcal{I}_{\boldsymbol{\vartheta}}(\boldsymbol{\vartheta}_0)\big).$$

Since $\nabla_{\boldsymbol{\vartheta}}\widehat{Q}_T(\boldsymbol{\vartheta}_0) = T^{-1}\sum_{t=1}^{T}\psi_t$, the displayed limit follows. $\qquad\square$

We can then establish a uniform law of large numbers for the Hessian matrix.

**Lemma A.5** (ULLN for the observed information). *Define the per-observation observed information*

$$\mathcal{J}_T(\boldsymbol{\vartheta}) := -\nabla_{\boldsymbol{\vartheta}}^2\widehat{Q}_T(\boldsymbol{\vartheta}) = \frac{1}{T}\sum_{t=1}^{T}h(Z_t; \boldsymbol{\vartheta}), \qquad h(z; \boldsymbol{\vartheta}) := -\nabla_{\boldsymbol{\vartheta}}^2 m(z; \boldsymbol{\vartheta}).$$

*Then, under the i.i.d. random-design model, compact localization $\Theta \ni \boldsymbol{\vartheta}_0$, and finite design support, the following hold:*

*(a)* $\mathcal{J}_T(\boldsymbol{\vartheta}_0) \xrightarrow{p} \mathcal{I}_{\boldsymbol{\vartheta}}(\boldsymbol{\vartheta}_0)$.

*(b) (Uniform LLN)* $\sup_{\boldsymbol{\vartheta} \in \Theta} \big\|\mathcal{J}_T(\boldsymbol{\vartheta}) - \mathbb{E}[h(Z; \boldsymbol{\vartheta})]\big\| \xrightarrow{p} 0$, *and* $\boldsymbol{\vartheta} \mapsto \mathbb{E}[h(Z; \boldsymbol{\vartheta})]$ *is continuous at* $\boldsymbol{\vartheta}_0$.

*(c) Consequently, for any random sequence* $\tilde{\boldsymbol{\vartheta}}_T \xrightarrow{p} \boldsymbol{\vartheta}_0$, $\mathcal{J}_T(\tilde{\boldsymbol{\vartheta}}_T) \xrightarrow{p} \mathcal{I}_{\boldsymbol{\vartheta}}(\boldsymbol{\vartheta}_0)$. *In particular,* $\mathcal{J}_T(\widehat{\boldsymbol{\vartheta}}_T) \xrightarrow{p} \mathcal{I}_{\boldsymbol{\vartheta}}(\boldsymbol{\vartheta}_0)$.

*Proof.* Let $H(X; \boldsymbol{\vartheta}) := \nabla_{\boldsymbol{\vartheta}}^2 \eta_{\boldsymbol{\vartheta}}(X)$. Using (A.1),

$$-\nabla_{\boldsymbol{\vartheta}}^2 m(Z; \boldsymbol{\vartheta}) = \sigma(\eta_{\boldsymbol{\vartheta}}(X))\big(1 - \sigma(\eta_{\boldsymbol{\vartheta}}(X))\big)gg^\top - \big(Y - \sigma(\eta_{\boldsymbol{\vartheta}}(X))\big)H, \tag{A.2}$$

where $g = g(X; \boldsymbol{\vartheta})$ and $H = H(X; \boldsymbol{\vartheta})$.

*(a) Pointwise LLN at $\vartheta_0$.* Taking expectations in (A.2) at $\boldsymbol{\vartheta} = \boldsymbol{\vartheta}_0$ and using $\mathbb{E}[Y - \sigma(\eta_{\boldsymbol{\vartheta}_0}(X)) \mid X] = 0$,

$$\mathbb{E}[h(Z; \boldsymbol{\vartheta}_0)] = \mathbb{E}\big[\sigma(\eta_{\boldsymbol{\vartheta}_0}(X))\big(1 - \sigma(\eta_{\boldsymbol{\vartheta}_0}(X))\big)g(X; \boldsymbol{\vartheta}_0)g(X; \boldsymbol{\vartheta}_0)^\top\big] = \mathcal{I}_{\boldsymbol{\vartheta}}(\boldsymbol{\vartheta}_0).$$

To establish the LLN, we construct an integrable envelope. Since $\Theta$ is compact and $X$ has finite support, the continuous functions $\boldsymbol{\vartheta} \mapsto g(x; \boldsymbol{\vartheta})$ and $\boldsymbol{\vartheta} \mapsto H(x; \boldsymbol{\vartheta})$ are uniformly bounded on $\Theta \times \mathcal{X}$. Define

$$C_g := \sup_{\boldsymbol{\vartheta} \in \Theta}\sup_{x \in \mathcal{X}}\|g(x; \boldsymbol{\vartheta})\| < \infty, \quad C_H := \sup_{\boldsymbol{\vartheta} \in \Theta}\sup_{x \in \mathcal{X}}\|H(x; \boldsymbol{\vartheta})\| < \infty.$$

Using a matrix norm compatible with the vector norm (e.g., the spectral norm satisfying $\|gg^\top\| = \|g\|^2$), and noting that $0 \leq \sigma(1 - \sigma) \leq 1/4$ and $|Y - \sigma| \leq 1$ almost surely, we have

$$\|h(Z; \boldsymbol{\vartheta})\| \leq \sigma(1 - \sigma)\|gg^\top\| + |Y - \sigma|\|H\| \leq \tfrac{1}{4}\|g\|^2 + \|H\| \leq \tfrac{1}{4}C_g^2 + C_H =: M_2 < \infty.$$

Thus $M_2$ is an integrable envelope for $\{h(Z; \boldsymbol{\vartheta}) : \boldsymbol{\vartheta} \in \Theta\}$, and $\{h(Z_t; \boldsymbol{\vartheta}_0)\}$ are i.i.d. with finite first moment. By the LLN for i.i.d. random matrices with finite first moments,

$$\mathcal{J}_T(\boldsymbol{\vartheta}_0) = \frac{1}{T}\sum_{t=1}^{T}h(Z_t; \boldsymbol{\vartheta}_0) \xrightarrow{p} \mathbb{E}[h(Z; \boldsymbol{\vartheta}_0)] = \mathcal{I}_{\boldsymbol{\vartheta}}(\boldsymbol{\vartheta}_0).$$

*(b) Uniform LLN.* We first verify pointwise continuity of $h(z; \boldsymbol{\vartheta})$ in $\boldsymbol{\vartheta}$. Fix $z = (x, y)$ with $x = (i, j, k)$. Recall

$$h(z; \boldsymbol{\vartheta}) = -\nabla_{\boldsymbol{\vartheta}}^2 m(z; \boldsymbol{\vartheta}) = \sigma(\eta_{\boldsymbol{\vartheta}}(x))\big(1 - \sigma(\eta_{\boldsymbol{\vartheta}}(x))\big)g(x; \boldsymbol{\vartheta})g(x; \boldsymbol{\vartheta})^\top - \big(y - \sigma(\eta_{\boldsymbol{\vartheta}}(x))\big)H(x; \boldsymbol{\vartheta}),$$

where $g(x; \boldsymbol{\vartheta}) := \nabla_{\boldsymbol{\vartheta}} \eta_{\boldsymbol{\vartheta}}(x)$ and $H(x; \boldsymbol{\vartheta}) := \nabla_{\boldsymbol{\vartheta}}^2 \eta_{\boldsymbol{\vartheta}}(x)$. Since $\eta_{\boldsymbol{\vartheta}}(x) = \exp(\mathbf{e}_k^\top \mathbf{A_\alpha} \mathbf{v}) \, \boldsymbol{\Delta}_{ij}^\top \mathbf{A_s} \mathbf{u}$ is a composition of linear maps, products, and the exponential, $\boldsymbol{\vartheta} \mapsto \eta_{\boldsymbol{\vartheta}}(x)$ is smooth. Hence $\boldsymbol{\vartheta} \mapsto g(x; \boldsymbol{\vartheta})$ and $\boldsymbol{\vartheta} \mapsto H(x; \boldsymbol{\vartheta})$ are continuous. Because $\sigma(\cdot)$ is smooth, $\boldsymbol{\vartheta} \mapsto \sigma(\eta_{\boldsymbol{\vartheta}}(x))$ and $\vartheta \mapsto \sigma(\eta_{\boldsymbol{\vartheta}}(x))(1 - \sigma(\eta_{\boldsymbol{\vartheta}}(x)))$ are continuous. Therefore $h(z; \boldsymbol{\vartheta})$, being a finite combination of products and sums of continuous terms, is continuous in $\boldsymbol{\vartheta}$.

By part (a), $h(z; \boldsymbol{\vartheta})$ is dominated by the integrable envelope $M_2$ above. Hence, with the compactness of $\Theta$, the i.i.d. uniform LLN (Newey & McFadden, 1994, Lemma 2.4) yields

$$\sup_{\boldsymbol{\vartheta} \in \Theta} \left\| \mathcal{J}_T(\boldsymbol{\vartheta}) - \mathbb{E}[h(Z; \boldsymbol{\vartheta})] \right\| \xrightarrow{p} 0.$$

Moreover, since $h(z; \boldsymbol{\vartheta})$ is continuous in $\boldsymbol{\vartheta}$ and dominated by $M_2$, the map $\boldsymbol{\vartheta} \mapsto \mathbb{E}[h(Z; \boldsymbol{\vartheta})]$ is continuous at $\boldsymbol{\vartheta}_0$ by dominated convergence (the same argument as for $Q_0$ in Lemma A.3).

*(c) Plug-in at a random consistent sequence.* For any random sequence $\tilde{\boldsymbol{\vartheta}}_T \xrightarrow{p} \boldsymbol{\vartheta}_0$,

$$\left\| \mathcal{J}_T(\tilde{\boldsymbol{\vartheta}}_T) - \mathcal{I}_{\boldsymbol{\vartheta}}(\boldsymbol{\vartheta}_0) \right\| = \left\| \mathcal{J}_T(\tilde{\boldsymbol{\vartheta}}_T) - \mathbb{E}[h(Z; \tilde{\boldsymbol{\vartheta}}_T)] + \mathbb{E}[h(Z; \tilde{\boldsymbol{\vartheta}}_T)] - \mathbb{E}[h(Z; \boldsymbol{\vartheta}_0)] \right\|$$
$$\leq \left\| \mathcal{J}_T(\tilde{\boldsymbol{\vartheta}}_T) - \mathbb{E}[h(Z; \tilde{\boldsymbol{\vartheta}}_T)] \right\| + \left\| \mathbb{E}[h(Z; \tilde{\boldsymbol{\vartheta}}_T)] - \mathbb{E}[h(Z; \boldsymbol{\vartheta}_0)] \right\|.$$

By the i.i.d. uniform LLN in (b), $\sup_{\boldsymbol{\vartheta} \in \Theta} \left\| \mathcal{J}_T(\boldsymbol{\vartheta}) - \mathbb{E}[h(Z; \boldsymbol{\vartheta})] \right\| \xrightarrow{p} 0$, so the first term is $o_p(1)$. By continuity of $\boldsymbol{\vartheta} \mapsto \mathbb{E}[h(Z; \boldsymbol{\vartheta})]$ at $\vartheta_0$ and $\tilde{\boldsymbol{\vartheta}}_T \xrightarrow{p} \boldsymbol{\vartheta}_0$ (continuous mapping theorem), the second term is also $o_p(1)$. Therefore $\mathcal{J}_T(\tilde{\boldsymbol{\vartheta}}_T) \xrightarrow{p} \mathcal{I}_{\boldsymbol{\vartheta}}(\boldsymbol{\vartheta}_0)$. In particular, since $\widehat{\boldsymbol{\vartheta}}_T \xrightarrow{p} \boldsymbol{\vartheta}_0$, we have $\mathcal{J}_T(\widehat{\boldsymbol{\vartheta}}_T) \xrightarrow{p} \mathcal{I}_{\boldsymbol{\vartheta}}(\boldsymbol{\vartheta}_0)$. $\square$

Notice the fact that the first-order condition gives $\nabla_{\boldsymbol{\vartheta}} \widehat{Q}_T(\widehat{\boldsymbol{\vartheta}}_T) = 0$. By the mean-value (Taylor) expansion around $\boldsymbol{\vartheta}_0$, there exists $\tilde{\boldsymbol{\vartheta}}_T$ on the line segment between $\widehat{\boldsymbol{\vartheta}}_T$ and $\boldsymbol{\vartheta}_0$ such that

$$0 = \nabla_{\boldsymbol{\vartheta}} \widehat{Q}_T(\vartheta_0) + \nabla_{\boldsymbol{\vartheta}}^2 \widehat{Q}_T(\tilde{\boldsymbol{\vartheta}}_T) \, (\widehat{\boldsymbol{\vartheta}}_T - \boldsymbol{\vartheta}_0).$$

Multiplying by $-\sqrt{T}$ and using $\mathcal{J}_T(\boldsymbol{\vartheta}) := -\nabla_{\boldsymbol{\vartheta}}^2 \widehat{Q}_T(\boldsymbol{\vartheta})$,

$$\sqrt{T} \, (\widehat{\boldsymbol{\vartheta}}_T - \boldsymbol{\vartheta}_0) = \mathcal{J}_T(\tilde{\boldsymbol{\vartheta}}_T)^{-1} \left( \sqrt{T} \, \nabla_{\boldsymbol{\vartheta}} \widehat{Q}_T(\boldsymbol{\vartheta}_0) \right).$$

By Lemma A.3, $\widehat{\boldsymbol{\vartheta}}_T \xrightarrow{p} \boldsymbol{\vartheta}_0$, hence $\tilde{\boldsymbol{\vartheta}}_T \xrightarrow{p} \boldsymbol{\vartheta}_0$ as well. By the score CLT (Lemma A.4), we have $\sqrt{T} \, \nabla_{\boldsymbol{\vartheta}} \widehat{Q}_T(\boldsymbol{\vartheta}_0) \xrightarrow{d} \mathcal{N}(0, \mathcal{I}_{\boldsymbol{\vartheta}}(\boldsymbol{\vartheta}_0))$.

Before applying Slutsky's theorem, we show that $\mathcal{I}_{\boldsymbol{\vartheta}}(\boldsymbol{\vartheta}_0)$ is invertible.

**Lemma A.6** (Weighted Gram representation and invertibility). *Let $X = (I, J, K) \sim \pi$ take values in the finite set*

$$\mathcal{X} = \{(i, j, k): \ 1 \leq i < j \leq N, \ k \in [K]\}, \qquad \mathrm{supp}(\pi) := \{x \in \mathcal{X}: \ \pi(x) > 0\}.$$

*Let $d = (N - 1) + (K - 1)$ be the dimension of the free coordinates $\boldsymbol{\vartheta} = (\mathbf{u}, \mathbf{v})$ under the zero-sum constraints, and stack the row vectors $g(x; \boldsymbol{\vartheta}_0)^\top$ for $x \in \mathrm{supp}(\pi)$ into a matrix $\mathbf{D} \in \mathbb{R}^{S \times d}$, where $S := |\mathrm{supp}(\pi)|$. Define*

$$w(x) := \sigma(\eta_{\boldsymbol{\vartheta}_0}(x))\big(1 - \sigma(\eta_{\boldsymbol{\vartheta}_0}(x))\big), \qquad \mathbf{W} := \mathrm{diag}\big(\{\pi(x) \, w(x)\}_{x \in \mathrm{supp}(\pi)}\big).$$

*Then $\mathbf{W} \succ 0$ and*

$$\mathcal{I}_{\boldsymbol{\vartheta}}(\boldsymbol{\vartheta}_0) = \mathbb{E}\big[w(X) \, g(X; \boldsymbol{\vartheta}_0) g(X; \boldsymbol{\vartheta}_0)^\top\big] = \mathbf{D}^\top \mathbf{W} \mathbf{D}.$$

*Consequently,*

$$\mathcal{I}_{\boldsymbol{\vartheta}}(\boldsymbol{\vartheta}_0) \succ 0 \quad \Longleftrightarrow \quad \mathrm{rank}(\mathbf{D}) = d.$$

*Proof.* With $w(\cdot)$, $\mathbf{D}$, and $\mathbf{W}$ defined in the statement, we have $\mathbf{W} \succ 0$ because $\pi(x) > 0$ for all $x \in \mathrm{supp}(\pi)$ and $w(x) > 0$. Then the Fisher information equals the weighted Gram matrix

$$\mathcal{I}_{\boldsymbol{\vartheta}}(\boldsymbol{\vartheta}_0) = \mathbb{E}\big[w(X) \, g(X; \boldsymbol{\vartheta}_0) g(X; \boldsymbol{\vartheta}_0)^\top\big] = \sum_{x \in \mathrm{supp}(\pi)} \pi(x) \, w(x) \, g(x; \boldsymbol{\vartheta}_0) g(x; \boldsymbol{\vartheta}_0)^\top = \mathbf{D}^\top \mathbf{W} \mathbf{D}.$$

For any $\mathbf{a} \in \mathbb{R}^d$, $\mathbf{a}^\top \mathcal{I}_{\boldsymbol{\vartheta}}(\boldsymbol{\vartheta}_0) \mathbf{a} = \|\mathbf{W}^{1/2} \mathbf{D} \mathbf{a}\|_2^2 \geq 0$, with equality iff $\mathbf{D}\mathbf{a} = \mathbf{0}$, since $\mathbf{W}^{1/2}$ is invertible. Hence $\mathcal{I}_{\boldsymbol{\vartheta}}(\boldsymbol{\vartheta}_0) \succ 0$ if and only if the null space of $\mathbf{D}$ satisfies $\ker(\mathbf{D}) = \{\mathbf{0}\}$, i.e., $\mathrm{rank}(\mathbf{D}) = d$, where $\ker(\mathbf{D}) := \{\mathbf{x} \in \mathbb{R}^d: \ \mathbf{D}\mathbf{x} = \mathbf{0}\}$ denotes the null space of $\mathbf{D}$. $\square$

**Corollary A.7** (Invertibility under full-support design). *Suppose the design triples are drawn i.i.d. from a fixed distribution $\pi$ satisfying*

$$\pi_{ijk} \geq \pi_{\min} > 0 \qquad \text{for all } 1 \leq i < j \leq N,\ k \in [K].$$

*Under Assumption 3.1, the Fisher information $\mathcal{I}_{\vartheta}(\vartheta_0)$ is positive definite and hence invertible.*

*Proof.* Under the i.i.d. full-support design above, $\pi_{ijk} > 0$ for all $1 \leq i < j \leq N$ and $k \in [K]$. Hence the support of $\pi$ is the full set of triples $\mathcal{X} = \{(i,j,k) : 1 \leq i < j \leq N,\ k \in [K]\}$: every unordered item pair $\{i,j\}$ appears (the item graph is complete, hence connected), and every judge $k$ appears with positive probability.

By Assumption 3.1 there exists an item edge $(i^*, j^*)$ with $\Delta_{i^*j^*}^{\top}\mathbf{s}_0 \neq 0$, and $K \geq 2$. Because $\pi_{i^*j^*k} > 0$ for all $k$, the same edge belongs to the population support under every judge. In particular, the row equations are available for two distinct judges $k \neq k'$. If $\mathbf{D}\mathbf{a} = \mathbf{0}$ for some $\mathbf{a} = (\mathbf{a_u}, \mathbf{a_v}) \in \mathbb{R}^d$, we first recall the explicit form of the gradient rows. At the true parameter $\vartheta_0 = (\mathbf{u}_0, \mathbf{v}_0)$,

$$\eta_{\vartheta_0}(i,j,k) = \exp\!\big(\mathbf{e}_k^{\top}\mathbf{A}_{\boldsymbol{\alpha}}\mathbf{v}_0\big)\,\Delta_{ij}^{\top}\mathbf{A_s}\mathbf{u}_0.$$

Hence the gradient with respect to the free coordinates $\vartheta = (\mathbf{u}, \mathbf{v})$ is

$$g(i,j,k;\vartheta_0) = \nabla_{\vartheta}\eta_{\vartheta}(i,j,k)\big|_{\vartheta_0} = \begin{pmatrix} \exp\!\big(\mathbf{e}_k^{\top}\mathbf{A}_{\boldsymbol{\alpha}}\mathbf{v}_0\big)\mathbf{A_s}^{\top}\Delta_{ij} \\ \exp\!\big(\mathbf{e}_k^{\top}\mathbf{A}_{\boldsymbol{\alpha}}\mathbf{v}_0\big)\big(\Delta_{ij}^{\top}\mathbf{A_s}\mathbf{u}_0\big)\mathbf{A}_{\boldsymbol{\alpha}}^{\top}\mathbf{e}_k \end{pmatrix}.$$

The row of $\mathbf{D}$ indexed by $(i,j,k)$ is $g(i,j,k;\vartheta_0)^{\top}$, so the equation $\mathbf{D}\mathbf{a} = \mathbf{0}$ means that, for each $(i,j,k)$,

$$0 = g(i,j,k;\vartheta_0)^{\top}\mathbf{a} = \exp\!\big(\mathbf{e}_k^{\top}\mathbf{A}_{\boldsymbol{\alpha}}\mathbf{v}_0\big)\Big(\big(\mathbf{A_s}^{\top}\Delta_{ij}\big)^{\top}\mathbf{a_u} + \big(\Delta_{ij}^{\top}\mathbf{s}_0\big)\big(\mathbf{A}_{\boldsymbol{\alpha}}^{\top}\mathbf{e}_k\big)^{\top}\mathbf{a_v}\Big).$$

Since $\exp\!\big(\mathbf{e}_k^{\top}\mathbf{A}_{\boldsymbol{\alpha}}\mathbf{v}_0\big) > 0$, we can divide by this scalar and obtain

$$\big(\mathbf{A_s}^{\top}\Delta_{ij}\big)^{\top}\mathbf{a_u} + \big(\Delta_{ij}^{\top}\mathbf{s}_0\big)\big(\mathbf{A}_{\boldsymbol{\alpha}}^{\top}\mathbf{e}_k\big)^{\top}\mathbf{a_v} = 0 \qquad \text{for all } (i,j,k).$$

In particular, writing the row equation for $(i^*, j^*, k)$ and $(i^*, j^*, k')$, we obtain

$$\big(\mathbf{A_s}^{\top}\Delta_{i^*j^*}\big)^{\top}\mathbf{a_u} + \big(\Delta_{i^*j^*}^{\top}\mathbf{s}_0\big)\big(\mathbf{A}_{\boldsymbol{\alpha}}^{\top}\mathbf{e}_k\big)^{\top}\mathbf{a_v} = 0$$

and

$$\big(\mathbf{A_s}^{\top}\Delta_{i^*j^*}\big)^{\top}\mathbf{a_u} + \big(\Delta_{i^*j^*}^{\top}\mathbf{s}_0\big)\big(\mathbf{A}_{\boldsymbol{\alpha}}^{\top}\mathbf{e}_{k'}\big)^{\top}\mathbf{a_v} = 0.$$

Subtracting cancels the item block and yields

$$\big(\Delta_{i^*j^*}^{\top}\mathbf{s}_0\big)\big(\mathbf{A}_{\boldsymbol{\alpha}}^{\top}(\mathbf{e}_k - \mathbf{e}_{k'})\big)^{\top}\mathbf{a_v} = 0 \quad \text{for all } k \neq k'.$$

Since $\Delta_{i^*j^*}^{\top}\mathbf{s}_0 \neq 0$, we obtain

$$\big(\mathbf{A}_{\boldsymbol{\alpha}}^{\top}(\mathbf{e}_k - \mathbf{e}_{k'})\big)^{\top}\mathbf{a_v} = 0 \quad \text{for all } k \neq k'.$$

Recall that the columns of $\mathbf{A}_{\boldsymbol{\alpha}}$ form an orthonormal basis of the judge zero-sum subspace $\mathcal{S} := \{\mathbf{z} \in \mathbb{R}^K : \mathbf{1}^{\top}\mathbf{z} = 0\}$, so the linear map $\mathbf{A}_{\boldsymbol{\alpha}}^{\top} : \mathcal{S} \to \mathbb{R}^{K-1}$ is an isomorphism. Moreover, the vectors $\{\mathbf{e}_k - \mathbf{e}_{k'} : k \neq k'\}$ span $\mathcal{S}$, so their images $\{\mathbf{A}_{\boldsymbol{\alpha}}^{\top}(\mathbf{e}_k - \mathbf{e}_{k'})\}_{k \neq k'}$ span $\mathbb{R}^{K-1}$. Thus $\big(\mathbf{A}_{\boldsymbol{\alpha}}^{\top}(\mathbf{e}_k - \mathbf{e}_{k'})\big)^{\top}\mathbf{a_v} = 0$ for all $k \neq k'$ implies $\mathbf{a_v} = \mathbf{0}$.

With $\mathbf{a_v} = \mathbf{0}$, the remaining equations read $\big(\mathbf{A_s}^{\top}\Delta_{ij}\big)^{\top}\mathbf{a_u} = 0$ for all $1 \leq i < j \leq N$. Since the item graph is complete, the vectors $\{\Delta_{ij} : 1 \leq i < j \leq N\}$ span the item zero-sum subspace in $\mathbb{R}^N$, and therefore $\{\mathbf{A_s}^{\top}\Delta_{ij} : 1 \leq i < j \leq N\}$ span $\mathbb{R}^{N-1}$. Hence $\big(\mathbf{A_s}^{\top}\Delta_{ij}\big)^{\top}\mathbf{a_u} = 0$ for all $i < j$ implies $\mathbf{a_u} = \mathbf{0}$. Thus $\ker(\mathbf{D}) = \{\mathbf{0}\}$, i.e., $\mathrm{rank}(\mathbf{D}) = d$. By Lemma A.6, $\mathbf{D}^{\top}\mathbf{W}\mathbf{D} = \mathcal{I}_{\vartheta}(\vartheta_0) \succ 0$, and the Fisher information is invertible. $\square$

By Lemma A.5(c), $\mathcal{J}_T(\tilde{\vartheta}_T) \xrightarrow{p} \mathcal{I}_{\vartheta}(\vartheta_0)$. Since matrix inversion is continuous on the set of nonsingular matrices and $\mathcal{I}_{\vartheta}(\vartheta_0) \succ 0$ (Corollary A.7), the continuous mapping theorem yields $\mathcal{J}_T(\tilde{\vartheta}_T)^{-1} \xrightarrow{p} \mathcal{I}_{\vartheta}(\vartheta_0)^{-1}$. Therefore, by Slutsky's theorem,

$$\sqrt{T}\,(\hat{\vartheta}_T - \vartheta_0) \xrightarrow{d} \mathcal{N}\big(0,\, \mathcal{I}_{\vartheta}(\vartheta_0)^{-1}\big).$$

Finally, we need to derive the conclusion for $(\mathbf{s}, \boldsymbol{\gamma})$. Recall the reparameterization $\mathbf{s} = \mathbf{A_s} \mathbf{u}$ and $\boldsymbol{\alpha} = \mathbf{A_\alpha} \mathbf{v}$, where $\mathbf{A_s} \in \mathbb{R}^{N \times (N-1)}$ and $\mathbf{A_\alpha} \in \mathbb{R}^{K \times (K-1)}$ have orthonormal columns spanning the item and judge zero-sum subspaces, respectively. Writing $\boldsymbol{\vartheta} = (\mathbf{u}, \mathbf{v})$, collect the linear map as

$$
\begin{pmatrix} \mathbf{s} \\ \boldsymbol{\alpha} \end{pmatrix} = \begin{pmatrix} \mathbf{A_s} & \mathbf{0}_{N \times (K-1)} \\ \mathbf{0}_{K \times (N-1)} & \mathbf{A_\alpha} \end{pmatrix} \begin{pmatrix} \mathbf{u} \\ \mathbf{v} \end{pmatrix} =: \mathbf{J}\,\boldsymbol{\vartheta}, \qquad \mathbf{J} := \mathrm{blkdiag}(\mathbf{A_s}, \mathbf{A_\alpha}),
$$

where $\mathrm{blkdiag}(B, C) := \begin{pmatrix} B & 0 \\ 0 & C \end{pmatrix}$ denotes the block-diagonal concatenation. Let $\boldsymbol{\Sigma}_{\boldsymbol{\vartheta}_0} := \mathcal{I}_{\boldsymbol{\vartheta}}(\boldsymbol{\vartheta}_0)^{-1}$ denote the asymptotic covariance of $\widehat{\boldsymbol{\vartheta}}_T$. By the asymptotic normality in the $(\mathbf{u}, \mathbf{v})$-coordinates and linearity of $\mathbf{J}$,

$$
\sqrt{T} \begin{pmatrix} \widehat{\mathbf{s}} - \mathbf{s}_0 \\ \widehat{\boldsymbol{\alpha}} - \boldsymbol{\alpha}_0 \end{pmatrix} = \mathbf{J}\,\sqrt{T}(\widehat{\boldsymbol{\vartheta}}_T - \boldsymbol{\vartheta}_0) \xrightarrow{d} \mathcal{N}\big(\mathbf{0},\, \boldsymbol{\Sigma}_{(\mathbf{s}_0, \boldsymbol{\alpha}_0)}\big), \qquad \boldsymbol{\Sigma}_{(\mathbf{s}_0, \boldsymbol{\alpha}_0)} = \mathbf{J}\,\boldsymbol{\Sigma}_{\boldsymbol{\vartheta}_0}\,\mathbf{J}^\top.
$$

Next, consider the transformation from $(\mathbf{s}, \boldsymbol{\alpha})$ to $(\mathbf{s}, \boldsymbol{\gamma})$, where $\gamma_k = \exp(\alpha_k)$. Define

$$
g\left( \begin{pmatrix} \mathbf{s} \\ \boldsymbol{\alpha} \end{pmatrix} \right) = \begin{pmatrix} \mathbf{s} \\ \exp(\boldsymbol{\alpha}) \end{pmatrix}, \qquad \exp(\boldsymbol{\alpha}) := (e^{\alpha_1}, \ldots, e^{\alpha_K})^\top.
$$

The Jacobian of $g$ at the true value $(\mathbf{s}_0, \boldsymbol{\alpha}_0)$ is

$$
\nabla g(\mathbf{s}_0, \boldsymbol{\alpha}_0) = \begin{pmatrix} \mathbf{I}_N & \mathbf{0}_{N \times K} \\ \mathbf{0}_{K \times N} & \mathbf{G} \end{pmatrix}, \qquad \mathbf{G} := \mathrm{diag}(\boldsymbol{\gamma}_0),
$$

where $\boldsymbol{\gamma}_0 = \exp(\boldsymbol{\alpha}_0)$ and $\mathrm{diag}(\boldsymbol{\gamma}_0)$ denotes the diagonal matrix with diagonal entries given by the vector $\boldsymbol{\gamma}_0$. Writing the covariance $\boldsymbol{\Sigma}_{(\mathbf{s}_0, \boldsymbol{\alpha}_0)}$ in block form as

$$
\boldsymbol{\Sigma}_{(\mathbf{s}_0, \boldsymbol{\alpha}_0)} = \begin{pmatrix} \boldsymbol{\Sigma}_{ss} & \boldsymbol{\Sigma}_{s\alpha} \\ \boldsymbol{\Sigma}_{\alpha s} & \boldsymbol{\Sigma}_{\alpha\alpha} \end{pmatrix},
$$

the multivariate Delta method applied to $(\mathbf{s}, \boldsymbol{\gamma}) = g(\mathbf{s}, \boldsymbol{\alpha})$ yields

$$
\sqrt{T} \begin{pmatrix} \widehat{\mathbf{s}} - \mathbf{s}_0 \\ \widehat{\boldsymbol{\gamma}} - \boldsymbol{\gamma}_0 \end{pmatrix} \xrightarrow{d} \mathcal{N}\big(\mathbf{0},\, \boldsymbol{\Sigma}_{(\mathbf{s}_0, \boldsymbol{\gamma}_0)}\big),
$$

where

$$
\boldsymbol{\Sigma}_{(\mathbf{s}_0, \boldsymbol{\gamma}_0)} = \nabla g(\mathbf{s}_0, \boldsymbol{\alpha}_0)\, \boldsymbol{\Sigma}_{(\mathbf{s}_0, \boldsymbol{\alpha}_0)}\, \nabla g(\mathbf{s}_0, \boldsymbol{\alpha}_0)^\top = \begin{pmatrix} \boldsymbol{\Sigma}_{ss} & \boldsymbol{\Sigma}_{s\alpha}\mathbf{G} \\ \mathbf{G}\boldsymbol{\Sigma}_{\alpha s} & \mathbf{G}\boldsymbol{\Sigma}_{\alpha\alpha}\mathbf{G} \end{pmatrix}.
$$

This establishes the asymptotic normality of the estimator in the original parameterization $(\mathbf{s}, \boldsymbol{\gamma})$. We denote this asymptotic covariance matrix by $\boldsymbol{\Sigma}_{\boldsymbol{\theta}_0}$, i.e. $\boldsymbol{\Sigma}_{\boldsymbol{\theta}_0} := \boldsymbol{\Sigma}_{(\mathbf{s}_0, \boldsymbol{\gamma}_0)}$.

For clarity, we summarize how the two parts of Theorem 3.4 follow:

1. **Consistency** of $\widehat{\boldsymbol{\theta}}_T$ follows from Lemma A.3 (consistency of the MLE in the free coordinates $\boldsymbol{\vartheta} = (\mathbf{u}, \mathbf{v})$), combined with the linear reparameterization $(\mathbf{s}, \boldsymbol{\alpha}) = \mathbf{J}\boldsymbol{\vartheta}$ and the continuous mapping theorem for $(\mathbf{s}, \boldsymbol{\gamma})$ with $\boldsymbol{\gamma} = \exp(\boldsymbol{\alpha})$.

2. **Asymptotic normality** is obtained by the score CLT in Lemma A.4, the Hessian uniform LLN in Lemma A.5, and the one-step Taylor expansion around $\boldsymbol{\vartheta}_0$, which yield $\sqrt{T}(\widehat{\boldsymbol{\vartheta}}_T - \boldsymbol{\vartheta}_0) \xrightarrow{d} \mathcal{N}(0, \mathcal{I}_{\boldsymbol{\vartheta}}(\boldsymbol{\vartheta}_0)^{-1})$, followed by linear transformations and the multivariate Delta method for $(\mathbf{s}, \boldsymbol{\gamma})$.

$\square$

# B. Algorithm

In Algorithm A.1, $(\cdot)^{\odot 2}$ denotes element-wise squaring, i.e., $x^{\odot 2} = (x_1^2, \ldots, x_m^2)$ for a vector $x \in \mathbb{R}^m$, and $\oslash$ denotes element-wise division. The projection step $\mathbf{s} \leftarrow \mathbf{s} - \frac{1}{N}\mathbf{1}\mathbf{1}^\top \mathbf{s}$ and $\boldsymbol{\alpha} \leftarrow \boldsymbol{\alpha} - \frac{1}{K}\mathbf{1}\mathbf{1}^\top \boldsymbol{\alpha}$ simply centers the iterates to enforce $\sum_i s_i = 0$ and $\sum_k \alpha_k = 0$.

---

**Algorithm A.1** Projected Adam for numerical MLE approximation with judge-specific scales

---

**Input:** Aggregated data $\{n_{ijk}, \bar{y}_{ijk}\}_{(i,j,k)\in\Omega}$, where $\Omega = \{(i,j,k) : 1 \le i < j \le N, \ k \in [K]\}$; number of items $N$; number of judges $K$; Adam parameters $\beta_1, \beta_2, \eta_s, \eta_\alpha, \mathrm{tol}, \varepsilon$; maximum number of iterations $\mathrm{max\_iter}$.

**Initialize:** $\mathbf{s}^{(0)}, \boldsymbol{\alpha}^{(0)}$ (zero-sum); $\mathbf{m}_s^{(0)}, \mathbf{m}_\alpha^{(0)}, \mathbf{v}_s^{(0)}, \mathbf{v}_\alpha^{(0)} \leftarrow \mathbf{0}$

**for** $t = 1, 2, \ldots, \mathrm{max\_iter}$ **do**

   Reset gradients $\mathbf{g}_s \leftarrow \mathbf{0}_N, \mathbf{g}_\alpha \leftarrow \mathbf{0}_K$

   $\boldsymbol{\gamma} \leftarrow \exp(\boldsymbol{\alpha}^{(t-1)})$

   **for all** $(i,j,k) \in \Omega$ **do**

   $z_{ijk} \leftarrow \gamma_k(s_i^{(t-1)} - s_j^{(t-1)}); \quad p_{ijk} \leftarrow \sigma(z_{ijk}); \quad \delta_{ijk} \leftarrow n_{ijk}(\bar{y}_{ijk} - p_{ijk})$

   $\mathbf{g}_s[i] \mathrel{+}= \gamma_k \delta_{ijk}; \quad \mathbf{g}_s[j] \mathrel{-}= \gamma_k \delta_{ijk}; \quad \mathbf{g}_\alpha[k] \mathrel{+}= \delta_{ijk} \cdot z_{ijk}$

   **end for**

   **Update moments:**

   $\mathbf{m}_s^{(t)} \leftarrow \beta_1 \mathbf{m}_s^{(t-1)} + (1-\beta_1)\mathbf{g}_s; \quad \mathbf{m}_\alpha^{(t)} \leftarrow \beta_1 \mathbf{m}_\alpha^{(t-1)} + (1-\beta_1)\mathbf{g}_\alpha$

   $\mathbf{v}_s^{(t)} \leftarrow \beta_2 \mathbf{v}_s^{(t-1)} + (1-\beta_2)\mathbf{g}_s^{\odot 2}; \quad \mathbf{v}_\alpha^{(t)} \leftarrow \beta_2 \mathbf{v}_\alpha^{(t-1)} + (1-\beta_2)\mathbf{g}_\alpha^{\odot 2}$

   **Bias correction:**

   $\widehat{\mathbf{m}}_s^{(t)} \leftarrow \mathbf{m}_s^{(t)}/(1-\beta_1^t), \widehat{\mathbf{v}}_s^{(t)} \leftarrow \mathbf{v}_s^{(t)}/(1-\beta_2^t); \quad \widehat{\mathbf{m}}_\alpha^{(t)} \leftarrow \mathbf{m}_\alpha^{(t)}/(1-\beta_1^t), \widehat{\mathbf{v}}_\alpha^{(t)} \leftarrow \mathbf{v}_\alpha^{(t)}/(1-\beta_2^t)$

   **Gradient-ascent Adam step:**

   $\tilde{\mathbf{s}} \leftarrow \mathbf{s}^{(t-1)} + \eta_s \, \widehat{\mathbf{m}}_s^{(t)} \oslash (\sqrt{\widehat{\mathbf{v}}_s^{(t)}} + \varepsilon); \quad \tilde{\boldsymbol{\alpha}} \leftarrow \boldsymbol{\alpha}^{(t-1)} + \eta_\alpha \, \widehat{\mathbf{m}}_\alpha^{(t)} \oslash (\sqrt{\widehat{\mathbf{v}}_\alpha^{(t)}} + \varepsilon)$

   **Projection:**

   $\mathbf{s}^{(t)} \leftarrow \tilde{\mathbf{s}} - \frac{1}{N}\mathbf{1}\mathbf{1}^\top \tilde{\mathbf{s}}; \quad \boldsymbol{\alpha}^{(t)} \leftarrow \tilde{\boldsymbol{\alpha}} - \frac{1}{K}\mathbf{1}\mathbf{1}^\top \tilde{\boldsymbol{\alpha}}$

   **Stopping criterion:**

   **if** $\max(\|\mathbf{s}^{(t)} - \mathbf{s}^{(t-1)}\|_\infty, \|\boldsymbol{\alpha}^{(t)} - \boldsymbol{\alpha}^{(t-1)}\|_\infty) \le \mathrm{tol}$ **then break**

**end for**

**Return:** $\widehat{\mathbf{s}} = \mathbf{s}^{(t)}, \widehat{\boldsymbol{\gamma}} = \exp(\boldsymbol{\alpha}^{(t)})$

---

### B.1. Gradient Expressions

For each canonical triple $(i,j,k) \in \Omega$, define

$$\ell_{ijk}(\mathbf{s}, \boldsymbol{\alpha}) = n_{ijk}\Big[\bar{y}_{ijk} \log p_{ijk} + (1-\bar{y}_{ijk})\log(1-p_{ijk})\Big], \qquad p_{ijk} = \sigma\big(e^{\alpha_k}(s_i - s_j)\big).$$

The partial derivatives are

$$\frac{\partial \ell_{ijk}}{\partial s_i} = n_{ijk}e^{\alpha_k}(\bar{y}_{ijk} - p_{ijk}), \qquad \frac{\partial \ell_{ijk}}{\partial s_j} = -n_{ijk}e^{\alpha_k}(\bar{y}_{ijk} - p_{ijk}),$$

$$\frac{\partial \ell_{ijk}}{\partial \alpha_k} = n_{ijk}e^{\alpha_k}(\bar{y}_{ijk} - p_{ijk})(s_i - s_j).$$

Algorithm A.1 performs gradient ascent using these expressions (with gradients accumulated over $(i,j,k) \in \Omega$) and enforces the identifiability constraints by centering after each update.

## C. Additional Simulation Details

This section provides the full simulation setup used in Sections 4.1 and 4.2, including the construction of ground-truth parameters, the data generating process, the $T$ values for each $(N, K)$ setting, and additional results omitted from the main text.

### C.1. Ground-Truth Parameter Construction

We generate the ground truth parameter $\boldsymbol{\theta}_0 = (\mathbf{s}_0, \boldsymbol{\gamma}_0)$ as follows. First, we draw $\tilde{s}_i \overset{\text{i.i.d.}}{\sim} \mathcal{N}(0, \sigma_s^2)$ for $i \in [N]$ and set $s_{0,i} = \tilde{s}_i - \frac{1}{N}\sum_{j=1}^N \tilde{s}_j$, so that $\sum_{i=1}^N s_{0,i} = 0$. Next, we draw $\tilde{\eta}_k \overset{\text{i.i.d.}}{\sim} \mathcal{N}(0, \sigma_\gamma^2)$ for $k \in [K]$ and set $\eta_{0,k} = \tilde{\eta}_k - \frac{1}{K}\sum_{\ell=1}^K \tilde{\eta}_\ell$, $\gamma_{0,k} = \exp(\eta_{0,k})$, which implies $\sum_{k=1}^K \log \gamma_{0,k} = 0$. In our experiments, we set $\sigma_s = 1.0$ for all settings. For $\sigma_\gamma$, we use $\sigma_\gamma = 1.5$ for $(N, K) = (10, 5)$ and $\sigma_\gamma = 1.0$ for the other three settings.

## C.2. Data-Generating Process

For a given $(N, K)$ and a target total number of comparisons $T$, we generate integer counts $\{n_{ijk}\}_{1 \leq i < j \leq N, \ 1 \leq k \leq K}$ as follows. We first create a random spanning tree over the $N$ items: for each $i = 2, \ldots, N$, sample a parent $j$ uniformly from $\{1, \ldots, i-1\}$ and a judge $k$ uniformly from $\{1, \ldots, K\}$, and set $n_{jik} \leftarrow 1$. This guarantees that the (undirected) item comparison graph is connected.

Let $R = T - (N-1)$ (we assume $T \geq N - 1$). We then allocate the remaining $R$ comparisons by sampling $i < j$ and $k$ uniformly from all $\binom{N}{2}K$ triples with replacement, and incrementing the corresponding $n_{ijk}$ by the number of times each triple is sampled. Equivalently, the additional counts follow a multinomial distribution over the $\binom{N}{2}K$ triples with equal probabilities. The overall design differs from uniform sampling only through the initial spanning-tree step.

For every $(i, j, k)$ and every replicate $t = 1, \ldots, n_{ijk}$, we draw

$$y_{ijkt} \sim \text{Bernoulli}\big(\sigma\big(\gamma_{0,k}\big(s_{0,i} - s_{0,j}\big)\big)\big),$$

where $\sigma(x) = (1 + \exp(-x))^{-1}$. The specific $T$ values used for each $(N, K)$ configuration are listed in Table A.1 and A.2.

*Table A.1.* Exact simulation settings in Section 4.1.

| $N$ | $K$ | COMPARISON BUDGETS $T$ |
|---|---|---|
| 10 | 5 | $\{100, 200, 400, 800, 1600, 3200, 6400\}$ |
| 20 | 10 | $\{200, 400, 800, 1600, 3200, 6400, 12800\}$ |
| 50 | 20 | $\{1000, 2000, 4000, 8000, 16000, 32000, 64000\}$ |
| 100 | 20 | $\{1500, 3000, 6000, 12000, 24000, 48000, 96000\}$ |

*Table A.2.* Exact simulation settings in Section 4.2.

| $N$ | $K$ | COMPARISON BUDGETS $T$ |
|---|---|---|
| 10 | 5 | $\{1600, 3000, 5000, 8000, 13000\}$ |
| 20 | 10 | $\{9000, 15000, 23000, 34000, 45000\}$ |
| 50 | 20 | $\{38000, 60000, 90000, 140000, 190000\}$ |
| 100 | 20 | $\{40000, 65000, 100000, 150000, 200000\}$ |

## C.3. Empirical log–log Slopes

We fit an ordinary least squares line to the last five $(\log T, \log \text{MSE})$ points for each $(N, K)$ to summarize the empirical scaling behavior. Small deviations of the estimated slope from $-1$ are attributable to finite-sample effects.

*Table A.3.* Empirical log–log slopes estimated from the last five points in Figure 1.

| $(N, K)$ | SLOPE: MSE$(s)$ | SLOPE: MSE$(\log \gamma)$ |
|---|---|---|
| $(10, 5)$ | $-1.109$ | $-1.094$ |
| $(20, 10)$ | $-1.040$ | $-1.223$ |
| $(50, 20)$ | $-1.042$ | $-1.076$ |
| $(100, 20)$ | $-1.050$ | $-1.109$ |

# D. Data

## D.1. MT-bench, Chatbot Arena and UltraFeedback

MT-Bench is a multi-turn, open-question benchmark designed to evaluate conversational and instruction-following capabilities across diverse tasks, serving as a source of evaluation prompts. Based on MT-bench, Chatbot Arena is a crowdsourced

platform that presents the same prompt to two sampled models and records human pairwise preferences via multi-turn interactions, yielding natural pairwise comparison records for our experiments. UltraFeedback is a large-scale collection created by sampling prompts across multiple evaluation layers/dimensions of LLM capability and recording model completions with scalar preference annotations and critiques. Figures A.1-A.3 respectively show the number of comparisons for each model pair in the three datasets.

### D.2. Self-collected Data

We also constructed a self-collected in-house dataset, taking questions from the UltraFeedback pool and collecting responses from 45 candidate models. For each prompt, we recorded 2 randomly selected models' completions. From these records, we sampled unordered model pairs per prompt and formatted each pair using the same anonymized judge prompt template introduced below.

The annotation followed the identical protocol below. Due to API and token-budget constraints, two judge models from the original list: Marin-8B-Instruct and Arcee-Apotlight, were excluded from the in-house annotation runs; all other judges were used with the same temperature and max-token settings as before. The model-fitting procedures remained the same. We report results for this in-house dataset alongside the UltraFeedback and MT-bench experiments. Table A.4 shows the total number of pair-wise comparisons of each dataset we used and Figures A.4 shows the number of comparisons for each model pair in our in-house data.

*Table A.4.* Total number of pairwise comparisons in each dataset.

| Dataset | No. Total Pairwise Comparisons |
| --- | --- |
| MT-Bench | 10K |
| Chatbot Arena | 10K |
| UltraFeedback | 10K |
| In-house Data (ours) | 36K |

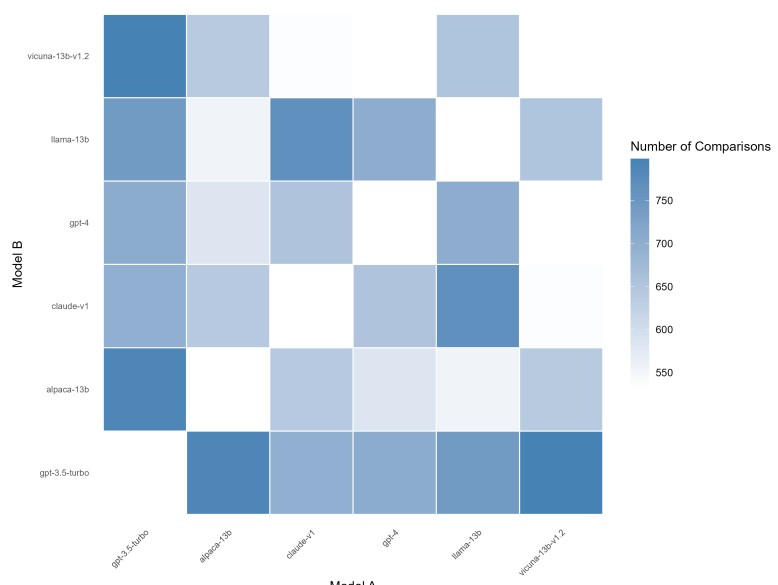

*Figure A.1.* Heatmap of the number of pairwise comparisons for the MT-bench dataset.

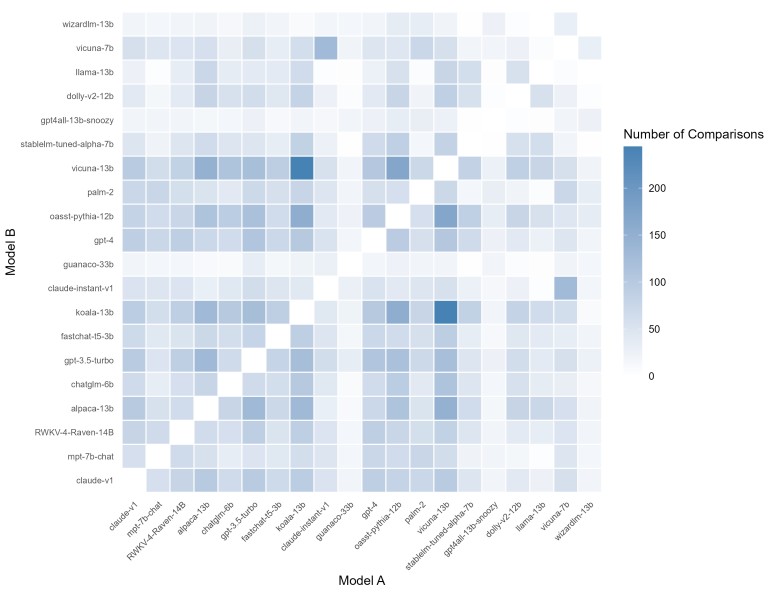

*Figure A.2.* Heatmap of the number of pairwise comparisons for the Chatbot Arena dataset.

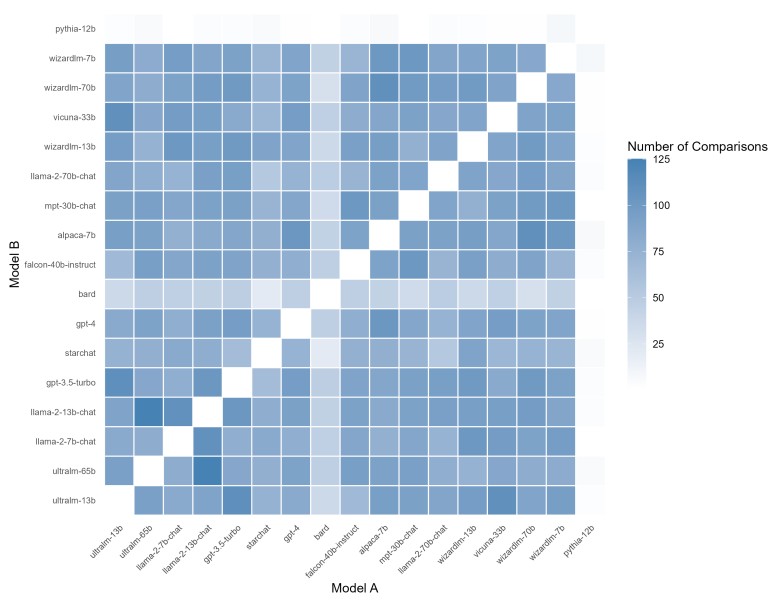

*Figure A.3.* Heatmap of the number of pairwise comparisons for the UltraFeedback dataset.

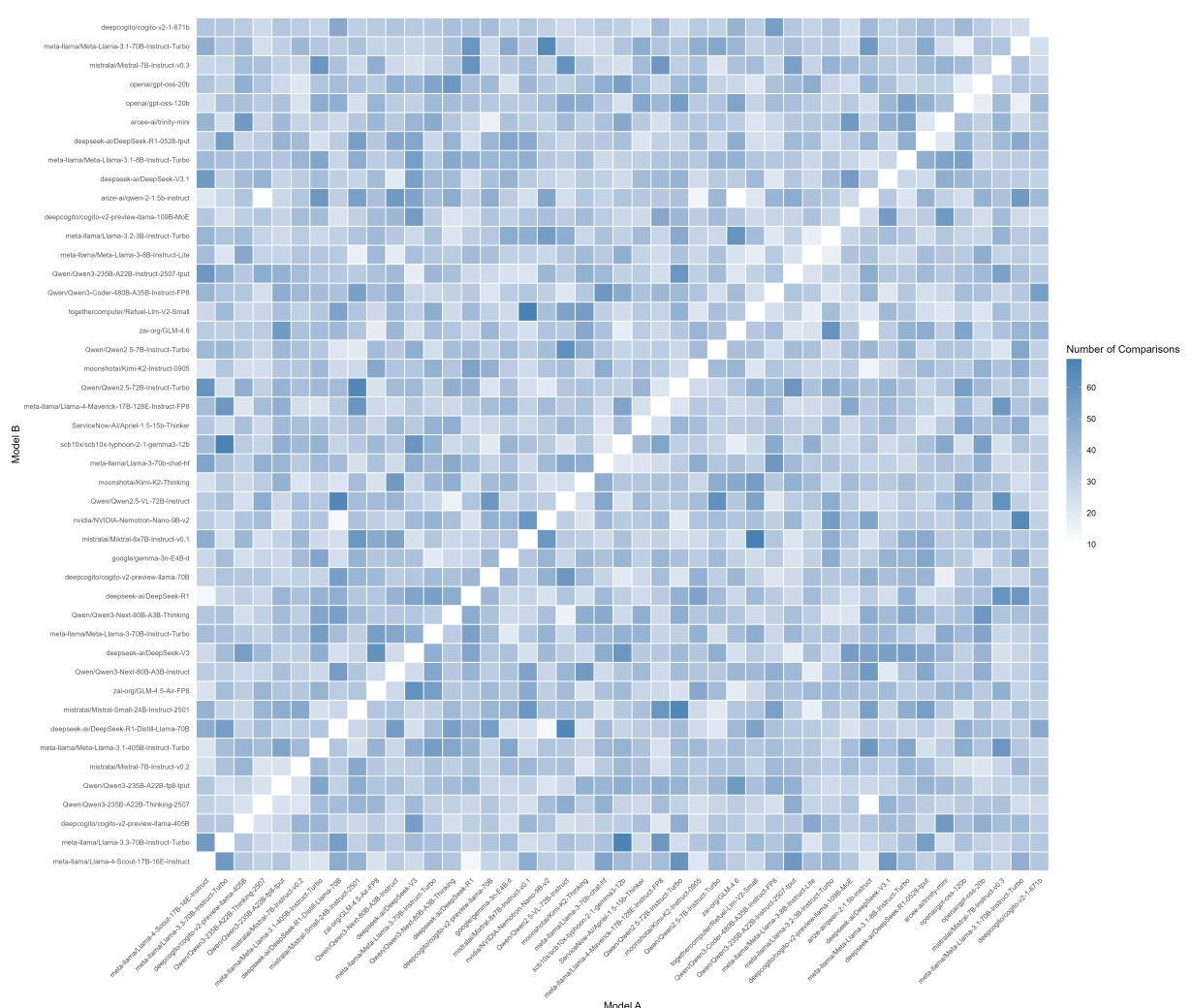

*Figure A.4.* Heatmap of the number of pairwise comparisons for the in-house dataset.

# E. Judge LLMs

## E.1. Selection

To study how different judge LLMs behave and to produce a diverse set of automated judgments, we select 20 judge LLMs (see Table A.5) that cover a range of model sizes, families, and apparent quality levels. The aim is to include both strong and weaker judges so that our fitted pairwise model can learn to accommodate systematic differences among judges. For all judge LLMs, we fix the same decoding hyperparameters (temperature $= 0$ and maximum token limit $= 1024$) to ensure consistency across judgments. We anonymize model names in the prompts, so the judge cannot rely on the identity of the generating model when making a preference decision.

*Table A.5.* Judge LLMs used in the experiments. Models were chosen to span sizes, architectures and producers.

| No. | Judge LLM (identifier) |
|---|---|
| 1 | `openai/gpt-oss-20b` |
| 2 | `Qwen/Qwen3-235B-A22B-Instruct-2507-tput` |
| 3 | `google/gemma-3n-E4B-it` |
| 4 | `meta-llama/Llama-4-Maverick-17B-128E-Instruct-FP8` |
| 5 | `zai-org/GLM-4.5-Air-FP8` |
| 6 | `marin-community/marin-8b-instruct` |
| 7 | `mistralai/Mistral-7B-Instruct-v0.1` |
| 8 | `arcee-ai/arcee-spotlight` |
| 9 | `kimi-k2-0905-preview` |
| 10 | `deepseek-chat` |
| 11 | `meta-llama/Llama-3.3-70B-Instruct-Turbo` |
| 12 | `Qwen/Qwen3-Next-80B-A3B-Instruct` |
| 13 | `deepseek-ai/DeepSeek-R1-Distill-Llama-70B` |
| 14 | `deepcogito/cogito-v2-preview-llama-109B-MoE` |
| 15 | `meta-llama/Llama-4-Scout-17B-16E-Instruct` |
| 16 | `openai/gpt-oss-120b` |
| 17 | `kimi-k2-thinking-turbo` |
| 18 | `moonshot-v1-32k` |
| 19 | `moonshot-v1-128k` |
| 20 | `Qwen/Qwen2.5-7B-Instruct-Turbo` |

## E.2. Judge Prompt Template

We construct a single, fixed-format prompt that includes: (i) the anonymized question, (ii) the two model responses labeled only as "Model A Response" and "Model B Response", and (iii) explicit instructions asking the judge LLM to (a) choose its preferred response ("a", "b", or "c" for tie), and (b) output a scalar confidence in $[0, 1]$ indicating its degree of preference. The instruction explicitly forbids the judge from providing explanatory text or revealing the (anonymized) model identities in its answer, thereby forcing a compact pairwise preference plus confidence output. All judge LLMs receive the same template and instructions, and we use identical decoding hyperparameters (temperature, max tokens) across judges. The detailed prompt template is shown in Figure A.5

# F. Extra Results

## F.1. Comparison with Ultrafeedback ranking

Table A.6 summarizes the evaluation results on the UltraFeedback dataset with 10K pairwise comparisons. The first three columns (AlpacaEval, Evol-Instruct, UltraChat) report the scores directly taken from UltraFeeback, while the last two columns show our model estimates: $\widehat{s}_u$ and $\widehat{s}_w$. We observe that the model rankings produced by our method generally align with the UltraFeedback scores. For example, WizardLM-13B achieves high scores across all three UltraFeedback metrics and attains the highest estimated $\widehat{s}$ in both the unweighted and weighted versions, whereas ULTRALM-13B consistently shows lower preference and, correspondingly, negative $\widehat{s}$ values.

```
Please act as an impartial judge and evaluate the quality of the responses provided by two AI
assistants to the user question displayed below. You should choose the assistant that follows
the user's instructions and answers the user's question better. Your evaluation should
consider factors such as the helpfulness, relevance, accuracy, depth, creativity, and level
of detail of their responses. Avoid any position biases and ensure that the order in which
the responses were presented does not influence your decision. Do not allow the length of the
responses to influence your evaluation. Do not favor certain names of the assistants. Be as
objective as possible.

Do NOT provide any explanation, justification, reasoning, or chain-of-thought. Do NOT output
any extra text other than the single-line verdict described below.

<CHOICE> <CONFIDENCE>

- <CHOICE> must be one of: a, b, or c    (a = model_a is better; b = model_b is better; c =
tie)

- <CONFIDENCE> must be a decimal number between 0.00 and 1.00 with exactly two digits after
the decimal point. The number expresses your confidence in the choice (higher means more
confident).

Examples of valid outputs (each must be exactly one line):"

a 0.85

b 0.60

c 0.40

Now evaluate the following:

User question: {user_question}

model_a response: {a_answer}

model_b response: {b_answer}
```

*Figure A.5.* Template of the prompt provided to the judge LLMs

*Table A.6.* Evaluation Results on UltraFeedback Data. $\widehat{s}_u$ denotes the estimated score under the unweighted BTL model, and $\widehat{s}_w$ denotes the estimated score under the judge-aware weighted model.

| Model | Size | AlpacaEval | Evol-Instruct | UltraChat | $\widehat{s}_u$ | $\widehat{s}_w$ |
|---|---|---|---|---|---|---|
| LLaMA2-13B | 13B | 81.1 | 44.1 | 34.5 | 0.09 | 0.01 |
| WizardLM-13B | 13B | 89.2 | 55.5 | **59.7** | **0.29** | **0.28** |
| LLaMA2-70B | 70B | **92.7** | **56.4** | 54.0 | 0.23 | 0.16 |
| UltraLM-13B | 13B | 80.7 | 39.9 | 38.2 | -0.17 | -0.09 |
| Vicuna-33B | 33B | 89.0 | 50.0 | 57.7 | 0.25 | 0.28 |

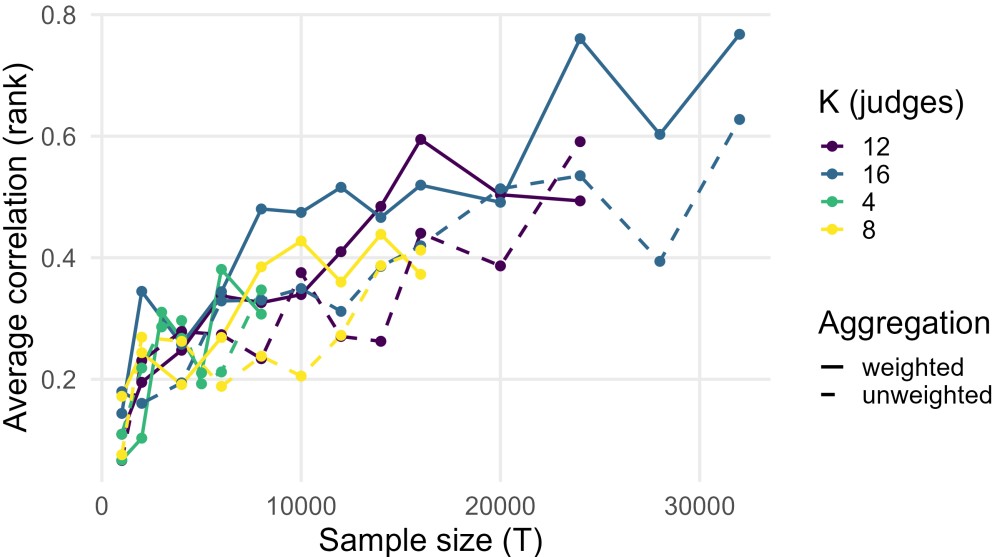

*Figure A.6.* Average Spearman correlation of rankings to the ground truth across different numbers of judges and sample sizes.

### F.2. Rank-based Correlation Analysis

Based on Section 5.3, the rank-based results in Figure A.6 exhibit a clear interaction between total sample size $T$ and judge budget $K$. When the total number of pairwise comparisons is small, increasing $K$ can degrade performance: with fixed $T$ the per-judge contribution falls roughly as $T/K$, so larger $K$ produces noisier individual-judge summaries and therefore a noisier aggregated ranking. In other words, under a tight data budget, spreading comparisons across many judges reduces the signal available from each judge and may harm the estimated ranking.

By contrast, once the overall sample size is sufficiently large the pattern reverses: larger $K$ monotonically improves rank correlations with the gold reference. With ample comparisons per judge the additional judges primarily reduce estimator variance and increase robustness to idiosyncratic judge bias, so the aggregated ranking benefits from greater panel diversity. These two regimes: a small-$T$ regime in which a modest $K$ preserves per-judge signal, and a large-$T$ regime in which large $K$ reduces variance, suggest a practical trade-off when allocating annotation budget and motivate choosing $K$ in proportion to the available sample size rather than fixing $K$ independently of $T$.

### F.3. Complete Ranking on In-house Data

Table A.7 reports the complete ranking of the 45 models from the in-house data, associated with the $\widehat{s}$ estimated from the weighted and unweighted models. The 95% confidence intervals of the estimates and the ranking differences are also reported. Note that the $\widehat{\gamma}$ of GLM-4.5-Air-FP8 was approximately 0, which caused numerical instability for confidence interval calculation. Hence, it is excluded from the final calculation.

### F.4. Robustness to Tie Handling

The resulting rankings remain unchanged on MT-Bench, Chatbot Arena, and UltraFeedback. On the in-house dataset, only two local swaps occur: Qwen/Qwen3-Next-80B-A3B-Thinking exchanges position with google/gemma-3n-E4B-it, and deepcogito/cogito-v2-preview-llama-70B exchanges position with deepcogito/cogito-v2-preview-llama-405B. The uncertainty estimates are also stable: the average confidence interval width on the in-house dataset increases only modestly, from $0.185$ to $0.195$, as expected because removing ties reduces the effective number of comparisons.

### F.5. Results on Judges

We investigated whether judge-specific discrimination scales ($\widehat{\gamma}$) correlate with model size (number of parameters). Table A.8 lists the fitted $\widehat{\gamma}$ values for the two evaluation datasets (MT-bench and UltraFeedback) and the approximate model sizes where publicly available.

*Table A.7.* Comparison of weighted and unweighted rankings with 95% confidence intervals. Subscripts $u$ and $w$ denote the unweighted and weighted fits, respectively.

| Model | $\hat{s}_w$ | $\hat{s}_u$ | $CI_w^{low}$ | $CI_w^{up}$ | $CI_u^{low}$ | $CI_u^{up}$ | $rank_w$ | $rank_u$ | $rank_{diff}$ |
|---|---|---|---|---|---|---|---|---|---|
| Qwen/Qwen3-Next-80B-A3B-Instruct | 0.747 | 0.949 | 0.597 | 0.897 | 0.839 | 1.059 | 1 | 1 | 0 |
| openai/gpt-oss-120b | 0.637 | 0.827 | 0.505 | 0.768 | 0.719 | 0.935 | 2 | 2 | 0 |
| moonshotai/Kimi-K2-Instruct-0905 | 0.593 | 0.717 | 0.467 | 0.719 | 0.609 | 0.826 | 3 | 4 | 1 |
| moonshotai/Kimi-K2-Thinking | 0.577 | 0.743 | 0.454 | 0.700 | 0.634 | 0.852 | 4 | 3 | -1 |
| Qwen/Qwen3-235B-A22B-Instruct-2507-tput | 0.520 | 0.697 | 0.406 | 0.633 | 0.591 | 0.803 | 5 | 5 | 0 |
| zai-org/GLM-4.6 | 0.520 | 0.678 | 0.406 | 0.633 | 0.571 | 0.785 | 6 | 6 | 0 |
| openai/gpt-oss-20b | 0.396 | 0.504 | 0.301 | 0.491 | 0.400 | 0.607 | 7 | 7 | 0 |
| deepseek-ai/DeepSeek-V3.1 | 0.360 | 0.463 | 0.269 | 0.452 | 0.359 | 0.568 | 8 | 9 | 1 |
| deepseek-ai/DeepSeek-R1 | 0.353 | 0.475 | 0.262 | 0.443 | 0.371 | 0.579 | 9 | 8 | -1 |
| deepseek-ai/DeepSeek-R1-0528-tput | 0.323 | 0.354 | 0.237 | 0.410 | 0.252 | 0.457 | 10 | 12 | 2 |
| deepseek-ai/DeepSeek-V3 | 0.308 | 0.432 | 0.225 | 0.391 | 0.331 | 0.534 | 11 | 10 | -1 |
| Qwen/Qwen3-Coder-480B-A35B-Instruct-FP8 | 0.254 | 0.367 | 0.175 | 0.334 | 0.262 | 0.471 | 12 | 11 | -1 |
| Qwen/Qwen3-235B-A22B-fp8-tput | 0.226 | 0.232 | 0.152 | 0.301 | 0.130 | 0.334 | 13 | 17 | 4 |
| Qwen/Qwen3-235B-A22B-Thinking-2507 | 0.224 | 0.255 | 0.145 | 0.302 | 0.148 | 0.363 | 14 | 16 | 2 |
| arcee-ai/trinity-mini | 0.198 | 0.300 | 0.124 | 0.272 | 0.196 | 0.404 | 15 | 13 | -2 |
| deepcogito/cogito-v2-1-671b | 0.170 | 0.255 | 0.097 | 0.242 | 0.150 | 0.360 | 16 | 15 | -1 |
| google/gemma-3n-E4B-it | 0.156 | 0.266 | 0.085 | 0.226 | 0.162 | 0.369 | 17 | 14 | -3 |
| Qwen/Qwen3-Next-80B-A3B-Thinking | 0.155 | 0.208 | 0.086 | 0.224 | 0.106 | 0.309 | 18 | 18 | 0 |
| ServiceNow-AI/Apriel-1.5-15b-Thinker | 0.144 | 0.204 | 0.074 | 0.215 | 0.100 | 0.309 | 19 | 19 | 0 |
| deepcogito/cogito-v2-preview-llama-405B | 0.115 | 0.144 | 0.046 | 0.183 | 0.040 | 0.248 | 20 | 22 | 2 |
| deepcogito/cogito-v2-preview-llama-70B | 0.114 | 0.173 | 0.047 | 0.182 | 0.071 | 0.276 | 21 | 21 | 0 |
| zai-org/GLM-4.5-Air-FP8 | 0.104 | 0.067 | 0.039 | 0.170 | -0.034 | 0.168 | 22 | 24 | 2 |
| deepcogito/cogito-v2-preview-llama-109B-MoE | 0.098 | 0.173 | 0.030 | 0.167 | 0.069 | 0.278 | 23 | 20 | -3 |
| scb10x/scb10x-typhoon-2-1-gemma3-12b | 0.040 | 0.121 | -0.023 | 0.104 | 0.019 | 0.223 | 24 | 23 | -1 |
| Qwen/Qwen2.5-72B-Instruct-Turbo | 0.005 | 0.026 | -0.059 | 0.068 | -0.074 | 0.126 | 25 | 25 | 0 |
| mistralai/Mistral-Small-24B-Instruct-2501 | -0.037 | -0.025 | -0.101 | 0.028 | -0.124 | 0.075 | 26 | 26 | 0 |
| meta-llama/Meta-Llama-3.1-405B-Instruct-Turbo | -0.079 | -0.088 | -0.142 | -0.015 | -0.186 | 0.010 | 27 | 27 | 0 |
| meta-llama/Llama-4-Scout-17B-16E-Instruct | -0.098 | -0.121 | -0.164 | -0.031 | -0.223 | -0.020 | 28 | 28 | 0 |
| meta-llama/Llama-4-Maverick-17B-128E-Instruct-FP8 | -0.105 | -0.140 | -0.172 | -0.038 | -0.244 | -0.037 | 29 | 29 | 0 |
| meta-llama/Llama-3.3-70B-Instruct-Turbo | -0.109 | -0.123 | -0.176 | -0.042 | -0.225 | -0.021 | 30 | 30 | 0 |
| meta-llama/Meta-Llama-3-70B-Instruct-Turbo | -0.111 | -0.137 | -0.176 | -0.046 | -0.236 | -0.038 | 31 | 31 | 0 |
| meta-llama/Llama-3-70b-chat-hf | -0.125 | -0.169 | -0.193 | -0.056 | -0.271 | -0.068 | 32 | 32 | 0 |
| meta-llama/Meta-Llama-3.1-70B-Instruct-Turbo | -0.171 | -0.240 | -0.243 | -0.100 | -0.342 | -0.138 | 33 | 33 | 0 |
| Qwen/Qwen2.5-7B-Instruct-Turbo | -0.259 | -0.355 | -0.340 | -0.179 | -0.459 | -0.252 | 34 | 34 | 0 |
| deepseek-ai/DeepSeek-R1-Distill-Llama-70B | -0.307 | -0.452 | -0.393 | -0.220 | -0.557 | -0.347 | 35 | 35 | 0 |
| nvidia/NVIDIA-Nemotron-Nano-9B-v2 | -0.393 | -0.620 | -0.489 | -0.297 | -0.726 | -0.513 | 36 | 38 | 2 |
| Qwen/Qwen2.5-VL-72B-Instruct | -0.411 | -0.565 | -0.510 | -0.313 | -0.670 | -0.461 | 37 | 36 | -1 |
| meta-llama/Meta-Llama-3.1-8B-Instruct-Turbo | -0.468 | -0.589 | -0.574 | -0.362 | -0.693 | -0.486 | 38 | 37 | -1 |
| mistralai/Mixtral-8x7B-Instruct-v0.1 | -0.500 | -0.632 | -0.612 | -0.388 | -0.739 | -0.525 | 39 | 39 | 0 |
| meta-llama/Meta-Llama-3-8B-Instruct-Lite | -0.534 | -0.662 | -0.654 | -0.415 | -0.775 | -0.549 | 40 | 40 | 0 |
| meta-llama/Llama-3.2-3B-Instruct-Turbo | -0.564 | -0.738 | -0.686 | -0.442 | -0.847 | -0.628 | 41 | 41 | 0 |
| mistralai/Mistral-7B-Instruct-v0.3 | -0.587 | -0.837 | -0.712 | -0.461 | -0.947 | -0.728 | 42 | 43 | 1 |
| mistralai/Mistral-7B-Instruct-v0.2 | -0.608 | -0.802 | -0.738 | -0.479 | -0.913 | -0.691 | 43 | 42 | -1 |
| togethercomputer/Refuel-Llm-V2-Small | -0.783 | -1.070 | -0.941 | -0.625 | -1.187 | -0.952 | 44 | 44 | 0 |
| arize-ai/qwen-2-1.5b-instruct | -1.089 | -1.263 | -1.301 | -0.878 | -1.384 | -1.141 | 45 | 45 | 0 |

Figure A.7 presents model parameter count (log-scale) vs. $\hat{\gamma}$. Here we observe that larger models (by parameter count) tend to have larger discrimination in some cases, though the relationship is noisy and not monotonic: certain very large models exhibit modest $\hat{\gamma}$ while some mid-sized models show high $\hat{\gamma}$. This heterogeneity indicates that parameter count alone is an insufficient predictor of judge discrimination; training recipe and alignment/post-processing decisions likely play a major role.

Figure A.8 directly compares $\hat{\gamma}$ estimated on MT-bench against $\hat{\gamma}$ estimated on UltraFeedback for the same judges. The two estimates are strongly positively correlated (points cluster near the diagonal), indicating that judge-specific discrimination is largely stable across these two evaluation datasets. Nevertheless, several judges show dataset-specific adjustments (points deviating from the diagonal), which suggests that dataset composition and prompt types modulate judge sharpness.

A positive correlation between pretraining token count and $\hat{\gamma}$ does not prove that more tokens cause judges to be more discriminative; other confounders (model architecture, pretraining corpus composition, alignment/finetuning procedures, temperature/hyperparameter choices at inference time) can explain the association. Moreover, judge discrimination is relatively stable across MT-bench and UltraFeedback (see Figure A.9), which supports using a single estimated $\hat{\gamma}$ when integrating multiple judge outputs. However, for high-stakes evaluation, one should prefer judges with consistently high and dataset-robust $\hat{\gamma}$.

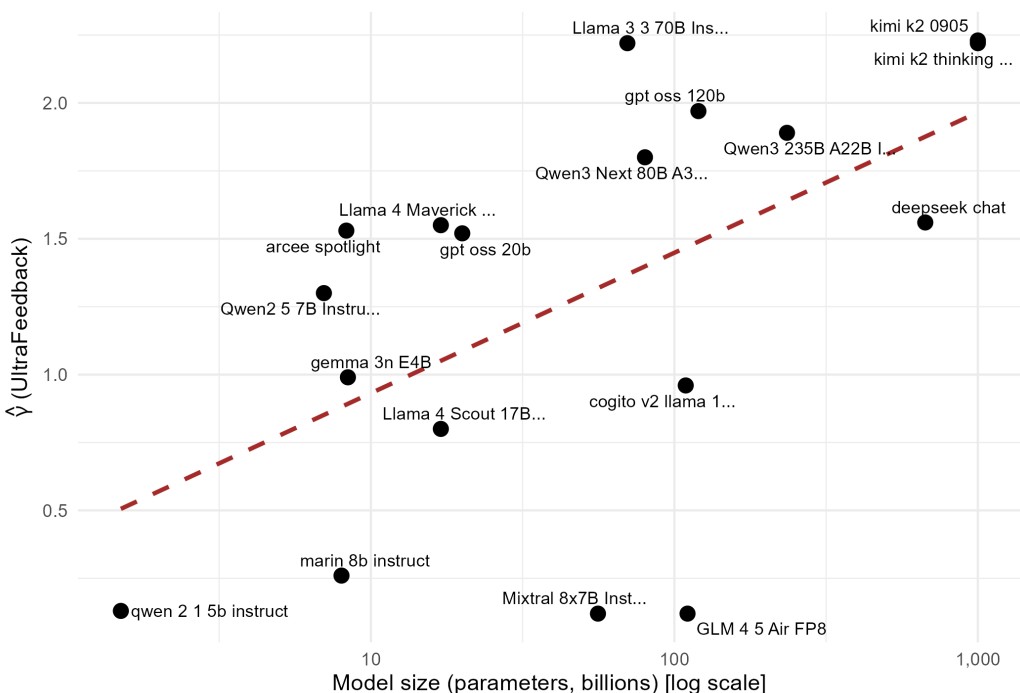

*Figure A.7.* Model size vs judge discrimination (UltraFeedback).

## F.6. Length-Augmented Judge Discrimination

### F.6.1. MODEL

For each comparison $t$, let $q_t$ denote the associated question and define

$$\ell_t = \log\{1 + \text{len}(q_t)\}, \qquad z_t = \frac{\ell_t - \bar{\ell}}{\widehat{\text{sd}}(\ell)},$$

where $\text{len}(q_t)$ is the question length and $z_t$ is the standardized log length.

We parameterize the judge-specific scale as

$$\gamma_k(z_t) = \exp(\alpha_k + \beta_k z_t),$$

where $\alpha_k$ is the baseline log-discrimination of judge $k$ at average question length and $\beta_k$ captures how the discrimination scale changes with question length. The comparison model becomes

$$\mathbb{P}(Y_t = 1 \mid i_t, j_t, k_t, z_t) = \sigma(\exp(\alpha_{k_t} + \beta_{k_t} z_t)(s_{i_t} - s_{j_t})).$$

The scalar judge-aware BTL model is recovered as the special case $\beta_k = 0$ for all judges.

### F.6.2. IDENTIFIABILITY

The identifiable judge-scale object in the length-augmented extension is the function

$$\gamma_k(z) = \exp(\alpha_k + \beta_k z)$$

over the observed covariate support.

**Theorem F.1** (Identifiability up to location-scale transformations). *Consider the length-augmented judge-aware BTL model*

$$\Pr(Y = 1 \mid i, j, k, z) = \sigma(\exp(\alpha_k + \beta_k z)(s_i - s_j)),$$

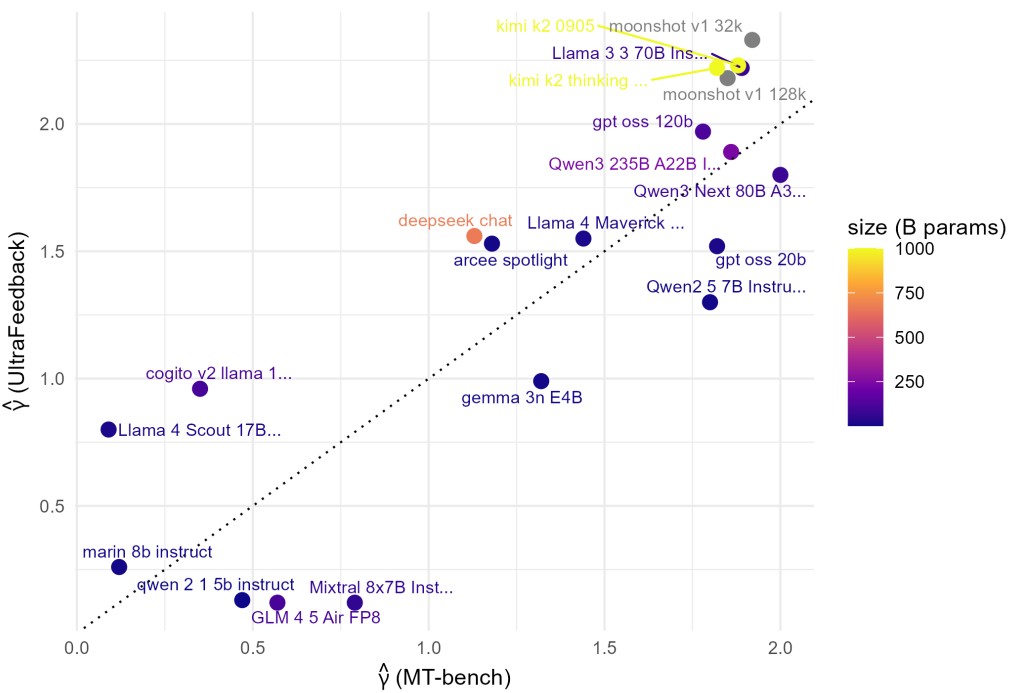

*Figure A.8.* Judge discrimination: MT-bench vs UltraFeedback.

*where $s \in \mathbb{R}^N$, $\alpha, \beta \in \mathbb{R}^K$, and $\sigma(x) = (1 + e^{-x})^{-1}$.*

*Suppose that Assumption 3.1 holds. Suppose also that the covariate support contains at least two distinct values $z_1 \neq z_2$, and that the comparison probabilities are identified for all $i \neq j$, all $k \in [K]$, and all $z \in \{z_1, z_2\}$.*

*If two parameter triples $(s, \alpha, \beta)$ and $(\widetilde{s}, \widetilde{\alpha}, \widetilde{\beta})$ induce the same comparison probabilities, namely*

$$\sigma(\exp(\alpha_k + \beta_k z)(s_i - s_j)) = \sigma\Big(\exp(\widetilde{\alpha}_k + \widetilde{\beta}_k z)(\widetilde{s}_i - \widetilde{s}_j)\Big)$$

*for all $i \neq j$, all $k \in [K]$, and all $z \in \{z_1, z_2\}$, then there exist constants $a > 0$ and $b \in \mathbb{R}$ such that*

$$\widetilde{s} = as + b\mathbf{1}, \qquad \widetilde{\alpha}_k = \alpha_k - \log a, \qquad \widetilde{\beta}_k = \beta_k, \quad k = 1, \ldots, K.$$

*Conversely, any transformation of this form leaves all comparison probabilities unchanged.*

*Proof.* Since the logistic link $\sigma$ is strictly increasing, equality of comparison probabilities implies equality of the linear predictors:

$$\exp(\alpha_k + \beta_k z)(s_i - s_j) = \exp(\widetilde{\alpha}_k + \widetilde{\beta}_k z)(\widetilde{s}_i - \widetilde{s}_j)$$

for all $i \neq j$, all $k \in [K]$, and all $z \in \{z_1, z_2\}$. Hence, for each fixed pair $(k, z)$,

$$\widetilde{s}_i - \widetilde{s}_j = c_{k,z}(s_i - s_j), \qquad c_{k,z} = \exp\Big(\alpha_k + \beta_k z - \widetilde{\alpha}_k - \widetilde{\beta}_k z\Big) > 0.$$

Therefore, for each fixed $(k, z)$, there exists a constant $b_{k,z} \in \mathbb{R}$ such that

$$\widetilde{s}_i = c_{k,z} s_i + b_{k,z}, \qquad i = 1, \ldots, N.$$

Because the same vector $\widetilde{s}$ must satisfy this affine representation for every $(k, z)$, the slope $c_{k,z}$ cannot depend on $(k, z)$. Indeed, for two pairs $(k, z)$ and $(\ell, z')$,

$$(c_{k,z} - c_{\ell,z'})s_i + (b_{k,z} - b_{\ell,z'}) = 0, \qquad i = 1, \ldots, N.$$

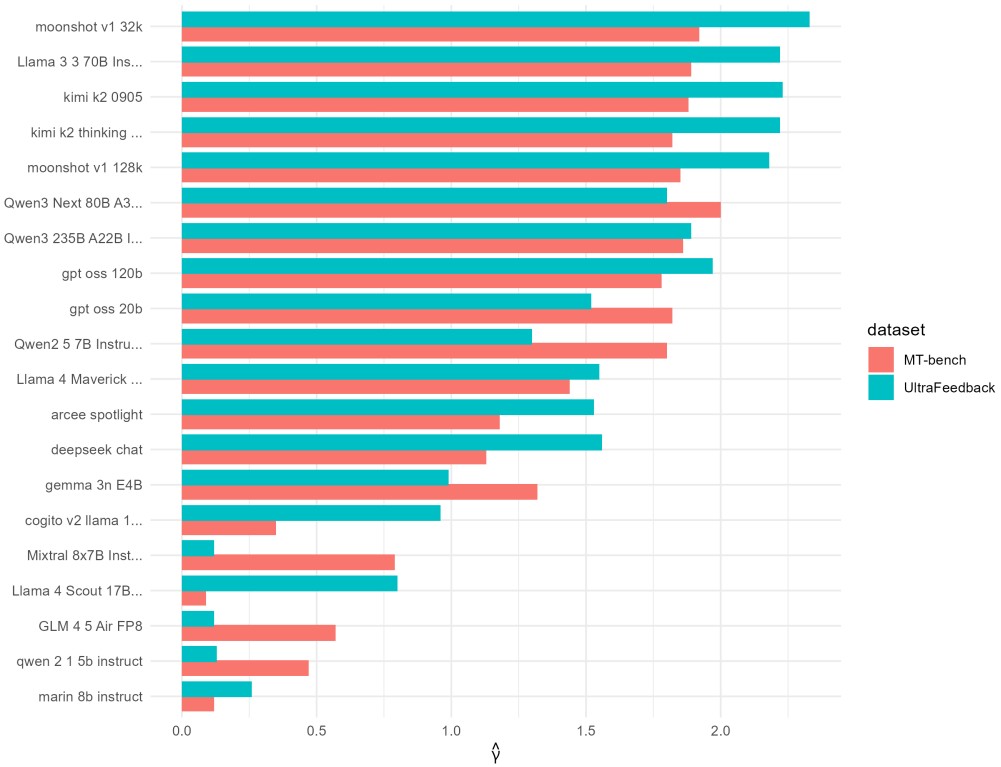

*Figure A.9.* Judge discrimination scales by model and dataset.

Taking two indices $i_1, i_2$ with $s_{i_1} \neq s_{i_2}$ gives

$$(c_{k,z} - c_{\ell,z'})(s_{i_1} - s_{i_2}) = 0,$$

and hence $c_{k,z} = c_{\ell,z'}$. Thus there exist common constants $a > 0$ and $b \in \mathbb{R}$ such that

$$\widetilde{s} = as + b\mathbf{1}.$$

Substituting this relation back into the equality of linear predictors gives, for all $k$ and $z \in \{z_1, z_2\}$,

$$\exp(\alpha_k + \beta_k z) = a \exp(\widetilde{\alpha}_k + \widetilde{\beta}_k z).$$

Taking logarithms yields

$$\alpha_k + \beta_k z = \log a + \widetilde{\alpha}_k + \widetilde{\beta}_k z.$$

Evaluating this identity at $z_1$ and $z_2$ and subtracting gives

$$(\beta_k - \widetilde{\beta}_k)(z_1 - z_2) = 0.$$

Since $z_1 \neq z_2$, we obtain

$$\widetilde{\beta}_k = \beta_k.$$

Substituting this back gives

$$\widetilde{\alpha}_k = \alpha_k - \log a.$$

This proves the necessity.

Conversely, if

$$\widetilde{s} = as + b\mathbf{1}, \qquad \widetilde{\alpha}_k = \alpha_k - \log a, \qquad \widetilde{\beta}_k = \beta_k,$$

Table A.8. Estimated judge-specific discrimination scales ($\widehat{\gamma}$).

| Model | Size | Tokens | MT-bench | UltraFeedback |
|---|---|---|---|---|
| kimi-k2-0905 | 1T | – | 1.88 | 2.23 |
| kimi-k2-thinking-turbo | 1T | – | 1.82 | 2.22 |
| deepseek-chat | 671B | 14.8T | 1.13 | 1.56 |
| Qwen3-235B-A22B-Instruct-2507-tput | 235B | 15T | 1.86 | 1.89 |
| gpt-oss-120b | 120B | – | 1.78 | 1.97 |
| GLM-4.5-Air-FP8 | 110.5B | – | 0.57 | 0.12 |
| cogito-v2-llama-109B-MoE | 109B | – | 0.35 | 0.96 |
| Qwen3-Next-80B-A3B-Instruct | 80B | – | 2.00 | 1.80 |
| Llama-3.3-70B-Instruct-Turbo | 70B | 15T | 1.89 | 2.22 |
| Mixtral-8x7B-Instruct-v0.1 | 56B | – | 0.79 | 0.12 |
| gpt-oss-20b | 20B | – | 1.82 | 1.52 |
| Llama-4-Maverick-17B-128E-Instruct-FP8 | 17B | – | 1.44 | 1.55 |
| Llama-4-Scout-17B-16E-Instruct | 17B | – | 0.09 | 0.80 |
| gemma-3n-E4B | 8.4B | – | 1.32 | 0.99 |
| arcee-spotlight | 8.3B | – | 1.18 | 1.53 |
| marin-8b-instruct | 8B | 13.7T | 0.12 | 0.26 |
| Qwen2.5-7B-Instruct-Turbo | 7B | 18T | 1.80 | 1.30 |
| qwen-2-1.5b-instruct | 1.5B | – | 0.47 | 0.13 |
| moonshot-v1-32k | – | – | 1.92 | 2.33 |
| moonshot-v1-128k | – | – | 1.85 | 2.18 |

then

$$\exp(\widetilde{\alpha}_k + \widetilde{\beta}_k z)(\widetilde{s}_i - \widetilde{s}_j) = \exp(\alpha_k - \log a + \beta_k z)a(s_i - s_j) = \exp(\alpha_k + \beta_k z)(s_i - s_j).$$

Therefore all comparison probabilities are unchanged. $\square$

**Corollary F.2** (Uniqueness under normalization). *Under the conditions of the theorem, impose the normalization constraints*

$$\sum_{i=1}^{N} s_i = 0, \qquad \sum_{k=1}^{K} \alpha_k = 0.$$

*Then the length-augmented model is identifiable. That is, if two normalized parameter triples induce the same comparison probabilities, then*

$$\widetilde{s} = s, \qquad \widetilde{\alpha} = \alpha, \qquad \widetilde{\beta} = \beta.$$

*Proof.* By Theorem F.1, there exist $a > 0$ and $b \in \mathbb{R}$ such that

$$\widetilde{s} = as + b\mathbf{1}, \qquad \widetilde{\alpha}_k = \alpha_k - \log a, \qquad \widetilde{\beta}_k = \beta_k.$$

Using $\sum_i s_i = \sum_i \widetilde{s}_i = 0$, we have

$$0 = \sum_{i=1}^{N} \widetilde{s}_i = a\sum_{i=1}^{N} s_i + Nb = Nb,$$

so $b = 0$. Using $\sum_k \alpha_k = \sum_k \widetilde{\alpha}_k = 0$, we have

$$0 = \sum_{k=1}^{K} \widetilde{\alpha}_k = \sum_{k=1}^{K} \alpha_k - K \log a = -K \log a.$$

Thus $a = 1$. Therefore

$$\widetilde{s} = s, \qquad \widetilde{\alpha} = \alpha, \qquad \widetilde{\beta} = \beta.$$

$\square$

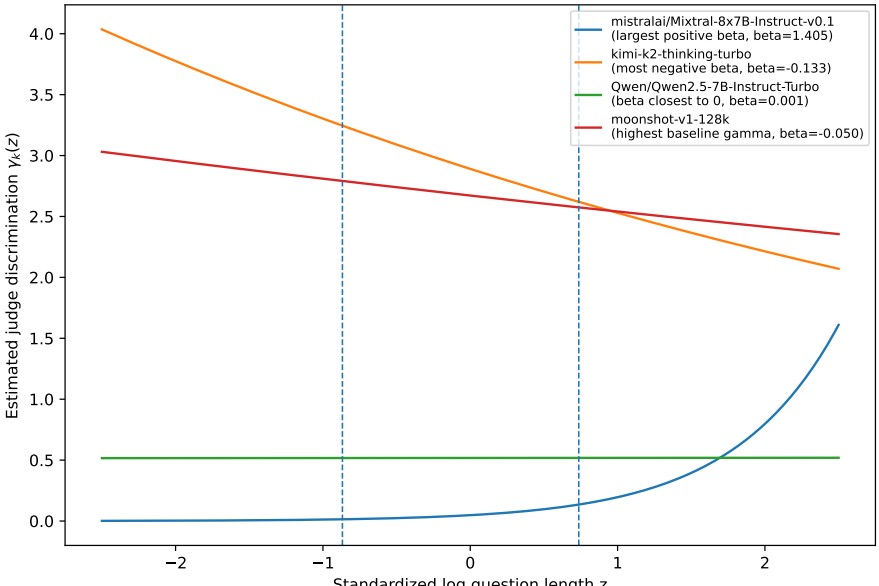

*Figure A.10.* Representative estimated judge-discrimination curves $\gamma_k(z) = \exp(\alpha_k + \beta_k z)$ as a function of standardized log question length $z$. Dashed vertical lines indicate the 25th and 75th percentiles of $z$. Judges vary substantially in both baseline discrimination and sensitivity to question length.

### F.6.3. ALGORITHM

For the length-augmented model, we use the same projected Adam scheme as Algorithm A.1, but replace the linear predictor by

$$\eta_t = \exp(\alpha_{k_t} + \beta_{k_t} z_t)(s_{i_t} - s_{j_t}).$$

The gradients are accumulated over row-level observations because $z_t$ may vary across comparisons. The projection step is applied only to $s$ and $\alpha$, while $\beta$ is updated without centering.

For a single observation, define

$$\ell_t = y_t \log \sigma(\eta_t) + (1 - y_t) \log\{1 - \sigma(\eta_t)\}.$$

The additional gradient contribution for $\beta_k$ is

$$\frac{\partial \ell_t}{\partial \beta_k} = \mathbf{1}\{k_t = k\}(y_t - \sigma(\eta_t))\eta_t z_t.$$

### F.6.4. EMPIRICAL RESULTS

We fit the length-augmented model on the in-house dataset. Figure A.10 shows the estimated discrimination curves $\gamma_k(z)$ for four representative judges. Judges differ substantially in both baseline discrimination and sensitivity to question length: some judges have larger estimated discrimination on longer questions ($\widehat{\beta}_k > 0$), some have smaller estimated discrimination ($\widehat{\beta}_k < 0$), and some remain largely unaffected ($\widehat{\beta}_k \approx 0$).

The resulting model rankings are nearly identical to those from the scalar judge-aware model: the Spearman rank correlation across the 45 candidate models is 0.9996, and the maximum absolute rank change is only one position. This confirms that the scalar model's main leaderboard conclusions are robust, while illustrating that question length can be incorporated as one natural observable covariate in the proposed framework.

These estimates should be interpreted as a proof of concept rather than causal evidence: in this design, question length is tied to the specific question and model-pair instance, and may therefore reflect question difficulty or pair-specific effects.

*Table A.9.* Pearson and Spearman correlations between rankings from sparse and full designs. The retained edge fraction $\rho_{\text{edge}}$ is the fraction of model-pair edges kept in the connected sparse graph.

| $\rho_{\text{edge}}$ | Pearson | Spearman |
| --- | --- | --- |
| 0.044 | 0.613 | 0.613 |
| 0.089 | 0.835 | 0.817 |
| 0.133 | 0.877 | 0.858 |
| 0.178 | 0.896 | 0.872 |
| 0.222 | 0.887 | 0.815 |
| 0.267 | 0.927 | 0.909 |
| 0.311 | 0.916 | 0.906 |
| 0.356 | 0.868 | 0.816 |
| 0.400 | 0.911 | 0.897 |
| 0.444 | 0.884 | 0.887 |

## F.7. Robustness to Sparse Comparison Graphs

The main framework assumes a well-covered comparison graph. To assess robustness to sparser designs, we fix the total number of comparisons and vary the fraction of retained model-pair edges. Let $E_{\text{full}}$ denote the complete set of model-pair edges and $E_{\text{sp}} \subseteq E_{\text{full}}$ denote the retained connected edge set. We define the retained edge fraction as

$$\rho_{\text{edge}} = \frac{|E_{\text{sp}}|}{|E_{\text{full}}|}.$$

For each value of $\rho_{\text{edge}}$, we retain a random connected subgraph, evenly redistribute the fixed comparison budget across the retained edges, refit the judge-aware model, and compare the resulting ranking against that from the full design using Pearson and Spearman correlations. Table A.9 reports the results. Correlation is lowest in the spanning-tree regime ($\rho_{\text{edge}} = 0.044$), where both correlations are 0.613. As the graph becomes modestly denser, both correlations rise quickly above 0.8 and remain high over a broad range of connected sparse designs.

These results suggest that, although extremely sparse connected designs can reduce stability, the method remains fairly stable over a broad range of sparse but connected comparison graphs that depart substantially from the idealized setting.

