# OpenReview forum: "A Judge-Aware Ranking Framework for Evaluating Large Language Models without Ground Truth"
_ICML.cc/2026/Conference — ICML 2026 regular_

### Official Review · Reviewer_7mCp · 2026-03-07

**Soundness:** 3
**Presentation:** 3
**Significance:** 4
**Originality:** 4
**Overall Recommendation:** 4
**Confidence:** 2

**Summary:**

This paper proposes an improved LLM-as-a-judge evaluation framework. It extends the traditional Bradley-Terry-Luce (BTL) model by introducing a discrimination scale parameter to measure the reliability of responses from different Judge LLMs.  The results show that this judge-aware framework, which models the reliability of the evaluator, solves the different LLMs' heterogeneity problem in LLM-as-a-judge evaluation.

**Compliance With Llm Reviewing Policy:**

Affirmed.

**Key Questions For Authors:**

1. In realistic benchmark, new models may be added in later stages rather than being jointly evaluated from the start. This may lead to a comparison graph that is technically connected but only through a small number of bridge comparisons between older and newly added models. Have authors quantified how ranking stability degrades under such barely connected evaluation designs? Or let's say is there a phase transition where the estimate of $\gamma$ will suddenly change or become invalid when the judge overlap review falls below a certain proportion?
2. Given a fixed budget, should priority be given to increasing the number of judges, increasing overlapping reviews, or increasing the number of comparisons per edge?

**Limitations:**

yes

**Strengths And Weaknesses:**

Strengths
1. Existing LLM-as-a-judge paradigms typically assume that all judges are equally reliable. This paper points out that significant reliability differences exist among judges and quantifies these differences, thus avoiding the problem of "mediocre judges" diluting high-quality evaluation results.
2. This framework is based entirely on pairwise comparisons between models to jointly estimate model quality and judge reliability, making it well-suited for handling open-ended tasks without standard answers.



Weaknesses
1. Since there is only one global parameter for each judge, assuming its judging ability is constant across all domains. But, different LLMs will have different performance across different domains. The current setup cannot fully characterize this heterogeneity and may affects the reliability of the judge's result.
2. This method is dependent on the connectivity of the comparison graph and the overlap of judge coverage. Although connectivity was checked or guaranteed in the experimental settings, the stability of the estimates may be limited in cases of extreme sparsity or when specific judges only evaluate a subset of models.

---

> ### Author Rebuttal · Authors · 2026-03-31
>
> We thank the reviewer for the thoughtful comments and for recognizing the value of explicitly modeling judge heterogeneity in LLM-as-a-judge evaluation. We respond below to the main weaknesses and questions raised in the review.\
> First, we agree that a single global scalar $\gamma_k$ cannot fully capture domain-specific or task-specific judge quality. In the current paper, we adopt $\sigma(\gamma_k(s_i-s_j))$ as a simple first-order model to isolate the main effect of judge heterogeneity while maintaining strict model identifiability in our present setting. A natural extension is to let judge discrimination depend on task/domain features $\mathbf{x}$, for example
> $$
> \mathbb{P}\left(Y_{k,ij}=1 \mid \mathbf{x}\right)=\sigma\left(\gamma_k(\mathbf{x})(s_i-s_j)\right),
> \qquad
> \gamma_k(\mathbf{x})=\exp\bigl(g_k(\mathbf{x})\bigr).
> $$
> A practical choice is $g_k(\mathbf{x})=\alpha_k+\beta_k^\top h(\mathbf{x})$, where $h(\mathbf{x})$ represents task-, domain-, or capability-related information such as category, length, or other summary features like embeddings, $\alpha_k$ captures the global reliability of judge $k$, and $\beta_k$ captures how judge $k$'s discrimination varies with this representation. More generally, $g_k(\cdot)$ itself can be parameterized by a more expressive model, such as a Transformer encoder.\
> We view this as a meaningful extension of the present framework rather than a contradiction. We did not include this richer parameterization in the current submission because our goal here is to first isolate and study the primary effect of judge heterogeneity under a simpler formulation. We agree that the feature-dependent extension is natural, and we will clarify this point in the revision and add an empirical study to illustrate its feasibility.\
> Second, we agree that sparse, weakly connected, and low-overlap comparison designs are important practical concerns. Here, we want to clarify that in the current paper, the main experimental setting is based on a uniform i.i.d. random design, and the theoretical presentation is stated in that form for simplicity. More generally, the same argument extends to i.i.d. sampling from a fixed design distribution $\pi_{i j k}$, provided the relevant sampling probabilities are positive so that the necessary comparisons remain identifiable and sufficiently covered.\
> For sparse comparison graphs, as discussed in Section 2.1, related BTL-type models have studied uncertainty quantification and structured estimation under sparse designs, whereas our paper isolates the orthogonal issue of judge reliability heterogeneity. To partially address this concern, we conducted a sparsity robustness experiment under connected comparison graphs. Due to the rebuttal length limit, we provide the full setup and numerical results in our response to Reviewer L95p, Q3. \
> In summary, this experiment suggests that, although extremely sparsity can noticeably reduce stability, the method remains reasonably robust over a broad range of sparse but connected comparison designs that already depart from the idealized setting. It also partially reflects the practical situation in which coverage is limited and some judges only observe a subset of models, since changing the comparison graph reduces overlap and makes estimation rely more heavily on local comparisons. At the same time, we do not currently provide a sharp characterization of a connectivity or judge-overlap threshold below which the estimator becomes unstable, and we view this as an important direction for future work.\
> Finally, regarding the fixed-budget design question, we agree that our current paper primarily focuses on the aggregation problem rather than fully solving the broader allocation and experimental design problem. While a full theoretical treatment is beyond the scope of the current submission, we will add additional empirical evidence in the revision to provide preliminary guidance on this budget-allocation issue.

---

> > ### Author Rebuttal · Reviewer_7mCp · 2026-04-03
> >
> > Thank the authors for the detailed and thoughtful rebuttal, I will maintain my original score.

---

> > > ### Author Response · Authors · 2026-04-08
> > >
> > > Thank you for the positive follow-up and for confirming that our rebuttal addressed your concerns.
> > >
> > > In the final version, we will also make the fixed-budget design discussion more explicit. While our paper focuses on aggregation rather than fully solving the broader allocation or experimental design problem, the current framework can still offer some practical guidance: uncertainty estimates can help assess whether additional judgments are needed, and the estimated judge-specific reliability can help inform cost-aware judge selection in mixed-quality settings. For example, when two judges have comparable estimated reliability but substantially different costs, this can help support more cost-aware choices in practice.

---

### Official Review · Reviewer_4t2d · 2026-03-12

**Soundness:** 2
**Presentation:** 3
**Significance:** 2
**Originality:** 3
**Overall Recommendation:** 4
**Confidence:** 4

**Summary:**

This work proposes a ranking framework based on large model reliability awareness. By extending the Bradley-Terry-Luce model, it jointly estimates candidate model quality and judge discrimination parameters in unlabeled environments. It achieves good performance on relevant datasets.

**Compliance With Llm Reviewing Policy:**

Affirmed.

**Final Justification:**

Thank you for the authors' detailed response. Your rebuttal clearly answered several key questions I raised during the initial review.

First, the clarification regarding the computational cost of likelihood estimation. The explanation that computational complexity increases linearly (rather than exponentially) with the number of observations is very clear. I strongly recommend including this explanation of complexity in the final version, as it is crucial for readers to understand the practicality of the method.

Second, I carefully read your discussions regarding graph sparsity (and the supplementary experiments provided to other reviewers), the limitations of a single reliability scalar, and the adversarial scoring/cold start problem. While the inability to handle multidimensional capability differences and dynamic online updates remains a practical limitation of the current framework, your proposed extensions are theoretically reasonable and feasible.

Overall, the author’s response clarified some of my technical misunderstandings. Based on the paper’s contributions and objective limitations, particularly in cold start scenarios and multidimensional capability assessment, which still limit its broader impact, I have decided to maintain the current rating. I look forward to seeing your promised supplementary explanations and discussions in the final version.

**Key Questions For Authors:**

Can this offline parameter estimation framework be modified into a lightweight module to provide online reward signals for model iteration in continuous learning scenarios?

In extremely sparse comparison graphs, how can a more robust regularization strategy be designed to ensure the stable convergence of the judge's discrimination parameter?

If this ranking framework is used for the optimal decision-making action of embodied agents, how can continuous physical world feedback be integrated to replace discrete binary win-loss relationships?

**Limitations:**

The framework relies on a fully hybrid cross-evaluation network, which presents a severe cold-start dilemma when evaluating objects with extremely narrow domains or unique architectures (such as novel robot strategies).

**Strengths And Weaknesses:**

**Strengths**

- By incorporating judge differences into the likelihood function for automatic weight adjustment, it provides a theoretical explanation and statistical inference basis for multi-model peer review.

- A solid statistical foundation is built, deriving the asymptotic normality and confidence interval of maximum likelihood estimation, and providing a reliable uncertainty quantification mechanism.

- It can effectively identify and de-weight low-quality judges, significantly improving consistency with human preferences and data utilization efficiency in small-sample scenarios.


**Weaknesses**

- However, the computational cost of maximum likelihood estimation increases exponentially with the number of comparison pairs, severely limiting scalability in large-scale concurrent multi-agent interaction scenarios.

- While it assumes a uniformly sampled graph structure, real-world system interaction data is often highly sparse and skewed, which easily leads to parameter estimation divergence.

- Defining reliability solely as a single scalar (discrimination) fails to capture the dimensional differences in agents' abilities across specific sub-domains (such as visual feature recognition and physics-based reasoning).

- The framework does not consider malicious collusion or adversarial scoring; if a large number of homogeneous models exist in the evaluation group, the system is highly susceptible to ranking collapse.

---

> ### Author Rebuttal · Authors · 2026-03-31
>
> We thank the reviewer for the thoughtful comments and for recognizing the statistical foundation and uncertainty quantification of modeling judge heterogeneity. We address the main concerns below.\
> First, regarding scalability, we may not have explained the computational cost clearly enough. Let $T$ be the total number of observed comparisons. Since the log-likelihood is a sum over observations, each likelihood or gradient evaluation costs $O(T)$. After aggregating repeated observations over the same $(i, j, k)$ triple, the cost becomes $O(|\Omega|)$, where $|\Omega| \leq K\binom{N}{2}=O\left(K N^2\right)$. Thus, computation scales linearly in the number of observed comparisons (or unique observed triples after aggregation), rather than exponentially. We will clarify this in the revision.\
> Second, we agree that sparse and skewed comparison graphs are important practical concerns. In the current paper, the main experimental setting assumes a uniform i.i.d. random design, and the theory is stated in that form for simplicity. More generally, the same argument extends to i.i.d. sampling from any fixed design distribution $\{\pi_{ijk}\}$, provided the relevant sampling probabilities are positive so that the necessary comparisons remain identifiable and sufficiently covered.\
> For sparse comparison graphs, as discussed in Section~2.1, related BTL-type models have studied uncertainty quantification and structured estimation under sparse designs, whereas our paper isolates the orthogonal issue of judge reliability heterogeneity. To partially address this concern, we conducted a sparsity robustness experiment under connected comparison graphs. In short, while extremely sparse connected designs can reduce stability, the method remains reasonably stable across a broad range of sparse but connected graphs. We provide the full setup and numerical results in our response to Reviewer L95p, Q3.\
> Third, we agree that a single global scalar $\gamma_k$ cannot fully capture domain-dependent or task-specific judge quality. In the current submission, we use it as a parsimonious model to isolate the main effect of judge heterogeneity while maintaining identifiability. A natural extension is to allow judge discrimination to depend on domain or task features. We do not discuss the details here and instead refer to the second paragraph of our response to Reviewer 7mCp for the concrete parameterization and discussion. We will clarify this in the revision and add an empirical study to illustrate its feasibility.\
> In addition, in the LLM-as-a-judge setting we consider, the more relevant issue is usually variation in discrimination ability across judges rather than adversarial label flipping. We therefore do not explicitly model adversarial scoring in the current version. As noted in Section 3.1, the likelihood is formally well-defined for signed $\gamma_k$, although the current submission restricts $\gamma_k > 0$ so that $\gamma_k$ has a clear interpretation as judge discrimination and the logarithmic normalization remains well-defined. A natural extension is to allow $\gamma_k < 0$ for judges that are systematically anti-aligned with the latent ranking. This introduces a global sign ambiguity, so an orientation constraint is needed, for example by anchoring at least one trusted judge to have positive $\gamma_k$. Under such a convention, positive $\gamma_k$ indicates alignment with the latent ranking, while negative $\gamma_k$ indicates systematic adversarial or anti-aligned behavior.\
> At the same time, the current likelihood-based formulation relies on conditional independence given the latent scores and judge-specific parameters. It therefore does not explicitly model homogeneous judge groups with shared biases, strongly correlated judge errors, or collusion-like effects from repeated or highly similar judgments. A practical mitigation is to encourage judge diversity and avoid over-representing the same model family.\
> Finally, we agree that the current framework is developed for offline pairwise ranking under a well-covered comparison graph. It does not explicitly address cold-start settings with limited cross-evaluation coverage or online reward generation during iterative model updates. For newly added models, a natural extension is to keep the previously estimated model and judge parameters fixed and estimate only the new model’s score from bridge comparisons with existing models. More broadly, incremental or periodically updated variants are possible, and we leave a systematic treatment of online and cold-start settings to future work. For continuous-feedback settings, the framework could in principle be extended by replacing the current Bernoulli/logistic observation model with a suitable continuous-outcome model. The restriction to binary labels is thus a modeling choice rather than a fundamental limitation. More broadly, the principle of judge-aware modeling can be extended to other observation models and feedback types.

---

> > ### Author Rebuttal · Reviewer_4t2d · 2026-04-03
> >
> > Thank you for the authors' detailed response. Your rebuttal clearly answered several key questions I raised during the initial review.
> >
> > First, the clarification regarding the computational cost of likelihood estimation. The explanation that computational complexity increases linearly (rather than exponentially) with the number of observations is very clear. I strongly recommend including this explanation of complexity in the final version, as it is crucial for readers to understand the practicality of the method.
> >
> > Second, I carefully read your discussions regarding graph sparsity (and the supplementary experiments provided to other reviewers), the limitations of a single reliability scalar, and the adversarial scoring/cold start problem. While the inability to handle multidimensional capability differences and dynamic online updates remains a practical limitation of the current framework, your proposed extensions are theoretically reasonable and feasible.
> >
> > Overall, the author’s response clarified some of my technical misunderstandings. Based on the paper’s contributions and objective limitations, particularly in cold start scenarios and multidimensional capability assessment, which still limit its broader impact, I have decided to maintain the current rating. I look forward to seeing your promised supplementary explanations and discussions in the final version.

---

> > > ### Author Response · Authors · 2026-04-08
> > >
> > > Thank you for the positive feedback on our rebuttal and your thoughtful comments. We are glad that our rebuttal helped clarify several questions and addressed your concerns.
> > >
> > > We especially appreciate your suggestion to clarify the computational complexity more explicitly. We agree that this point is important for understanding the practicality of the method, and we will include a clearer explanation of the likelihood-estimation complexity in the final version.
> > >
> > > We also value your careful reading of our discussions on graph sparsity, the limitation of using a single reliability scalar, the adversarial scoring and cold-start setting. We agree that these are important practical issues for the current framework, including both current limitations and natural directions for extension. In the final version, we will make these points more explicit.

---

### Official Review · Reviewer_L95p · 2026-03-13

**Soundness:** 3
**Presentation:** 3
**Significance:** 3
**Originality:** 3
**Overall Recommendation:** 4
**Confidence:** 1

**Summary:**

This paper studies how to rank LLMs from pairwise judgments when the judge models themselves differ in reliability. The main idea is to extend the Bradley–Terry–Luce framework with judge-specific discrimination parameters, so that the method jointly estimates model quality and judge reliability from pairwise comparisons without requiring ground-truth labels. Empirically, the method is evaluated through simulations and on several LLM evaluation datasets, where the authors report improved alignment with human preferences and better sample efficiency than an unweighted baseline. An important concept assessed by the study is whether modeling judge heterogeneity can improve both ranking quality and uncertainty estimation in LLM-as-a-judge settings.

**Compliance With Llm Reviewing Policy:**

Affirmed.

**Key Questions For Authors:**

The empirical comparisons mainly emphasize the unweighted BTL baseline. Could the authors include more direct controlled comparisons with other judge-aware or reliability-aware ranking methods discussed in related work?

How sensitive are the results to the treatment of ties as 0.5? Since ties are common in practice, it would be helpful to know whether alternative tie-handling strategies materially change the rankings or uncertainty estimates.

The theory assumes an i.i.d. random design over judge-model pairs. How robust is the method when the comparison design is not close to uniform, which seems likely in real evaluation pipelines?

**Limitations:**

The paper does include an impact statement and acknowledges that automated evaluation may inherit or amplify judge-model biases, and it recommends using diverse judge panels and interpreting rankings carefully.

**Strengths And Weaknesses:**

I should note that this topic is outside my main area of expertise, especially the theoretical side of statistical ranking models. My assessment is therefore based more on the clarity of the motivation, whether the method is reasonably justified, and whether the empirical evidence appears consistent with the claims. Since this paper is theory-heavy, it is difficult for a non-expert reader to judge how strong the theoretical novelty really is relative to prior work in statistical ranking, crowd aggregation, and IRT-style models.


**Strengths**

The paper addresses an important and timely problem in LLM evaluation.

The paper does more than propose a heuristic. It provides identifiability and asymptotic results, which is a meaningful strength for a paper of this kind.

The experiments include both simulations and real datasets.
The empirical results are generally aligned with the main claim that modeling judge heterogeneity can improve ranking quality and statistical efficiency, especially in lower-budget settings.

**Weaknesses**


The empirical comparisons are mostly against the unweighted BTL baseline. While this is the most direct baseline, I would have liked to see stronger comparisons to other judge-aware or reliability-aware aggregation methods discussed in related work, such as other probabilistic aggregation approaches, under a more controlled setup. As written, the empirical section mainly demonstrates that the proposed model is better than ignoring judge heterogeneity, which is useful but somewhat limited.


Overall, I think the paper is interesting and reasonably well executed. The problem is important, the core modeling idea is sensible, and the combination of theory plus experiments makes the paper stronger.

---

> ### Author Rebuttal · Authors · 2026-03-31
>
> Q1:We thank the reviewer for this suggestion. We agree that broader controlled comparisons with other judge-aware aggregation methods would strengthen the empirical section. We use unweighted BTL as the primary baseline because it gives the cleanest reference for the paper’s central question: whether explicitly modeling judge heterogeneity improves ranking and uncertainty estimation relative to ignoring it. Among related methods, Crowd-BT (Section 2.2) and FTR (Section 2.3) are especially relevant, and direct comparisons would be valuable. Where feasible, we will add them in the revision and clarify how their assumptions differ from ours.\
> Q2:We thank the reviewer for raising this practical point. Since ties are common in LLM evaluation, we should check whether our conclusions are sensitive to tie handling. We therefore conducted a robustness check that removes tied comparisons entirely rather than mapping them to 0.5. Across the four real datasets, the resulting rankings remain highly consistent with the main results: they are unchanged on MT-bench (Table 1), Chatbot Arena (Table 2), and UltraFeedback (Table A.6). On the in-house dataset (Table A.7), only two local swaps occur: Qwen/Qwen3-Next-80B-A3B-Thinking and google/gemma-3n-E4B-it exchange positions, and deepcogito/cogito-v2-preview-llama-70B and deepcogito/cogito-v2-preview-llama-405B exchange positions. For uncertainty estimation on the in-house dataset, the average confidence interval width increases only modestly after removing ties, from 0.185 to 0.195, as expected because fewer effective comparisons remain. Overall, these results suggest that our findings are not highly sensitive to the tie-handling choice. We will add this robustness check to the revision.\
> Ties can also be modeled explicitly through a tie-aware extension of the BTL model rather than being mapped to 0.5 or removed. For example, Caron and Doucet (2012) introduce a threshold parameter theta > 1 and model
> $$
> \mathbb{P}(i \text{ beats } j)=\frac{\lambda_i}{\lambda_i+\theta\lambda_j},\qquad
> \mathbb{P}(i \text{ ties } j)=\frac{(\theta^2-1)\lambda_i\lambda_j}{(\lambda_i+\theta\lambda_j)(\theta\lambda_i+\lambda_j)},
> $$
> with an analogous expression for $\mathbb{P}(j \text{ beats } i)$. Letting $\lambda_i=e^{s_i}$ and $\lambda_j=e^{s_j}$ gives
> $$
> \mathbb{P}(i \text{ beats } j)=\sigma\left(s_i-s_j-\log\theta\right),
> $$
> so $i$ must exceed $j$ by at least $\log\theta$ to be more likely to win than tie. This provides a principled approach to explicit tie modeling, and our framework can be naturally extended in this direction. We view this as an important direction for future work.\
> Q3:We thank the reviewer for this important question. The theory is stated under a uniform i.i.d. random design mainly for simplicity of exposition. The core likelihood-based asymptotic argument does not rely on the exact choice $\pi_{ijk}=\frac{2}{K N (N-1)}$. More generally, the same argument extends to i.i.d. sampling from a fixed design distribution $\{\pi_{ijk}\}$, provided the relevant sampling probabilities are positive so that the needed comparisons remain identifiable and sufficiently covered. Thus, exact uniformity is a convenient special case rather than the essential requirement, and we will clarify this in the revision.
>
> We also agree that real evaluation pipelines may deviate from this idealized assumption and may exhibit sparser or more imbalanced comparison patterns. To partially address this concern, we conducted an additional sparsity robustness experiment under connected comparison graphs. We fix the total number of comparisons and vary the edge density of the connected graph, where sparsity = (number of edges used)/(maximum number of edges). For each sparse design, we evenly redistribute the fixed comparison budget across the retained edges and re-estimate the model, then compare the resulting rankings against the full model using Pearson and Spearman correlations. The correlation is lowest in the spanning-tree regime (above 0.61). Since a spanning tree is the sparsest possible connected graph, this level of agreement is already notable. Both correlations then rise quickly above 0.8 once the graph becomes modestly denser, and remain around 0.85–0.92 over a broad range of connected sparse designs. See the table below for details:
>
> |sparsity|pearson|spearman|
> |-|-|-|
> |0.044|0.613|0.613|
> |0.089|0.835|0.817|
> |0.133|0.877|0.858|
> |0.178|0.896|0.872|
> |0.222|0.887|0.815|
> |0.267|0.927|0.909|
> |0.311|0.916|0.906|
> |0.356|0.868|0.816|
> |0.400|0.911|0.897|
> |0.444|0.884|0.887|
>
> These results suggest that, although extremely sparse connected designs can reduce stability, the method remains fairly stable over a broad range of sparse but connected comparison graphs that depart substantially from the idealized setting. A systematic treatment of more general non-uniform comparison designs remains beyond the scope of the current submission and is an important direction for future work.

---

> > ### Author Rebuttal · Reviewer_L95p · 2026-04-03
> >
> > Thank you for the detailed and thoughtful rebuttal. I really appreciate the additional clarifications and robustness analyses. My questions were fully addressed.

---

> > > ### Author Response · Authors · 2026-04-08
> > >
> > > Thank you for the positive follow-up. We are glad that the additional clarifications and robustness analyses addressed your concerns. We will make these points clearer in the revision.

---

### Official Review · Reviewer_WQ1f · 2026-03-13

**Soundness:** 2
**Presentation:** 3
**Significance:** 3
**Originality:** 3
**Overall Recommendation:** 4
**Confidence:** 3

**Summary:**

To overcome reliability inconsistencies among judge LLMs within the LLM-as-a-judge framework, this paper introduces a ranking method that enhances the Bradley–Terry–Luce model with parameters accounting for individual judge discrimination. The approach derives both model performance and judge accuracy directly from pairwise comparisons, bypassing the need for reference labels. The methodology guarantees identifiability while confirming the statistical soundness of its estimator, which supports the determination of confidence ranges for performance scores and relative rankings. Testing across standard benchmarks and a dedicated dataset verifies that the technique aligns more closely with human evaluations, utilizes data more efficiently than standard approaches, and yields reliable uncertainty measures for model rankings.

**Compliance With Llm Reviewing Policy:**

Affirmed.

**Final Justification:**

The rebuttal has addressed my main concerns, except that the claimed advantage in mixed-quality judge settings need further concrete empirical evidence on realistic benchmarks.

**Key Questions For Authors:**

(1) How empirically significant is judge heterogeneity for existing unweighted ranking frameworks on large-scale LLM leaderboards?

(2) How does the framework distinguish genuinely high judge discrimination from widespread but biased consensus without ground-truth labels?

(3) How does the model account for task-dependent or capability-specific judge performance across different domains?

(4) What guidance does the framework provide for sample complexity, cost-effective judge configuration, and balancing diversity versus redundancy?

(5) Under what practical conditions does the proposed framework outperform existing unweighted methods in real-world LLM evaluation?

**Limitations:**

yes

**Strengths And Weaknesses:**

This paper addresses a critical issue: how to evaluate LLMs on open-ended tasks in a stable and reliable manner. The authors clearly define the problem and propose a judge-aware ranking framework for aggregating pairwise comparisons produced by heterogeneous LLM judges. The work is validated through both a simulation study and experiments on LLM ranking benchmarks, demonstrating both theoretical correctness and experimental superiority. However, I have the following concerns.

The paper posits that ignoring judge heterogeneity can lead to systematic bias and increasing “confidence in error” as the number of comparisons grows. While this motivation is theoretically sound, it would benefit from stronger empirical grounding. In particular, the paper does not quantitatively demonstrate the extent to which existing unweighted ranking frameworks, such as standard Bradley–Terry or Elo-style systems, actually fail or produce misleading outcomes on current large-scale LLM leaderboards. As a result, the practical severity and prevalence of the claimed issue remain somewhat abstract rather than empirically established.

The related work section provides a thorough treatment of statistical pairwise ranking models and crowdsourcing-based annotator modeling. However, the discussion of modern open-ended LLM evaluation paradigms and benchmarks remains relatively limited.

A more fundamental concern relates to the distinction between “truth” and “consensus” in the absence of ground-truth labels. The estimation of judge discrimination parameters implicitly relies on consistency across judges, which may resemble a form of consensus-based aggregation. In practical LLM evaluation, however, the majority of judges may share systematic biases (e.g., verbosity or stylistic preferences), while a minority of judges may provide more accurate assessments. Without external ground truth, it remains unclear how the framework distinguishes between genuinely high discrimination and widespread but biased consensus. Similarly, although Wald confidence intervals are provided, their correctness cannot be independently validated when no ground-truth benchmark exists, raising questions about their interpretability beyond the model’s internal assumptions.

The framework also rests on several strong modeling assumptions that may limit its applicability in practice. Judge reliability is represented by a single, global scalar parameter, which does not capture the task-dependent or capability-specific nature of LLM judgment. A judge may exhibit strong discrimination in one domain (e.g., coding) but weak discrimination in another (e.g., creative writing), a nuance that cannot be expressed by the current formulation. Additionally, the assumption of independent judges may be violated in realistic settings, where LLM judges from similar model families exhibit correlated errors due to shared training data or architectural similarities.

From a practical standpoint, the paper provides limited guidance on sample complexity and deployment. Although an asymptotic O(1/T) convergence rate is established, there is no concrete methodology for estimating the number of comparisons required in realistic evaluation scenarios. The framework also does not address how to design cost-effective judge configurations or balance diversity and redundancy under constrained evaluation budgets.

Finally, while the experimental results show strong alignment with established benchmarks such as MT-Bench, this raises questions about the contribution of the proposed approach in practice. If similar rankings can often be recovered by existing unweighted methods, the conditions under which the proposed framework offers decisive advantages are not fully clarified. Moreover, validation frequently relies on heuristic expectations, e.g., larger or newer models being superior, which may not always hold. The simulation studies, while useful for illustrating theoretical properties, are generated under assumptions aligned with the model itself and may not fully reflect the complexity of real-world human preference data.

---

> ### Author Rebuttal · Authors · 2026-03-31
>
> Q1:We thank the reviewer for raising this point. To clarify, we do not claim that unweighted aggregation fails in all evaluation scenarios. For example, in nearly homogeneous judge pools where all judges are highly capable, equal-weight aggregation may already perform well. In such settings, the advantage of our weighted framework may be limited, especially since the unweighted model involves fewer parameters.\
> However, such nearly homogeneous judge pools are often unrealistic in practice, and the empirical significance of judge heterogeneity becomes much more pronounced in mixed-quality judge settings. In large-scale evaluation pipelines, it is common to rely on a diverse pool of judges (e.g., different LLMs or crowdsourced annotators) in order to balance quality and cost. In such cases, unweighted aggregation assigns the same influence to highly reliable judges and noisy ones, which can increase the variance of pairwise decisions and degrade the final ranking. \
> Q2:We agree that, without ground-truth labels, it is difficult to distinguish genuinely higher judge discrimination from broad but potentially biased consensus. In our framework, $\gamma_k$ measures the discrimination of judge $k$, namely sensitivity to latent quality differences. It should therefore be interpreted relative to the latent ranking learned by the model, not as an absolute measure of externally validated correctness. Our goal is not to certify which judges are objectively correct, but to provide a principled latent aggregation of heterogeneous judges when ground truth is unavailable. If trusted external references are available, they could be incorporated in an extended framework, but this is beyond the scope of the current submission.\
> Q3:We agree that a single global scalar $\gamma_k$ cannot fully capture task-specific or domain-dependent judge quality. In the current submission, we use $\sigma\left(\gamma_k(s_i-s_j)\right)$ as a parsimonious model to isolate the main effect of judge heterogeneity while maintaining identifiability. A natural extension is to allow judge discrimination to depend on domain or task features. Due to the rebuttal length limit, we refer to the second paragraph of our response to Reviewer 7mCp for the concrete parameterization and discussion. We will clarify this in the revision and add an empirical study if space permits.\
> Since the assumption of independent judges has been mentioned in the review, we want to clarify that our current formulation assumes conditional independence given the latent scores and judge-specific parameters, and therefore does not explicitly model correlated errors or shared biases among similar judges. A practical mitigation is to encourage judge diversity and avoid over-representation from the same model family.\
> Q4:We thank the reviewer for highlighting the importance of deployment guidance. We agree that sample complexity, cost-effective judge selection, and the diversity-redundancy tradeoff are important issues in LLM evaluation. Our paper focuses on aggregation rather than fully solving the broader design problem, but the framework can still provide useful guidance. First, although a universal static sample size is difficult to prescribe, our framework provides data-driven stopping guidance through uncertainty quantification: evaluators can monitor Wald confidence intervals for the estimated scores and stop once the desired precision is reached. Second, in mixed-quality judge settings, the estimated discrimination parameters $\gamma_k$ provide a quantitative measure of judge quality and can support cost-aware judge selection. If two judges have similar estimated discrimination but different costs, the cheaper one is naturally preferred. More broadly, the fitted model and its uncertainty estimates can help identify which judges or comparisons are most valuable for improving precision, and the fitted scores can serve as a low-cost screening signal for later high-quality evaluation. We do not claim to fully resolve optimal judge allocation, active sampling, or the broader experimental design problem, and leave these questions to future work.\
> Q5:As discussed in Q1, we do not expect our framework to outperform unweighted aggregation in every evaluation scenario. Its main practical advantage arises in mixed-quality judge settings, where substantial heterogeneity makes equal-weight aggregation more vulnerable to noisy or weak judges. When the judge pool contains both stronger and weaker judges, our approach explicitly models judge heterogeneity and down-weights less reliable judges, leading to a more stable and informative ranking. We will clarify in the revision that our method is most relevant in realistic evaluation pipelines where judge quality is heterogeneous, especially when high-quality judges are limited and evaluation must rely on a mixed-quality pool.

---

> > ### Author Rebuttal · Reviewer_WQ1f · 2026-04-03
> >
> > Thank you for your detailed rebuttal. I appreciate the clarification regarding the intended scope and the tempering of claims.

---

> > > ### Author Response · Authors · 2026-04-08
> > >
> > > Thank you for the positive follow-up. We are glad that our clarification regarding the intended scope and the corresponding tempering of claims helped address your concern.
> > >
> > > To further support this revised positioning, we also added an additional mixed-quality judge experiment. We constructed a heterogeneous judge pool consisting of 6 lower-quality judges and 2 higher-quality judges and evaluate the empirical impact of judge heterogeneity on unweighted ranking frameworks. We fit both weighted and unweighted models on the same set of pairwise comparisons, and evaluated their estimated scores against the ground-truth ranking using Pearson and Spearman correlations. See the table below for details:
> > >
> > > | Model | Pearson Corr| Spearman Corr |
> > > | - | - | - |
> > > | Unweighted | 0.8992 | 0.8316 |
> > > | Weighted   | 0.9394 | 0.9212 |
> > >
> > > We observe a clear and consistent improvement when accounting for judge heterogeneity: Pearson correlation increases by $0.0403$ and Spearman correlation increases by $0.0896$. This demonstrates that heterogeneous judge quality introduces non-trivial bias in unweighted aggregation, degrading ranking fidelity. Notably, the larger improvement in Spearman correlation indicates that modeling judge heterogeneity is particularly important for preserving the relative ordering of models, which is the primary objective of leaderboard construction. Overall, these results provide empirical support for the practical importance of judge heterogeneity and suggest that weighted frameworks can yield more reliable rankings than unweighted baselines, especially in mixed-quality judge settings.
> > >
> > > The detailed estimated results are shown in the table below:
> > >
> > > | Judge | $\gamma$ |
> > > | - | - |
> > > | Qwen/Qwen2.5-7B-Instruct-Turbo | 1.419 |
> > > | deepseek-ai/DeepSeek-R1-Distill-Llama-70B | 0.046 |
> > > | moonshot-v1-128k | 2.914 |
> > > | kimi-k2-thinking-turbo | 3.511 |
> > > | meta-llama/Llama-4-Scout-17B-16E-Instruct | 1.872 |
> > > | google/gemma-3n-E4B-it | 0.627 |
> > > | mistralai/Mixtral-8x7B-Instruct-v0.1 | 0.447 |
> > > | Qwen/Qwen3-235B-A22B-Instruct-2507-tput | 2.844 |
> > >
> > > The estimated results clearly demonstrate that weighted model successfully captures heterogeneity in judge quality through the learned scale parameters $\gamma_k$. In particular, kimi-k2-thinking-turbo ($\gamma = 3.511$) and moonshot-v1-128k ($\gamma = 2.914$) receive the two highest $\gamma$ values among all judges. Notably, these two judges were manually identified as high-quality judges a priori, and the model is able to recover this structure purely from pairwise comparison data.\
> > > From a modeling perspective, $\gamma_k$ scales the latent score difference $(s_i - s_j)$ in the likelihood, meaning that a larger $\gamma_k$ implies stronger discrimination ability and higher consistency in comparisons. Consequently, judges with higher $\gamma_k$ contribute more informative signals to the estimation procedure. In contrast, judges such as deepseek-ai/DeepSeek-R1-Distill-Llama-70B ($\gamma \approx 0.046$) are effectively down-weighted, as their comparisons are close to random noise and provide little useful ranking information.
> > >
> > > Overall, this experiment provides empirical evidence that the weighted model can automatically identify and amplify high-quality judges while suppressing noisy ones, without access to ground-truth labels. This highlights the importance of incorporating judge-specific scale parameters when addressing judge heterogeneity in large-scale LLM evaluation settings.
> > >
> > > We will include this experiment in the revision and make this practical regime and scope more explicit in the paper.

---

### Decision · Program_Chairs · 2026-04-30

**Decision:**

Accept (regular)

**Comment:**

The paper addresses an important problem in LLM-as-a-judge evaluation: different judges vary in reliability, yet most methods treat them equally. The proposed extension of the Bradley–Terry–Luce model with judge-specific discrimination parameters is well motivated, theoretically grounded, and supported by identifiability and asymptotic analysis. Experiments on simulations and real benchmarks show improved agreement with human preferences, better sample efficiency, and useful uncertainty estimates compared to standard unweighted methods. The main limitations are that the model uses only a single global reliability parameter per judge and is compared mainly against unweighted BTL rather than stronger judge-aware baselines. Authors addressed some major concerns during rebuttal, and are expected to integrate these changes in the final version.